# A Unified Framework for Forward and Inverse Problems in Subsurface Imaging using Latent Space Translations

**Naveen Gupta**[1]*, **Medha Sawhney**[1]*, **Arka Daw**[2]*, **Youzuo Lin**[3], **Anuj Karpatne**[1]
[1]Virginia Tech [2]Oak Ridge National Lab [3]UNC at Chapel Hill

## Abstract

In subsurface imaging, learning the mapping from velocity maps to seismic waveforms (forward problem) and waveforms to velocity (inverse problem) is important for several applications. While traditional techniques for solving forward and inverse problems are computationally prohibitive, there is a growing interest in leveraging recent advances in deep learning to learn the mapping between velocity maps and seismic waveform images directly from data. Despite the variety of architectures explored in previous works, several open questions remain unanswered such as the effect of latent space sizes, the importance of manifold learning, the complexity of translation models, and the value of jointly solving forward and inverse problems. We propose a unified framework to systematically characterize prior research in this area termed the Generalized Forward-Inverse (GFI) framework, building on the assumption of manifolds and latent space translations. We show that GFI encompasses previous works in deep learning for subsurface imaging, which can be viewed as specific instantiations of GFI. We also propose two new model architectures within the framework of GFI: Latent U-Net and Invertible X-Net, leveraging the power of U-Nets for domain translation and the ability of IU-Nets to simultaneously learn forward and inverse translations, respectively. We show that our proposed models achieve state-of-the-art performance for forward and inverse problems on a wide range of synthetic datasets and also investigate their zero-shot effectiveness on two real-world-like datasets. The code is available at `https://github.com/KGML-lab/Generalized-Forward-Inverse-Framework-for-DL4SI`

## 1 Introduction

The goal of subsurface imaging is to identify the structure and geophysical properties of layers underneath the Earth's surface such as *velocity maps* ($v$), which is useful for a number of applications including energy exploration, carbon capture and sequestration, and developing earthquake early warning systems (Zhang et al., 2013). A typical approach for subsurface imaging is to conduct seismic surveys, where the elastic wave energy generated from one or more controlled sources is made to propagate through the Earth's layers that are refracted, reflected, and received as *seismic waveforms* ($p$) on the surface by multiple receivers. Mathematically, the relation between velocity maps and seismic waveforms is governed by the following acoustic wave equation:

$$\nabla^2 p(x, z, t) - \frac{1}{v^2(x, z)} \frac{\partial^2}{\partial t^2} p(x, z, t) = s(x, z, t), \tag{1}$$

where $x$, $z$, and $t$ represent the horizontal direction, depth, and time, respectively, and $\nabla^2$ is the Laplacian operator. Using Equation 1, the *forward problem* in subsurface imaging is to compute seismic waveforms $p$ given velocity maps $v$ by learning the forward mapping $f_{v \to p} : \mathcal{V} \to \mathcal{P}$. The *inverse problem*, also referred to as Full Waveform Inversion (FWI), is then to learn the reverse mapping $f_{p \to v} : \mathcal{P} \to \mathcal{V}$ for inferring velocity maps $v$ given observations of $p$.

---

*Equal Contribution. Corresponding authors: *dawa@ornl.gov*, {*naveengupta, medha*}@*vt.edu*

While solving the forward problem requires expensive numerical solutions (Tago et al., 2012) of Equation 1, learning the inverse is even more computationally demanding as it involves iterating over velocity maps $v$ to minimize the difference between simulated waveforms from the forward model and ground-truth observations of $p$ (Tarantola, 1988). As a result, traditional techniques for solving forward and inverse problems in subsurface imaging are difficult to scale in operational settings at the desired resolutions. In response, there is a growing interest in the geophysics community to leverage deep learning for subsurface imaging (DL4SI), in particular, using image-to-image-translation methods to map seismic waveforms to velocity maps and back directly from data.

For example, several deep learning methods have recently been proposed for FWI using encoder-decoder architectures such as InversionNet (Wu & Lin, 2019) and VelocityGAN (Zhang et al., 2019), where the encoder maps seismic data to latent feature representations that are decoded back to the velocity domain (Lin et al., 2023). Techniques for solving the forward problem include the seminal work on physics-informed neural networks (PINNs) (Raissi et al., 2019) and recent works such as Fourier Neural Operators (Li et al., 2020). A recent work has also pursued solving both forward and inverse problems together using a common architecture, Auto-Linear (Feng et al., 2024), where two auto-encoders are first separately trained to project velocity and waveform fields to higher-dimensional manifolds, which are then linked via linear transforms for manifold translation.

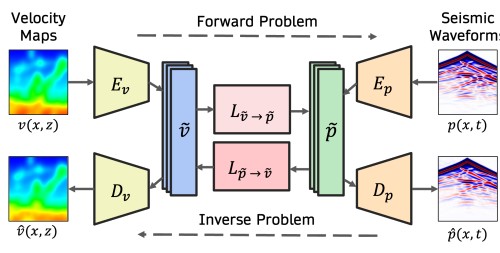

Figure 1: A unified framework for solving forward and inverse problems in subsurface imaging.

Despite the variety of architectures explored in previous works in DL4SI, several open questions still remain unanswered such as: (1) How does the *size of latent spaces* in encoder-decoder frameworks influence the quality of domain translations in DL4SI? While previous works such as InversionNet used lower-dimensional embeddings in their bottleneck layers, recent methods such as Auto-Linear advocate for higher-dimensional mappings. (2) Do we need *complex architectures* for latent space translations or are linear transforms sufficient? (3) What is the role of *manifold learning* in DL4SI where latent spaces are used not just for translation but also for reconstruction? (4) Can we simultaneously solve *both forward and inverse problems* using joint frameworks?

To answer these questions, we propose a unified framework to systematically characterize prior research in DL4SI termed the *Generalized Forward-Inverse* (GFI) framework (see Figure 1). The GFI framework builds on two key philosophies that are at the core of prior research in DL4SI. First, we assume that both velocity $v$ and seismic waveforms $p$ can be projected to latent space manifolds $\tilde{v}$ and $\tilde{p}$ using encoder-decoder pairs that allow reconstruction of the original domains, referred to as the *manifold assumption*. Second, we assume that it is possible to learn bidirectional translations between the two manifolds $\tilde{v}$ and $\tilde{p}$, referred to as the *latent space translation assumption*. We show that GFI encompasses previous works in DL4SI including InversionNet, VelocityGAN, and Auto-Linear, which can be viewed as specific instantiations of GFI. This unifying perspective helps us analyze the field of DL4SI going beyond specific architecture choices and also discover novel formulations in DL4SI such as those explored in this paper.

In particular, we propose two new architectures within the framework of GFI: *Latent U-Net* (see Figure 2a) and *Invertible X-Net* (see Figure 2b). Latent U-Net uses two separate U-Nets (Ronneberger et al., 2015) to model latent space translations in both directions. This takes inspiration from the success of U-Nets in mainstream computer vision applications to explore their benefits in DL4SI. Invertible-XNet uses Invertible U-Net (IU-Net) (Etmann et al., 2020) to simultaneously learn both forward and inverse translations in the latent space using a single architecture. Invertible U-Nets builds on the idea of Invertible Neural Networks (INNs) (Ardizzone et al., 2018) to learn bijective mappings between input and output domains in DL4SI. We show that both Latent U-Net and Invertible X-Net achieve state-of-the-art (SOTA) performance for forward and inverse problems on a wide range of benchmark datasets commonly used in DL4SI. We show that jointly solving forward and inverse problems using Invertible X-Net especially helps in improving the accuracy on the forward problem. We also investigate the zero-shot effectiveness of our proposed approaches on synthetic and real-world-like datasets, contributing to novel insights in the field of DL4SI.

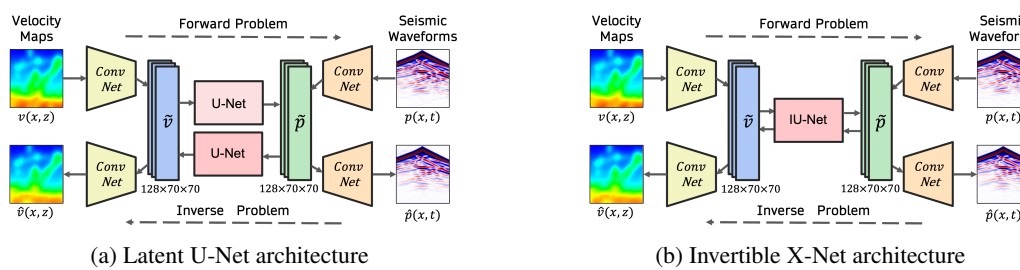

Figure 2: Schematic representation of proposed Latent U-Net and Invertible X-Net model.

## 2 RELATED WORKS

### 2.1 DEEP LEARNING FOR SUBSURFACE IMAGING (DL4SI)

Solving inverse problems in DL4SI has received considerable attention in recent years, thanks to the seminal work by Deng et al. (2022) in creating OpenFWI, an open-access synthetic dataset involving paired examples of velocity and waveforms with varying levels of complexity. This has enabled the formulation of FWI as an image-to-image translation problem, where the relationship between waveforms and velocity is learned through supervised, unsupervised, or self-supervised methods (Lin et al., 2023). One of the early works in supervised learning for FWI includes InversionNet (Wu & Lin, 2019), which uses an encoder-decoder architecture to project waveforms into flattened vector representations acting as the bottleneck layer, which are decoded back to velocity maps. VelocityGAN (Zhang et al., 2019) builds on the idea of InversionNet by using an adversarial training scheme for learning the model parameters along with prediction loss. These works have also been extended for estimating 3D velocity fields in Feng et al. (2021).

Since ground-truth observations of velocity maps are difficult to obtain in real-world settings, a number of unsupervised and self-supervised formulations have also been explored for FWI. This includes the framework of UPFWI (Jin et al., 2021) that uses the supervision of a physics-based forward model of the wave equation to enforce cycle-consistency with a data-driven inverse model. While the training of UPFWI is completely unsupervised and achieves comparable performance as InversionNet in some settings, it requires computationally expensive simulations of a forward model at every training epoch, restricting its scalability. Recent advancements, such as IFWI Du et al. (2024), FWIPLR Sun et al. (2023), and FWIRLG Yang & Ma (2023), integrate physics-based constraints and deep learning to tackle challenges like low-quality data and geological complexity. A recent line of work in semi-supervised learning for FWI involves the framework of Auto-Linear (Feng et al., 2024). In this framework, both velocity and waveforms are first mapped to higher-dimensional manifolds using two auto-encoders separately trained to minimize reconstruction loss in their respective domains. This is followed by a supervised learning of translations between the manifolds using two independent linear transforms (one for forward and one for inverse), building on previous observations of linearity in high-dimensional spaces for FWI (Feng et al., 2022).

### 2.2 DOMAIN TRANSLATION METHODS

The goal in domain translation (Isola et al., 2017; Zhu et al., 2017; Choi et al., 2018) is to learn the mapping between input and output domains (e.g., the CT scan of a patient and the organ segmentation from the scan), either in a paired or unpaired setting. A recent work on Latent Space Mapping (LSM) (Mayet et al., 2023) has introduced a unifying framework for domain translation based on the manifold assumption (i.e., both input and output domains reside in low-dimensional manifolds learned by minimizing reconstruction loss) and shared latent space assumption (i.e., both manifolds reside in a shared latent space making it possible to translate from one manifold to another). We take inspiration from LSMs in domain translation to develop our proposed framework of GFI.

## 3 PROPOSED METHODOLOGIES

### 3.1 GENERALIZED FORWARD INVERSE FRAMEWORK

As shown schematically in Figure 1, our proposed framework of GFI unifies the learning of the forward problem $f_{v \to p} : \mathcal{V} \to \mathcal{P}$ and the inverse problem $f_{p \to v} : \mathcal{P} \to \mathcal{V}$ using a common archi-

tecture involving latent space translations. The GFI framework is inspired by two key assumptions. First, according to the *manifold assumption*, we assume that the velocity maps $v \in \mathcal{V}$ and seismic waveforms $p \in \mathcal{P}$ can be projected to their corresponding latent space representations, $\tilde{v}$ and $\tilde{p}$, respectively, which can be mapped back to their reconstructions in the original space, $\hat{v}$ and $\hat{p}$. Note that the sizes of the latent spaces can be smaller or larger than the original spaces. Further, the size of $\tilde{v}$ may not match with the size of $\tilde{p}$ (e.g., in Auto-Linear). Second, according to the *latent space translation assumption*, we assume that the problem of learning forward and inverse mappings in the original spaces of velocity and waveforms can be reformulated as learning translations in their latent spaces. We provide theoretical justifications for the existence of these latent space translations in the Appendix A. Note that these assumptions can also be relaxed in specific instantiations of GFI.

Formally, let $\mathtt{E}_v : \mathcal{V} \rightarrow \tilde{\mathcal{V}}$ and $\mathtt{E}_p : \mathcal{P} \rightarrow \tilde{\mathcal{P}}$ be the encoders that map velocity and waveforms to their respective latent representations, $\tilde{v} = \mathtt{E}_v(v)$ and $\tilde{p} = \mathtt{E}_p(p)$. Similarly, let $\mathtt{D}_v : \tilde{\mathcal{V}} \rightarrow \mathcal{V}$ and $\mathtt{D}_p : \tilde{\mathcal{P}} \rightarrow \mathcal{P}$ be the decoders that map the latent representations back to the original spaces. Additionally, let $L_{\tilde{v} \rightarrow \tilde{p}} : \tilde{\mathcal{V}} \rightarrow \tilde{\mathcal{P}}$ and $L_{\tilde{p} \rightarrow \tilde{v}} : \tilde{\mathcal{P}} \rightarrow \tilde{\mathcal{V}}$ be the latent space translations between $\tilde{v}$ and $\tilde{p}$ and vice versa. The forward and inverse mappings between velocity $v$ and waveform $p$ can then be formulated as:

$$\hat{p} = f_{v \rightarrow p}(v) = \mathtt{D}_p \circ L_{\tilde{v} \rightarrow \tilde{p}} \circ \mathtt{E}_v(v) \tag{2}$$

$$\hat{v} = f_{p \rightarrow v}(p) = \mathtt{D}_v \circ L_{\tilde{p} \rightarrow \tilde{v}} \circ \mathtt{E}_p(p) \tag{3}$$

**Learning GFI with Paired Data:** Let us assume that we have a paired i.i.d. dataset $(v, p) \sim q(v, p)$, where $q(v, p)$ is a joint probability distribution. We can then learn the parameters of the GFI framework in a supervised manner by optimizing the following loss functions:

$$\min_{\theta_{\mathtt{E}_v}, \theta_{L_{\tilde{v} \rightarrow \tilde{p}}}, \theta_{\mathtt{D}_p}} \mathbb{E}_{q(v,p)} \mathcal{L}(\mathtt{D}_p \circ L_{\tilde{v} \rightarrow \tilde{p}} \circ \mathtt{E}_v(v), p) = \mathcal{L}(\hat{p}, p) \tag{4}$$

$$\min_{\theta_{\mathtt{E}_p}, \theta_{L_{\tilde{p} \rightarrow \tilde{v}}}, \theta_{\mathtt{D}_v}} \mathbb{E}_{q(v,p)} \mathcal{L}(\mathtt{D}_v \circ L_{\tilde{p} \rightarrow \tilde{v}} \circ \mathtt{E}_p(p), v) = \mathcal{L}(\hat{v}, v) \tag{5}$$

where $\mathcal{L}$ denotes a loss function such as the Mean Squared Error (MSE) loss, Mean Absolute Error (MAE) loss, or even an adversarial loss if trained with a discriminator.

### 3.2 Prior Works in DL4SI as Special Cases of GFI

In this section, we revisit previous works in DL4SI and describe them using GFI terminologies, i.e., *modeling mode* (forward/inverse), *size of latent space*, whether they use *manifold learning* or not, and the nature of *latent space translations*. Table 1 summarizes prior works as specific realizations of the unifying GFI Framework. Schematic illustrations of all prior works in the framework of GFI are provided in the Appendix Figure 9.

Table 1: Characterizing prior and proposed works using GFI Framework.

| Model | Modeling Mode | Manifold Learning | Latent Space Translation | Sizes of Latent Spaces |
|---|---|---|---|---|
| InversionNet | Inverse | No | Identity | Low |
| VelocityGAN | Inverse | No | Identity | Low |
| FNO | Forward | No | Fourier Layers | N/A |
| Auto-Linear | Forward and Inverse Disjointly | Yes | Linear | High and Different |
| Latent U-Net (ours) | Forward and Inverse Disjointly | Possible | U-Net | Low and Same |
| Invertible X-Net (ours) | Forward and Inverse Simultaneously | Implicit | IU-Net | Low and Same |

**InversionNet and VelocityGAN:** Both InversionNet and VelocityGAN operate exclusively in the *inverse mode*, mapping seismic waveforms to velocity maps. They utilize the same encoder-decoder architecture, with the encoder $\mathtt{E}_p$ projecting seismic waveforms $(p)$ to a lower-dimensional latent space and the decoder $\mathtt{D}_v$ reconstructing the velocity maps $(v)$ from this representation. The *sizes of the latent spaces* are low-dimensional, determined by the bottleneck layer of the U-Net architecture.

In terms of *latent space translation*, these models rely on an identity mapping in the bottleneck layer, $L_{\tilde{p}\to\tilde{v}} = I$. However, we can also interpret them as having a U-Net for translation, if we view the first few and last few layers as the encoder-decoder pair while the intermediate layers perform translation. The sizes of the latent spaces are low-dimensional and they do not perform *manifold learning* since the latent spaces of $\tilde{p}$ and $\tilde{v}$ are solely used for translation and not for reconstruction.

**Auto-Linear:** Auto-Linear operates in both *forward and inverse modes*, using two disjoint latent space translations. It follows a two-stage approach: first, encoder-decoder pairs ( $E_v$, $D_v$, $E_p$ and $D_p$) are independently trained using reconstruction loss to project $v$ and $p$ into high-dimensional latent spaces ($\tilde{v}$ and $\tilde{p}$), where $dim(\tilde{v}) \neq dim(\tilde{p})$. Second, linear transforms $L_{\tilde{v}\to\tilde{p}}$ and $L_{\tilde{p}\to\tilde{v}}$ are trained for translation, keeping the encoders and decoders frozen. Auto-Linear leverages the *manifold assumption* as the latent manifolds $\tilde{v}$ and $\tilde{p}$ are useful for reconstruction. The *sizes of the latent spaces* are higher-dimensional, designed specifically to facilitate linear mappings.

### 3.3 PROPOSED MODEL: LATENT U-NET

We propose a novel architecture to solve forward and inverse problems using two latent space translation models implemented using U-Nets, termed *Latent U-Net*. As shown in Figure 2a, Latent U-Net uses ConvNet backbones for both encoder-decoder pairs: ($E_v$, $D_v$) and ($E_p$, $D_p$), to project $v$ and $p$ to lower-dimensional representations. We also constrain the sizes of the latent spaces of $\tilde{v}$ and $\tilde{p}$ to be identical, i.e., $\dim(\tilde{v}) = \dim(\tilde{p})$, so that we can train two separate U-Net models to implement the latent space mappings $L_{\tilde{v}\to\tilde{p}}$ and $L_{\tilde{p}\to\tilde{v}}$. Note that InversionNet can also be viewed as a special case of Latent U-Net in inverse-only mode with subtle architectural differences such as absence of skip connections, design of encoder/decoder and translation layers, and different dimensions of $\tilde{v}$ and $\tilde{p}$.

We train Latent U-Net in an end-to-end manner to directly learn latent spaces useful for translation. This is in contrast to the two-stage training process employed in Auto-Linear where latent spaces are first pre-trained for reconstruction, followed by translation. Specifically, for the forward problem, the components $D_p$, $L_{\tilde{v}\to\tilde{p}}$, and $E_v$ are trained together by optimizing Equation 4, while for the inverse problem, the components $D_v$, $L_{\tilde{p}\to\tilde{v}}$, and $E_p$ are trained together by optimizing Equation 5. Since the training of forward and inverse modeling components of Latent U-Net are disjoint from each other, Latent U-Net does not explicitly learns manifolds useful for reconstruction. However, it is possible to modify the training objective of Latent U-Net by adding reconstruction losses in both forward and inverse problems to perform manifold learning. See Appendix sections D.3.1 for details

### 3.4 PROPOSED MODEL: INVERTIBLE X-NET

We propose another novel architecture within the GFI framework termed *Invertible X-Net* to answer the question: "can we learn a single latent space translation model that can simultaneously solve both forward and inverse problems?" Recently, Invertible Neural Networks (INNs) (Ardizzone et al., 2018) have been introduced as a specialized class of neural networks capable of learning bijective mappings between inputs and outputs. INNs are inherently invertible by design; once the INN is trained to learn a forward mapping $f : \mathcal{X} \to \mathcal{Y}$ between inputs $x \in \mathcal{X}$ and outputs $y \in \mathcal{Y}$, the inverse mapping $f^{-1} : \mathcal{Y} \to \mathcal{X}$ can be obtained for free. A variant of INN has also been developed for image-to-image translation problems using the U-Net backbone, termed IU-Net (Etmann et al., 2020). However, one of the key constraints of INN and IU-Net is that they require $\dim(\mathcal{X}) = \dim(\mathcal{Y})$, which is difficult to ensure in subsurface imaging problems (since velocity and waveforms reside in entirely different spaces).

This is where we can leverage the GFI framework to employ IU-Net in the latent spaces of velocity and waveforms, which can be constrained to be of the same size (just like Latent-UNets), i.e., $\dim(\tilde{v}) = \dim(\tilde{p})$. By using a single IU-Net, we can simultaneously learn both forward and inverse translations in the latent space, $L_{\tilde{p}\to\tilde{v}}$ and $L_{\tilde{v}\to\tilde{p}}$. We refer to this formulation as Invertible "X"-Net (owing to the cross-shape of translations between the latent spaces) defined as follows:

$$\tilde{p} = L_{\tilde{v}\to\tilde{p}}(\tilde{v}) = f_{\text{IU-Net}}(\tilde{v}); \quad \tilde{v} = L_{\tilde{p}\to\tilde{v}}(\tilde{p}) = f_{\text{IU-Net}}^{-1}(\tilde{p}) \tag{6}$$

$$\hat{p} = D_p \circ f_{\text{IU-Net}}(\tilde{v}) \circ E_v(v); \quad \hat{v} = D_v \circ f_{\text{IU-Net}}^{-1}(\tilde{p}) \circ E_p(p) \tag{7}$$

The objective function for training Invertible X-Net involves prediction losses on both velocity maps and waveforms defined as follows:

$$\mathcal{L}_{\text{X-Net}} = \mathcal{L}_{\text{forward}} + \mathcal{L}_{\text{inverse}} \tag{8}$$

$$= \mathbb{E}_{q(v,p)}\mathcal{L}(\text{D}_p \circ f_{\text{IU-Net}} \circ \text{E}_v(v), p) + \mathbb{E}_{q(v,p)}\mathcal{L}(\text{D}_v \circ f_{\text{IU-Net}}^{-1} \circ \text{E}_p(p), v) \tag{9}$$

Note that Invertible X-Nets facilitate *bidirectional training* as both the forward and inverse modeling components are trained simultaneously. As a result, both encoder-decoder pairs of Invertible X-Nets implicitly learn manifolds useful for reconstruction. Also, Invertible X-Net offers several key advantages over baselines. First, it simultaneously addresses both the forward and inverse problems within a single model architecture, whereas other baselines typically require training separate models for each task (e.g., Latent U-Net and Auto-Linear), leading to greater parameter efficiency. Additionally, the use of IU-Net ensures that the mappings between the latent spaces of velocity maps and seismic waveforms are bijective, guaranteeing a one-to-one mapping between these representations – a property not necessarily true for other models such as Latent U-nets and Auto-Linear. Further, the bidirectional training of the forward and inverse problems introduces a strong regularization effect as the gradients of the forward loss $\mathcal{L}_{\text{forward}}$ affects the parameters of $f_{\text{IU-Net}}$, thereby affecting both forward and inverse performance. Similarly, the gradients of the inverse loss $\mathcal{L}_{\text{inverse}}$ also impact both forward and inverse performance by updating the parameters of $f_{\text{IU-Net}}$. Further details about the Invertible X-Net architecture are provided in Appendix Section D.3.2.

**Cycle Loss for Training Invertible X-Net:** Another advantage of Invertible X-net is that the architecture can be trained with unpaired examples using cycle-loss consistency (Zhu et al., 2017). We consider both variants of Invertible X-Net (with and without cycle loss) in our experiments to demonstrate its effect on generalization performance.

## 4 EXPERIMENTAL SETUP

### 4.1 DATASETS

We consider the OpenFWI collection of datasets (Deng et al., 2022), comprising multi-structural benchmark datasets for DL4SI grouped into: Vel, Fault, and Style Families. The Vel Family includes four datasets with simple geological patterns, while the Fault Family contains four datasets with fault-like deformations, presenting greater modeling challenges. The Style Family uses style transfer methods to create complex geological settings. The waveform data is structured as (# source $\times$ recording time $\times$ receiver length), resulting in a shape of $(5 \times 1000 \times 70)$ for five uniformly spaced seismic sources and 70 receivers over 1000 milliseconds. In contrast, the velocity maps are represented as ($1 \times$ depth $\times$ receiver length), with a shape of $(1 \times 70 \times 70)$ reflecting spatial dimensions of depth and horizontal coverage. We also conduct out-of-distribution evaluations on the Marmousi (Martin et al., 2006) and Overthrust datasets (Aminzadeh et al., 1996). They are standard benchmarks that closely mimic real-world scenarios and exceed the complexity of OpenFWI datasets.

### 4.2 MODEL ARCHITECTURES

**Latent U-Net and Invertible X-Net:** We use two variants for Latent U-Net: (1) (Small) with 17M parameters, and (2) (Large) with 33M parameters. For Invertible X-Net, we used an architecture with 24M parameters trained with and without cycle loss. See Appendix D.1 and D.3 for details. **Baselines:** For the inverse problem, we considered InversionNet, VelocityGAN, and Auto-Linear as baselines. We obtained their trained model checkpoints to reproduce their results. For the forward problem, we implemented FNO (Li et al., 2020) and a U-Net model (termed WaveformNet) as baseline models (see Appendix C for details), along with using Auto-Linear.

**Training Details:** The input seismic waveforms were standardized using a StandardScaler to center the data with a zero mean and unit standard deviation, while the velocity data were scaled through MinMax normalization. See Appendix D.2 for details on hyper-parameter settings. To make a fair comparison with baseline methods, we kept the original train-test splits and reported results in the unnormalized space of both velocity and waveforms. We used Mean Absolute Error (MAE), Mean Square Error (MSE), and Structured Similarity (SSIM) as evaluation metrics since neither metric alone is fully comprehensive. MAE captures pixel-level accuracy while SSIM highlights structural similarity.

## 5 RESULTS AND DISCUSSIONS

**Quantitative Comparisons:** Figure 3a compares SSIM on velocity predictions while solving the inverse problem across different OpenFWI datasets. We can see that our proposed models, which capture complex non-linear relationships in the latent space using U-Net and IU-Net architectures, consistently outperform baseline methods InversionNet, VelocityGAN, & Auto-Linear across majority of datasets. Overall, the large Latent U-Net model shows the best performance in inverse modeling followed by the small Latent U-Net model. On complex datasets such as Style family and CFB, Invertible X-Net with and without cycle loss are also comparable to the small Latent U-Net model. In addition to SSIM, we also evaluate baselines using other metrics reported in Table 7(Appendix E.1) and compare MAE across models in Appendix figure 3a.

Figure 3b compares the performance of baseline methods on the forward problem (see Appendix E.2 Table 9 for other evaluation metrics). We can see that both Latent U-Net (Large) and Invertible X-Net show better performance than most baselines across all datasets. However, in contrast to the trends observed in the inverse problem, we observe that Invertible X-Net with and without cycle loss shows the best performance compared to Latent U-Net models. We further notice that for simple datasets such as FVA and FFA, FNO has significantly higher SSIM compared to other models. We later qualitatively explain this pattern by showing that FNO tends to capture dominant modes of the waveform distribution while neglecting subtler reflection patterns.

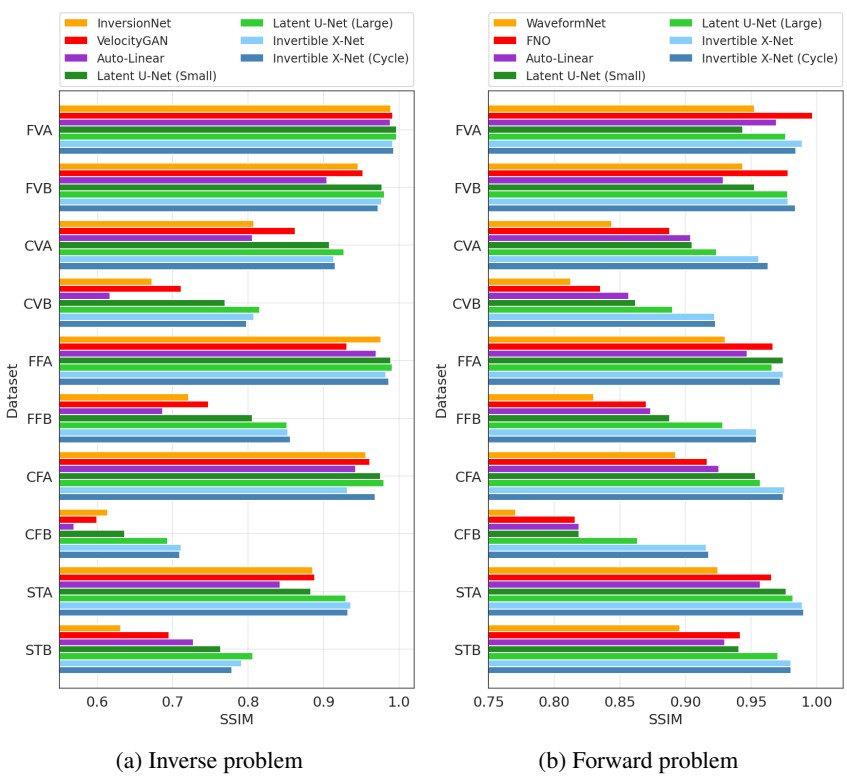

(a) Inverse problem         (b) Forward problem

Figure 3: Comparison of Latent U-Nets (Small and Large), Invertible X-Net, Invertible X-Net (Cycle) with different baseline methods across different OpenFWI datasets.

**Qualitative Comparisons:** In Figure 4a, we compare inverse model predictions on three datasets, CVB, CFB, and STA, choosing one from each family of OpenFWI. We observe that our proposed methods (Latent U-Net and Invertible X-Net) produce considerably better velocity maps than the two SOTA baselines, InversionNet and Auto-Linear. We chose these 3 datasets as they are complex and have high heterogeneity as the velocity rapidly changes in all directions. For heterogeneous velocity maps, the predictions of baseline methods are smooth and they fail to delineate sharp changes in velocities. On the other hand, Latent U-Net and Invertible X-Net are able to predict better velocity fields capturing rapid variations in both shallow and deeper parts (highlighted using black-boxes in

the figure). Between Latent U-Net (Large) and Invertible X-Net, we can see that the performance of Latent U-Net is slightly better especially on CFB and STA datasets. Note that Invertible X-Net uses a smaller number of parameters (24M) compared to Latent U-Net (Large) (33M). We provide additional visualizations of more baseline methods and across other datasets in the Appendix H.

Figure 4b shows visualizations of model predictions for the forward problem on three datasets. Since these datasets are heterogeneous, the recorded seismic waveforms exhibit convoluted interference patterns. In particular, we can see that in all ground truth visualizations, there is a direct arrival in the shallower layers represented as a 'slanted line' signature. While capturing the direct arrival is relatively simple, the primary challenge in solving forward problems is to capture the subtler reflections in the deeper layers, where even small magnitudes of errors can result in large differences in inferred velocity maps. We can see that while all baseline methods are able to predict shallow waveforms easily (like direct arrivals), methods such as FNO and Auto-Linear are struggling to predict the seismic waveforms, especially in the deeper regions (highlighted using black-boxes in the Figure). On the other hand, our proposed models consistently improve results over baselines across all datasets. Additional visualizations for the forward problem are provided in the Appendix H. Building on these quantitative and qualitative results, we provide key insights on four open questions in the field of DL4SI in the following.

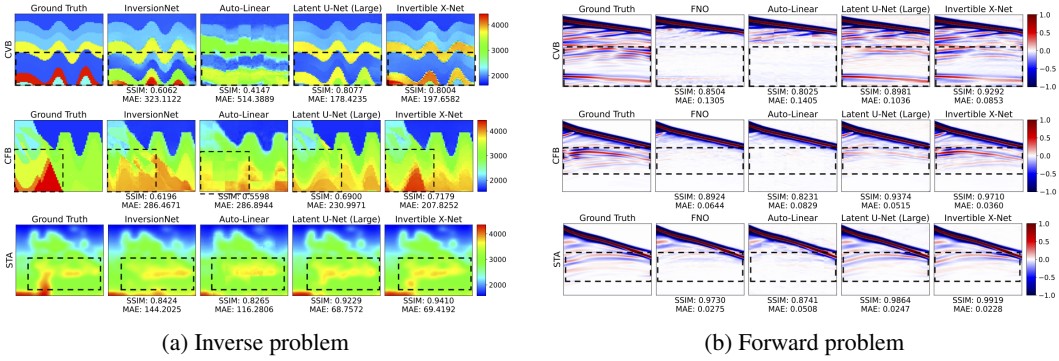

(a) Inverse problem        (b) Forward problem

Figure 4: Visualization of model predictions for inverse and forward problems across different Open-FWI datasets: CVB, CFB, STA.

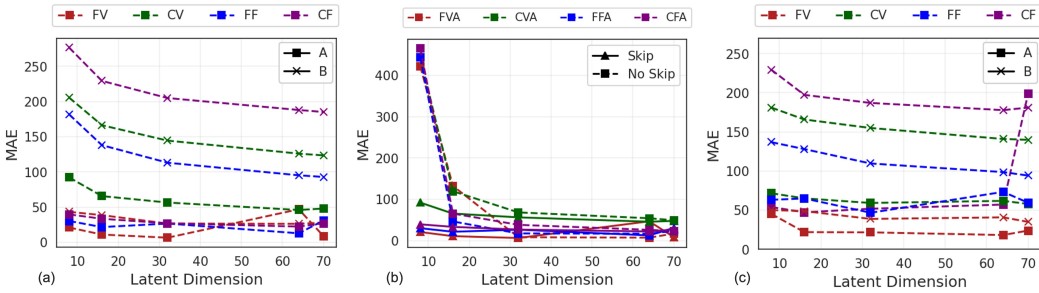

Figure 5: Effect of latent dimensions and skip connections (across latent dimensions) on the performance of Latent U-Net (Large) (a and b) and Invertible X-Net (c). 'A' and 'B' show 'low' and 'high' complexity scenarios for each data group.

## 5.1 KEY INSIGHTS ON OPEN QUESTIONS IN DL4SI

### 5.1.1 WHAT IS THE EFFECT OF LATENT SPACE SIZES ON TRANSLATION PERFORMANCE?

In Figure 5, we investigate the impact of varying latent space sizes on the quality of domain translations using the Latent U-Net (Large) (a) and Invertible X-Net(c) models on the inverse problem. We compare MAE by reducing latent space dimensions from $70 \times 70$ to $8 \times 8$. We can see that the quality of translations is not affected much for 'A'-family of OpenFWI datasets, while the 'B'-family shows monotonically increasing trend in MAE as we reduce latent dimension. Since the 'A'-family

was intentionally created to represent simpler velocity distributions, even the smallest latent space ($8 \times 8$) is sufficient to estimate velocity maps. However, the 'B'-family of datasets represent more complex velocity distributions, requiring a sufficiently larger latent space for the translation. This indicates that the ideal size of the latent space should be decided based on the requirements of the problem being solved.

### 5.1.2 DO WE NEED COMPLEX ARCHITECTURES FOR TRANSLATIONS?

We further study the role of model size on translation performance by comparing Latent U-Net (Small) and Latent U-Net (Large) with varying latent dimensions in Figure 18b (b). We find that for a given latent dimension, Latent U-Net (Large) outperforms Latent U-Net (Small), motivating architectures with sufficient complexity for domain translations. We further study the impact of using skip connections in the latent space translation models on varying latent space dimensions in Figure 5 (b). We can see that adding skip connections shows marginal improvements in performance for larger latent dimensions. However, when the number of dimensions is low (less than 20), we can see a sharp degradation in performance by removing skip-connections. This indicates that smaller latent representations are weaker and thus insufficient for domain translation in U-Net frameworks without being augmented with skip connections.

### 5.1.3 WHAT IS THE ROLE OF MANIFOLD LEARNING IN DL4SI?

Manifold learning for latent spaces in domain translation can follow three strategies: (1) the "Reconstruct then Translate" (Auto-Linear), where encoder-decoder pairs are first trained for reconstruction, followed by translation on frozen latent spaces; (2)"Directly Translate" strategy (Latent U-Net), which jointly trains encoder-decoder pairs and translation models to optimize latent spaces for translation; and (3) a hybrid "Reconstruct and Translate" approach, where the model is trained with both reconstruction and translation losses. To investigate their impact, we evaluate these approaches across varying training data fractions for OpenFWI families (Figure 6). Our findings reveal that the "Directly Translate" approach consistently outperforms the other two, highlighting that latent spaces optimized solely for translation are most effective. Hence, manifold learning, while beneficial for reconstruction,

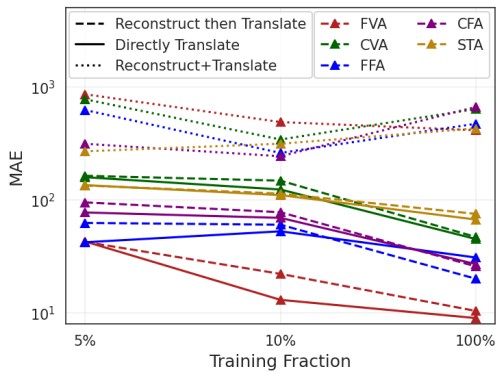

Figure 6: Comparison of Latent U-Net (Large) across three training objectives: direct translation, reconstruction followed by translation, and combined learning of both, evaluated at different fractions of training data.

may lead to suboptimal performance when translation is the primary goal. We see a similar trend for Latent U-Net (Small) and Invertible X-Net shown in Appendix F.3 Figure 22.

### 5.1.4 IS IT USEFUL TO JOINTLY SOLVE FORWARD AND INVERSE PROBLEMS?

As discussed in the quantitative comparisons of Latent U-Net (Large) and Invertible X-Net on the forward problem in Figure 3b, we see a surprising trend that Invertible X-Net generally outperforms Latent U-Net (Large) despite having smaller number of parameters. Analysis of its training behavior reveals that Invertible X-Net initially focuses on solving the inverse problem, gradually shifting to the forward problem in later epochs. (see Appendix E.3 Figures 12 and 13 for visualizations). This is facilitated by the combined loss function, where gradients from the inverse problem enhance forward problem optimization once a good velocity solution is achieved. This bidirectional learning enables Invertible X-Net to outperform Latent U-Net (Large). Further comparisons in Appendix E.3 confirm that Invertible X-Net trained only on the forward problem performs worse than both Latent U-Net (Large) and the jointly trained Invertible X-Net, emphasizing the advantage of joint optimization.With average ranks of 2.7 (MAE) and 1.9 (SSIM) for inverse problems, and 1.4 (MAE) and 1.1 (SSIM) for forward problems (Appendix Tables 7 and 9), Invertible X-Net offers a balanced and efficient solution for joint tasks with just 26M parameters. In contrast, Latent U-Net, while slightly better for inverse-only tasks, requires 70M parameters for both forward and inverse problems (35M x2), making Invertible X-Net the more practical and effective choice for the combined optimization.

## 5.2 ZERO-SHOT GENERALIZATION RESULTS

**Extrapolating across OpenFWI Datasets:** To assess the out-of-distribution generalizability of best-performing models on unseen datasets, in Figure 7, we evaluate zero-shot performance of models trained across all datasets (shown as rows) when tested across all datasets (shown as columns), for both forward and inverse problems. Here, the color intensity indicates difference in SSIM performance of two competing methods. We can see that Invertible X-Net outperforms other baselines for both forward and inverse problems across majority of train-test cases. The only exception is when the models are trained on FVB, a relatively simpler dataset than other data families such as CurveVel and FlatVel. From a geophysics perspective, we expect to have better generalizability for models trained on complex geological settings and tested on relatively simpler ones, as complex geological structures can be thought of as juxtapositions of simpler geological structures. Additionally, we compare the zero-shot generalization for all methods in detail in Appendix E.4 (Figures 14 - 17).

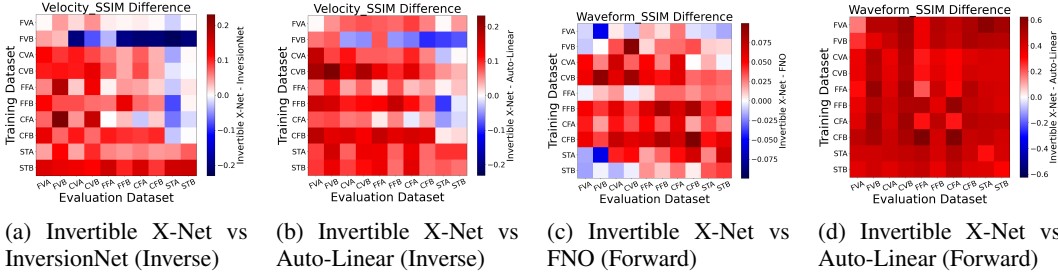

(a) Invertible X-Net vs InversionNet (Inverse)  (b) Invertible X-Net vs Auto-Linear (Inverse)  (c) Invertible X-Net vs FNO (Forward)  (d) Invertible X-Net vs Auto-Linear (Forward)

Figure 7: Zero-shot generalization comparison for both inverse and forward problem using SSIM difference across all OpenFWI datasets. Red color indicates our proposed method (Invertible X-Net) is better whereas blue color indicate otherwise.

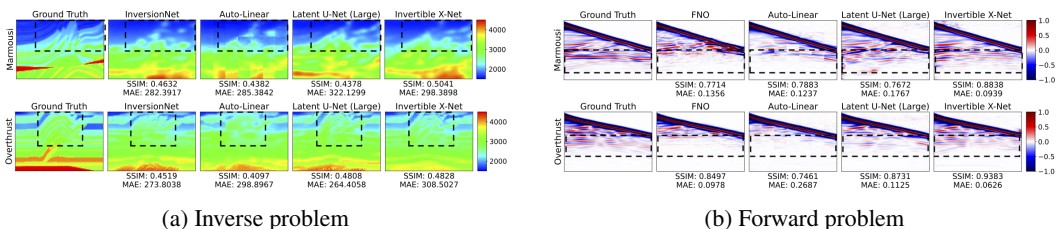

(a) Inverse problem  (b) Forward problem

Figure 8: Zero-shot generalization (both forward and inverse) on Marmousi and Overthrust datasets using models trained on STA and STB datasets, respectively.

**Extrapolating on Marmousi and Overthrust Datasets:** We show the zero-shot generalizability of our models on the more complex and real-world-like geological settings of Marmousi and Overthrust in Figure 8. Here, we chose models trained on the most complex OpenFWI datasets, i.e., STA (for Marmousi) and STB (for overthrust) datasets, and compared their zero-shot performance in predicting velocity maps (inverse problem). We observe that none of the baselines are able to invert velocity maps at high resolution. However, the predictions of Latent U-Net and Invertible X-Net are still able to delineate the sharp changes in velocity maps especially in the shallow part of the data. On the forward problem, our models are able to accurately predict seismic waveforms for both Marmousi and Overthrust data relative to baselines.

## 6 CONCLUSIONS AND FUTURE RESEARCH DIRECTIONS

In this work, we introduced a unified framework for solving both forward and inverse problems in subsurface imaging, termed the Generalized-Forward-Inverse (GFI) framework. Within this framework, we proposed two novel architectures, Latent U-Net and Invertible X-Net, that leverage the power of U-Nets and IU-Nets to perform latent space translations, respectively. Our study addresses several key questions left unanswered by prior research in DL4SI. Future research directions include training a single model for all datasets together, given that models trained on complex geological environments can effectively extrapolate on unseen simpler geological structures.

## 7 Reproducibility Statement

All the code required to train and evaluate the proposed methods, as well as the baselines, has been uploaded to an anonymous GitHub repository: `https://github.com/KGML-lab/Generalized-Forward-Inverse-Framework-for-DL4SI`. The data and corresponding processing code used in this work are sourced from the OpenFWI website: `https://openfwi-lanl.github.io/docs/data.html`. The dataset is further described in Appendix Section B and Table 2. The model architecture and hyper-parameter details for all proposed models and baselines are thoroughly discussed in Appendix Section D.

## 8 Ethics Statement

We would like to emphasize that the primary goal of this paper is to facilitate a scientific inquiry of DL4SI. While advancements in subsurface imaging have traditionally been used for energy exploration, there are several emerging applications of subsurface imaging with positive societal consequences such as Carbon Sequestration, Earthquake Hazard Warning, and Environmental Monitoring. Research in DL4SI thus has similar characteristics as research in mainstream AI applications such as computer vision and language modeling, with a diverse range of societal, environmental, and economic impacts.

## Acknowledgments

This work was supported in part by NSF awards IIS-2239328 and IIS-2107332. We are grateful to the Advanced Research Computing (ARC) Center at Virginia Tech for providing access to GPU compute resources for this project. This manuscript has been authored by UT-Battelle, LLC, under contract DE-AC05-00OR22725 with the US Department of Energy (DOE). The US government retains and the publisher, by accepting the article for publication, acknowledges that the US government retains a nonexclusive, paid-up, irrevocable, worldwide license to publish or reproduce the published form of this manuscript, or allow others to do so, for US government purposes. DOE will provide public access to these results of federally sponsored research in accordance with the DOE Public Access Plan ( https://www.energy.gov/doe-public-access-plan).

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

APPENDICES

## A  THEORETICAL JUSTIFICATIONS FOR LATENT SPACE TRANSLATION ASSUMPTION

In this Section, we provide two Lemmas to theoretically support our Latent Space Translation assumption for the Generalized-Forward Inverse (GFI) Framework.

**Lemma A.1** (Forward Latent Space Translation Assumption). *Let $f : \mathcal{V} \to \mathcal{P}$ be an arbitrary forward operator mapping velocity maps $\mathcal{V}$ to seismic waveforms $\mathcal{P}$. Let $E_v : \mathcal{V} \to \tilde{\mathcal{V}}$ and $D_v : \tilde{\mathcal{V}} \to \mathcal{V}$ denote the encoder and decoder for the velocity space $\mathcal{V}$, respectively. Similarly, let $E_p : \mathcal{P} \to \tilde{\mathcal{P}}$ and $D_p : \tilde{\mathcal{P}} \to \mathcal{P}$ denote the encoder and decoder for the seismic waveform space $\mathcal{P}$. Here, $\tilde{v} \in \tilde{\mathcal{V}}$ and $\tilde{p} \in \tilde{\mathcal{P}}$ represent the latent space encodings. If we assume that the auto-encoder for the velocity are optimal, i.e., $D_v \circ E_v(v) = \hat{v} \approx v$, then there exists a functional mapping in the latent space $L_{\tilde{v} \to \tilde{p}} : \tilde{\mathcal{V}} \to \tilde{\mathcal{P}}$.*

*Proof.* Given the forward operator $f : \mathcal{V} \to \mathcal{P}$, by definition, for any $v \in \mathcal{V}$, there exists $p \in \mathcal{P}$ such that $p = f(v)$.

Let the latent space representations $\tilde{v} \in \tilde{\mathcal{V}}$ and $\tilde{p} \in \tilde{\mathcal{P}}$ be defined by the auto-encoders as follows:

$$\tilde{v} = E_v(v), \quad \hat{v} = D_v(\tilde{v}) \tag{10}$$
$$\tilde{p} = E_p(p), \quad \hat{p} = D_p(\tilde{p}) \tag{11}$$

To construct the latent space mapping $\tilde{p} = L_{\tilde{v} \to \tilde{p}}(\tilde{v})$, consider the sequence of compositions involving the encoders, decoders, and the forward operator $f$:

$$\begin{aligned}
\tilde{p} &= E_p(p) \\
&= E_p(f(v)) \quad \text{(since } p = f(v)) \\
&= E_p(f(\hat{v})) \quad \text{(assuming reconstruction: } \hat{v} \approx v) \\
&= E_p(f(D_v(\tilde{v}))) \quad \text{(since } \hat{v} = D_v(\tilde{v}))
\end{aligned} \tag{12}$$

Thus, by definition of the composition of functions, the latent space mapping can be expressed as:

$$L_{\tilde{v} \to \tilde{p}} = E_p \circ f \circ D_v \tag{13}$$

$\square$

**Lemma A.2** (Inverse Latent Space Translation Assumption). *Let $f^{-1} : \mathcal{P} \to \mathcal{V}$ be an arbitrary inverse operator mapping seismic waveforms $\mathcal{P}$ to velocity maps $\mathcal{V}$ that is unique. Let $E_p : \mathcal{P} \to \tilde{\mathcal{P}}$ and $D_p : \tilde{\mathcal{P}} \to \mathcal{P}$ denote the encoder and decoder for the seismic waveform space $\mathcal{P}$, respectively. Similarly, let $E_v : \mathcal{V} \to \tilde{\mathcal{V}}$ and $D_v : \tilde{\mathcal{V}} \to \mathcal{V}$ denote the encoder and decoder for the velocity space $\mathcal{V}$. Here, $\tilde{p} \in \tilde{\mathcal{P}}$ and $\tilde{v} \in \tilde{\mathcal{V}}$ represent the latent space encodings. If we assume that the auto-encoder for the seismic waveform space is optimal, i.e., $D_p \circ E_p(p) = \hat{p} \approx p$, then there exists a functional mapping in the latent space $L_{\tilde{p} \to \tilde{v}} : \tilde{\mathcal{P}} \to \tilde{\mathcal{V}}$.*

*Proof.* Given the inverse operator $f^{-1} : \mathcal{P} \to \mathcal{V}$, by definition, for any $p \in \mathcal{P}$, there exists $v \in \mathcal{V}$ such that $v = f^{-1}(p)$.

Let the latent space representations $\tilde{p} \in \tilde{\mathcal{P}}$ and $\tilde{v} \in \tilde{\mathcal{V}}$ be defined by the auto-encoders as follows:

$$\tilde{p} = E_p(p), \quad \hat{p} = D_p(\tilde{p}) \tag{14}$$
$$\tilde{v} = E_v(v), \quad \hat{v} = D_v(\tilde{v}) \tag{15}$$

To construct the latent space mapping $\tilde{v} = L_{\tilde{p} \to \tilde{v}}(\tilde{p})$, consider the sequence of compositions involving the encoders, decoders, and the inverse operator $f^{-1}$:

$$
\begin{aligned}
\tilde{v} &= E_v(v) \\
&= E_v(f^{-1}(p)) \quad \text{(since } v = f^{-1}(p)) \\
&= E_v(f^{-1}(\hat{p})) \quad \text{(assuming reconstruction: } \hat{p} \approx p) \\
&= E_v(f^{-1}(D_p(\tilde{p}))) \quad \text{(since } \hat{p} = D_p(\tilde{p}))
\end{aligned}
\tag{16}
$$

Thus, by definition of the composition of functions, the latent space mapping can be expressed as:

$$
L_{\tilde{p} \to \tilde{v}} = E_v \circ f^{-1} \circ D_p
\tag{17}
$$

$\square$

| Dataset | Examples | Velocity shape | Waveform shape |
|---|---|---|---|
| FlatVel-A | 30,000 | (1, 70, 70) | (5, 1000, 70) |
| FlatVel-B | 30,000 | (1, 70, 70) | (5, 1000, 70) |
| CurveVel-A | 30,000 | (1, 70, 70) | (5, 1000, 70) |
| CurveVel-B | 30,000 | (1, 70, 70) | (5, 1000, 70) |
| FlatFault-A | 60,000 | (1, 70, 70) | (5, 1000, 70) |
| FlatFault-B | 60,000 | (1, 70, 70) | (5, 1000, 70) |
| CurveFault-A | 60,000 | (1, 70, 70) | (5, 1000, 70) |
| CurveFault-B | 60,000 | (1, 70, 70) | (5, 1000, 70) |
| Style-A | 67,000 | (1, 70, 70) | (5, 1000, 70) |
| Style-B | 67,000 | (1, 70, 70) | (5, 1000, 70) |

Table 2: Statistics on the number of samples, the size of the velocity and waveforms for each dataset in OpenFWI Deng et al. (2022).

## B  DATASET DESCRIPTION

The OpenFWI comprises multi-structural benchmark datasets of significant size that can be used for solving full waveform inversion (FWI) using machine learning techniques (Deng et al., 2022). In particular, the repository contains 3 major groups of data: (1) Vel Family, (2) Fault Family, and (3) Style Family. These groups represent simple to complex sub-surface geological settings with seismic velocity and waveforms information. The Vel family is the simplest geological patterns including four datasets - (1) FlatVel-A (FVA), (2) FlatVel-B (FVB), (3) CurveVel-A (CVA), and (4) CurveVel-B (CVB). The difference between FlatVel and CurveVel is that the former represents low-energy geological environments where the rock layers are deposited horizontally and the latter consists of curved layers which are formed due to structural deformation of flat layers. The Fault family also has four datasets - (1) FlatFault-A (FFA), (2) FlatFault-B (FFB), (3) CurveFault-A (CFA), and (4) CurveFault-B (CFB). Unlike Vel datasets, the Fault family contains fault-like deformations, which is fracturing of rocks under certain pressure conditions. Due to the presence of faults, the Fault family becomes more complicated and challenging to model. The Style family has two datasets - (1) Style-A (STA), and (2) Style-B (STB). This dataset is generated using the style transfer method where the COCO dataset (Lin et al., 2014) is set as the content images and the Marmousi dataset is set as the style image. This is the most complex OpenFWI dataset as it represents highly complex geological settings where the velocity is changing rapidly and abruptly. In summary, the details about the Vel and Fault datasets are described in table 2.

The waveform data is represented as (# source $\times$ recording time $\times$ receiver length) whereas the velocity follow (1 $\times$ depth $\times$ receiver length) shape. In seismic surveys, the wave arrival time is recorded at the surface and thus, the waveform data for a single source is represented as a function of time and receiver length. In total, there are 5 seismic sources uniformly spaced along the surface, and

wavefields are recorded by 70 receivers (uniformly spaced) along the surface for 1000 milliseconds. Therefore, the seismic wavefields are of the shape $(5 \times 1000 \times 70)$. On the other hand, velocity maps are represented as functions of spatial dimensions, depth and horizontal coverage, and thus have the shape $(1 \times 70 \times 70)$.

## C  PRIOR WORKS IN DL4SI AS SPECIAL CASES OF GFI

Figure 9 provides additional schematic illustrations of prior works in DL4SI such as InversionNet, WaveformNet, and AutoLinear as special cases of our proposed GFI framework.

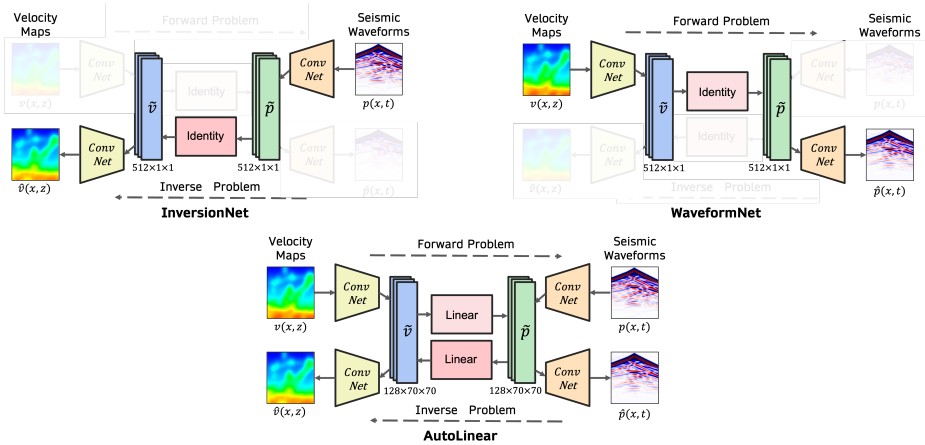

Figure 9: Prior Works exprerssed as GFI

## D  ADDITIONAL EXPERIMENTAL DETAILS

### D.1  CYCLE LOSS FOR INVERTIBLE X-NET

Given the velocity maps $v$ and the waveforms $p$, the predictions for the waveform and velocity can be obtained using the Invertible X-Net as follows:

$$\hat{p} = f_{v \to p}(v) = \mathtt{D}_p \circ L_{\tilde{v} \to \tilde{p}} \circ \mathtt{E}_v(v) \tag{18}$$
$$\hat{v} = f_{p \to v}(p) = \mathtt{D}_v \circ L_{\tilde{p} \to \tilde{v}} \circ \mathtt{E}_p(p) \tag{19}$$

Now, the Invertible X-Net architecture can be further applied on $\hat{p}$ and $\hat{v}$ to create the following transformations:

$$\hat{\hat{p}} = f_{v \to p}(\hat{v}) = \mathtt{D}_p \circ L_{\tilde{v} \to \tilde{p}} \circ \mathtt{E}_v(\hat{v}) \tag{20}$$

$$\hat{\hat{v}} = f_{p \to v}(\hat{p}) = \mathtt{D}_v \circ L_{\tilde{p} \to \tilde{v}} \circ \mathtt{E}_p(\hat{p}) \tag{21}$$

The cycle-loss for Invertible X-Net can be mathematically defined as follows:

$$\mathcal{L}_{\text{cycle}} = \mathcal{L}(p, \hat{\hat{p}}) + \mathcal{L}(v, \hat{\hat{v}}) \tag{22}$$

where $\mathcal{L}$ is the loss function that can be MSE, MAE or Elastic Loss. Note that the formulation of Cycle-Loss does not rely on paired examples, and can be applied on un-paired data as well. The combined loss function including cycle-loss can be given as:

$$\mathcal{L}_{\text{X-Net (Cycle)}} = \mathcal{L}_{\text{forward}} + \mathcal{L}_{\text{inverse}} + \mathcal{L}_{\text{cycle}} \tag{23}$$

For training Invertible X-Net (Cycle) model, we use loss function shown in Equation 23.

## D.2   ADDITIONAL TRAINING DETAILS

For training, we normalize the velocity using min-max normalization and seismic waveform using standard normalization to rescale the data to mean 0 and standard deviation as 1. Table 5 shows other hyperparameter details for training Latent U-Net and Invertible X-Net models on OpenFWI datasets.

Since the baseline models have different normalization schemes than our models, we compare model predictions during evaluation by unnormalizing the predictions to original domains. For example, AutoLinear uses min-max normalization for velocity and a combination of log-normalization $x_{\text{norm}} = (\log_e(1 + |x|) * \texttt{sign}(x)$ followed by min-max in the log-normalized domain for waveform. We unnormalized both the velocity and waveform predictions so that we can measure and visualize errors in the predictions in the original space, allowing easy comparison across models.

## D.3   MODEL ARCHITECTURE

We summarize details related to model architecture, layers, and number of parameters related to seismic and velocity encoder-decoder architecture in Table 3, and regarding the latent models used for translation in Table 4. Further, we compare our model parameters with baseline models in Table 6.

### D.3.1   LATENT U-NET

Latent U-Net architectures are specifically adapted to subsurface imaging by employing U-Net-based encoder-decoder pairs designed to handle domain-specific input and output dimensions, such as velocity maps and seismic waveforms, while enabling latent dimension processing. Both the Large and Small variants utilize identical encoder and decoder designs, featuring 5 layers in the encoder and 6 layers in the decoder. These layers operate on an embedding dimension of 128x70x70, with channel sizes ranging from 8 to 128 in the encoder and 128 to 1 (velocity) or 128 to 5 (waveform) in the decoder.

The Large Latent U-Net incorporates 2 depth levels and 4 convolutional blocks per depth level in its latent space translation model, resulting in a total parameter size of 34.96M. The Small variant, designed for reduced complexity, includes 1 convolutional block per depth level in its latent translation model, reducing the total parameter size to 18.13M.

### D.3.2   INVERTIBLE X-NET

Invertible X-Net leverages an iUNet-based architecture to achieve bidirectional mappings, enabling consistent forward (velocity-to-waveform) and inverse (waveform-to-velocity) transformations within a shared framework. iUNet introduces invertible coupling blocks, ensuring bijectivity for the latent space translation model. Max pooling is replaced with orthogonal convolutional filters for downsampling, while upsampling is performed using orthogonal deconvolution filters to maintain invertibility.

While the latent space translation model in Invertible X-Net is fully invertible due to its iUNet structure, the encoder-decoder pairs are not invertible. Therefore, Invertible X-Net as a whole is not fully invertible. However, by employing two separate encoder-decoder pairs for velocity and waveform domains, Invertible X-Net achieves an architecture that enables bidirectional mappings between the two domains while preserving consistency.

Invertible X-Net features 4 depth levels, with 4 invertible coupling blocks per level, and adopts an encoder-decoder structure similar to Latent U-Net, with 5 encoder layers and 6 decoder layers, an embedding dimension of 128x70x70, and channel sizes ranging from 8 to 128. The latent translation model - iUNet consists of 25.78M parameters, contributing to a total model size of 26.06M.

Table 3: Architecture Details of Seismic Waveform and Velocity Encoder-Decoder models.

| Model | #Layers | #Embedding Dim | Channels | #Params |
|---|---|---|---|---|
| Velocity Encoder | 5 | 128x70x70 | [8, 16, 32, 64, 128] | 11632 |
| Velocity Decoder | 6 | 128x70x70 | [128, 64, 32, 16, 1, 1] | 27877 |
| Seismic Waveform Encoder | 5 | 128x70x70 | [8, 16, 32, 64, 128] | 55680 |
| Seismic Waveform Decoder | 6 | 128x70x70 | [128, 64, 32, 16, 5, 5] | 186497 |

Table 4: Architecture details of Latent U-Net and IU-Net latent space translation models

| Model | #Depths | #Conv blocks/ Coupling Blocks | #Params |
|---|---|---|---|
| Latent U-Net | 2 | 4 | 34.68M |
| IU-Net | 4 | 4 | 25.78M |

Table 5: Hyperparameter details for training Latent U-Net and Invertible X-Net models.

| Model | #Epochs | Optimizer | LR | LR Scheduler |
|---|---|---|---|---|
| Latent U-Net | 450 | Adam | 2e-3 | StepLR |
| Invertible X-Net | 450 | Adam | 2e-3 | StepLR |

Table 6: Comparison of encoder, decoder, and latent model parameters for our model (Latent U-Net and Invertible X-Net) with other baseline models. The parameters for Latent U-Nets and Invertible X-Nets are calculated for Latent dimension 70.

| Model | #Vel Encoder Params | #Vel Decoder Params | #Amp Encoder Params | #Amp Decoder Params | #Translation Params | #Total Params |
|---|---|---|---|---|---|---|
| FNO | - | - | - | - | - | 7.38M |
| InversionNet | - | 9.34M | 35.76M | - | Identity | 24.41M |
| VelocityGAN | - | 9.34M | 35.76M | - | Identity | 24.41M |
| Autolinear | 12.98M | 9.98M | 2.29M | 10.18M | 16.5K | 35.45M |
| Latent U-Net(Small) | 11.6K | 27.88K | 55.68K | 186.5K | 17.86M | 18.13M |
| Latent U-Net(Large) | 11.6K | 27.88K | 55.68K | 186.5K | 34.68M | 34.96M |
| Invertible X-Net | 11.6K | 27.88K | 55.68K | 186.5K | 25.78M | 26.06M |

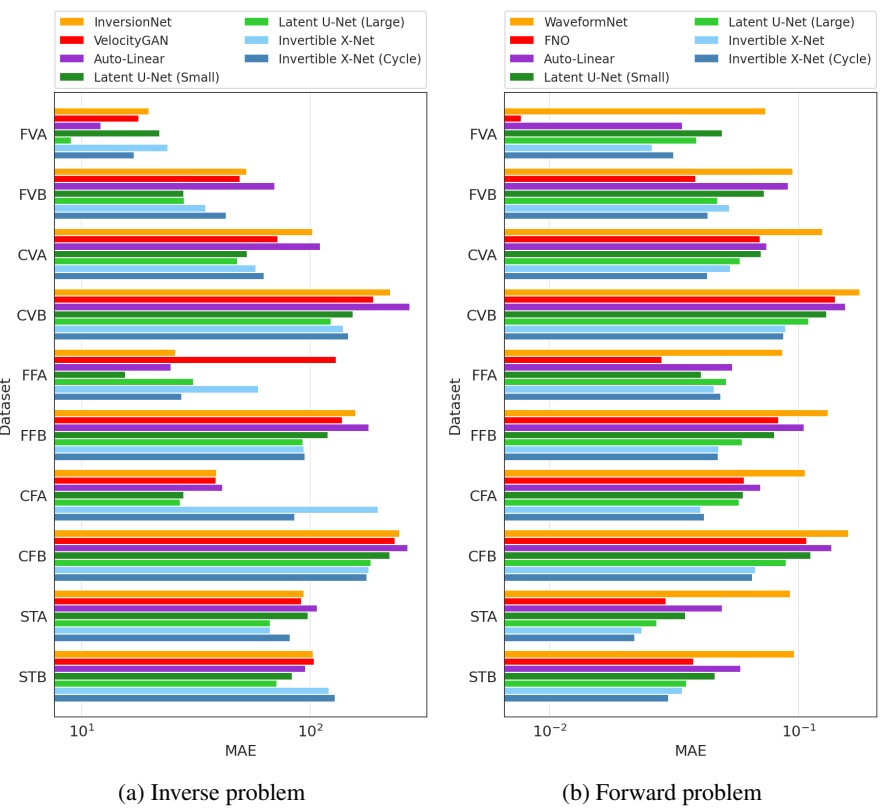

(a) Inverse problem

(b) Forward problem

Figure 10: Comparison of Latent U-Nets (Small and Large), Invertible X-Net, Invertible X-Net (Cycle) with different baseline methods across different OpenFWI datasets.

# E ADDITIONAL RESULTS

## E.1 ADDITIONAL INVERSE MODELING RESULTS

We provide more detailed comparison of our models with other baseline models in Table 7. Our proposed models consistently outperform baseline models on multiple datasets indicating superior generalizability on in-distributions examples.

Additionally, we also show zero shot generalization of our models on the Marmousi and Overthrust dataset in Tables 8. Our model Invertible X-Net shows generalizability in SSIM indicating that overall prediction has better geological understanding than other baseline models.

Table 7: Comparison of our models (Latent U-Net (Large) and Invertible X-Net) with other baseline models for the inverse problem across 10 OpenFWI datasets. The bold highlights the best performing model on that dataset.

| Metric | Model | FVA | FVB | CVA | CVB | FFA | FFB | CFA | CFB | STA | STB | Average Rank |
|---|---|---|---|---|---|---|---|---|---|---|---|---|
| MAE↓ | InversionNet | 19.67 | 52.77 | 102.77 | 224.57 | 25.80 | 158.31 | 38.90 | 246.94 | 93.96 | 103.37 | 2.8 |
| | Auto-Linear | 12.16 | 70.05 | 110.72 | 273.02 | **24.55** | 181.39 | 41.38 | 268.47 | 107.85 | 95.63 | 3.5 |
| | Latent U-Net | **9.01** | **28.11** | **48.05** | **123.56** | 30.91 | **92.91** | **27.05** | 185.22 | **67.00** | **71.41** | **1.3** |
| | Invertible X-Net | 23.80 | 34.94 | 57.86 | 139.81 | 59.45 | 94.11 | 198.56 | **181.03** | 67.09 | 121.42 | 2.7 |
| MSE↓ | InversionNet | 1002.74 | 17271.62 | 36438.33 | 188044.46 | 4081.10 | 68214.92 | 9490.73 | 136327.84 | 23626.76 | 58622.35 | 2.8 |
| | Auto-Linear | 1053.03 | 33907.57 | 42391.39 | 236457.48 | 5952.25 | 81789.80 | 13715.42 | 154713.12 | 31274.60 | 21819.14 | 3.5 |
| | Latent U-Net | **216.92** | **6464.06** | **12609.65** | **91784.50** | **2348.22** | 33935.12 | **3771.38** | 93249.92 | 14353.55 | **14564.94** | **1.3** |
| | Invertible X-Net | 1245.52 | 6969.42 | 14659.46 | 96121.90 | 7719.55 | **32559.95** | 77105.41 | **86512.36** | **13106.96** | 31550.22 | 2.4 |
| SSIM↑ | InversionNet | 0.9894 | 0.9461 | 0.8073 | 0.6726 | 0.9765 | 0.7208 | 0.9566 | 0.6136 | 0.8858 | 0.6314 | 3 |
| | Auto-Linear | 0.9887 | 0.9044 | 0.8056 | 0.6169 | 0.97 | 0.6865 | 0.9424 | 0.5695 | 0.8422 | 0.7274 | 3.8 |
| | Latent U-Net | **0.9967** | **0.9809** | **0.9273** | **0.8156** | **0.991** | 0.8515 | **0.98** | 0.6930 | 0.9298 | **0.8064** | **1.3** |
| | Invertible X-Net | 0.9917 | 0.9769 | 0.9135 | 0.8076 | 0.9826 | **0.8532** | 0.9316 | **0.7116** | **0.9360** | 0.7913 | 1.9 |

Table 8: Comparison of our models (Latent U-Net (Large) and Invertible X-Net) on real world like datasets - Marmousi, Overthrust, Marmousi Smooth, Overthrust Smooth for the Inverse Problem. The bold highlights the best performing model on that dataset.

| Metric | Model | Marmousi | Overthrust | Marmousi Smooth | Overthrust Smooth |
|---|---|---|---|---|---|
| MAE ↓ | InversionNet | **282.39** | 273.80 | **179.34** | **148.34** |
| | Auto-Linear | 285.38 | 298.89 | 206.38 | 207.41 |
| | Latnet U-Net | 322.13 | **264.40** | 242.71 | 198.53 |
| | Invertible X-Net | 298.39 | 308.50 | 245.86 | 258.26 |
| MSE ↓ | InversionNet | **160084.5** | 135988.73 | **67978.40** | **37804.58** |
| | Auto-Linear | 159517.41 | 157434.56 | 78207.07 | 71128.92 |
| | Latnet U-Net | 217508.17 | **122398.64** | 112685.39 | 69122.58 |
| | Invertible X-Net | 180250.47 | 179093.10 | 107344.46 | 124700.75 |
| SSIM ↑ | InversionNet | 0.46 | 0.4519 | 0.6044 | **0.7217** |
| | Auto-Linear | 0.438 | 0.4097 | 0.5423 | 0.6447 |
| | Latnet U-Net | 0.438 | 0.4807 | 0.6371 | 0.7031 |
| | Invertible X-Net | **0.504** | **0.4827** | **0.6633** | 0.6952 |

## E.2 ADDITIONAL FORWARD MODELING RESULTS

Similar to the inverse problem, we provide detailed comparison of our models with other baseline models in Table 9 and 10. Our proposed models consistently outperform baseline models on multiple datasets indicating superior in-distributions generalizability.

Table 9: Comparison of our models (Latent U-Net (Large) and Invertible X-Net) with other baseline models for the forward problem across 10 OpenFWI datasets. The bold highlights the best performing model on that dataset.

| Metric | Model | FVA | FVB | CVA | CVB | FFA | FFB | CFA | CFB | STA | STB | Average Rank |
|---|---|---|---|---|---|---|---|---|---|---|---|---|
| MAE ↓ | FNO | **0.0077** | **0.0385** | 0.0700 | 0.1405 | **0.0282** | 0.0829 | 0.0605 | 0.1075 | 0.0292 | 0.0379 | 2.4 |
| | Auto-Linear | 0.0340 | 0.0906 | 0.0744 | 0.1537 | 0.0541 | 0.1048 | 0.0703 | 0.1356 | 0.0492 | 0.0584 | 3.9 |
| | Latent U-Net | 0.0389 | 0.0473 | 0.0581 | 0.1098 | 0.0512 | 0.059 | 0.0576 | 0.0891 | 0.0269 | 0.0354 | 2.3 |
| | Invertible X-Net | 0.0257 | 0.0527 | **0.0532** | **0.0887** | 0.0457 | **0.0477** | **0.0404** | **0.0671** | **0.0235** | **0.0341** | **1.4** |
| MSE ↓ | FNO | **0.0004** | **0.0066** | 0.0259 | 0.0825 | 0.0059 | 0.0358 | 0.0251 | 0.0471 | 0.0041 | **0.0053** | 1.9 |
| | Auto-Linear | 0.0084 | 0.0427 | 0.0303 | 0.1045 | 0.0217 | 0.0625 | 0.033 | 0.0866 | 0.0174 | 0.0173 | 3.7 |
| | Latent U-Net | 0.0173 | 0.0127 | **0.0160** | 0.0489 | 0.0344 | 0.0175 | 0.0442 | 0.0297 | 0.0053 | 0.0082 | 2.6 |
| | Invertible X-Net | 0.0062 | 0.0176 | 0.0175 | **0.0305** | **0.0185** | **0.0117** | **0.0126** | **0.0163** | **0.004** | 0.0087 | **1.8** |
| SSIM ↑ | FNO | **0.9967** | 0.9781 | 0.8881 | 0.8354 | 0.9667 | 0.8702 | 0.9166 | 0.8160 | 0.9655 | 0.9417 | 3.1 |
| | Auto-Linear | 0.9694 | 0.9289 | 0.9038 | 0.8567 | 0.9470 | 0.8736 | 0.9253 | 0.8188 | 0.9569 | 0.9300 | 3.5 |
| | Latent U-Net | 0.9764 | 0.9779 | 0.9237 | 0.89 | 0.9659 | 0.9283 | 0.9571 | 0.8636 | 0.9819 | 0.9702 | 2.3 |
| | Invertible X-Net | 0.9887 | **0.9782** | **0.9559** | **0.9221** | **0.9744** | **0.954** | **0.9757** | **0.9156** | **0.989** | **0.9805** | **1.1** |

## E.3 IMPORTANCE OF COMBINED LOSS FUNCTION FOR INVERTIBLE-XNET

In this section, we focus on the training of the Invertible X-Net model using a combined loss function (incorporating both the forward and inverse problems), as opposed to training it solely with a forward loss function. Figure 11 shows that when model is trained using only forward loss, then its performance falls short compared to the Latent U-Net (Large) model. This discrepancy can be attributed to the fact that the Latent U-Net has a higher model complexity than Invertible X-Net, despite their similar sizes. Nonetheless, when the Invertible X-Net model is trained with combined loss (forward and inverse), the model is able to outperform Latent U-Net model with a good mar-

Table 10: Comparison of our models (Latent U-Net (Large) and Invertible X-Net) on real world like datasets - Marmousi, Overthrust, Marmousi Smooth, Overthrust Smooth for the Forward Problem. The bold highlights the best performing model on that dataset.

| Metric | Model | Marmousi | Overthrust | Marmousi Smooth | Overthrust Smooth |
|---|---|---|---|---|---|
| MAE ↓ | FNO | 0.1484 | 0.2726 | 0.1077 | 0.1892 |
| | Auto-Linear | 0.2818 | 0.3018 | 0.2821 | 0.2730 |
| | Latnet U-Net | **0.1338** | 0.2502 | **0.1013** | **0.1875** |
| | Invertible X-Net | 0.1425 | **0.2311** | 0.1056 | 0.2062 |
| MSE ↓ | FNO | **0.1110** | 0.580 | **0.0811** | **0.3495** |
| | Auto-Linear | 0.4227 | 0.6203 | 0.5325 | 0.5593 |
| | Latnet U-Net | 0.1116 | 0.5494 | 0.0927 | 0.4133 |
| | Invertible X-Net | 0.1168 | **0.4026** | 0.08130 | 0.5049 |
| SSIM ↑ | FNO | 0.8148 | 0.7404 | 0.9021 | 0.8447 |
| | Auto-Linear | 0.6344 | 0.672 | 0.6670 | 0.7192 |
| | Latnet U-Net | **0.8343** | 0.7700 | **0.9143** | **0.8626** |
| | Invertible X-Net | 0.827 | **0.7863** | 0.9112 | 0.8500 |

gin. This highlights the value of joint training, demonstrating that simultaneously learning both the forward and inverse problems can lead to better results than learning the two models separately.

In Figures 12 and 13, we illustrate asymmetry in learning the translation for the forward and inverse problems using CVB and CFA datasets. From the figures, we observe that the model learns the inverse mapping in the initial epochs and gradually starts to learn the solution to the forward problem in later epochs. Since the network optimizes the combined loss function on both velocity and waveform together, the gradients from the combined loss help the model to achieve better forward solution. This corroborates with our hypothesis that the model trained on combined loss is able to learn the connection between forward and inverse problem.

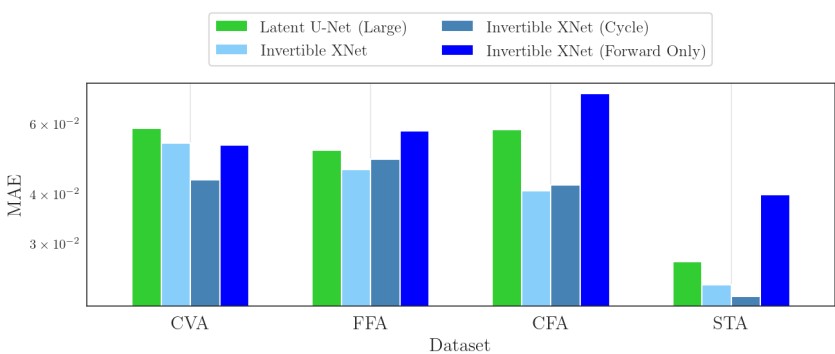

Figure 11: Comparison of Latent U-Net (Large), Invertible X-Net, Invertible X-Net (Forward Only), and Invertible X-Net (Cycle) on the forward problem across various OpenFWI datasets.

### E.4 ZERO-SHOT PERFORMANCE

In this part, we provide detailed insights into zero shot performance of our models (Latent U-Net (Large) and Invertible X-Net) with other baselines across all datasets. This investigation helps us understand overall out-of-distribution generalization of a model and underscore the importance of learning the translation problem in the latent space.

### E.5 INVERSE PROBLEM

Figure 14 and 15 show generalization performance of Invertible X-Net and Latent U-Net (Large) models respectively on the inverse problem using MAE, MSE, and SSIM metrics. As described in the main paper, we show models trained across all dataset as rows and evaluated across all test datasets as columns. For comparison, we evaluate metrics such as MAE, MSE, and SSIM and calculate its difference between our models and other baseline models. For MAE and MSE, when the color intensity is blue, our model show better generalizability and vice-versa whereas, for SSIM, when the color intensity is red indicates better generalization of our model and vice-versa.

From Figure 14, we observe that the Invertible X-Net shows superior generalization over the baseline models (AutoLinear, InversionNet, and VelocityGAN) across all metrics except on the FVB dataset. Figure 14 (d) compare Invertible X-Net with Latent U-Net (Large) model where we see that Invertible X-Net shows better generalization on complex datasets such as CFB, STA, and STB datasets, while Latent U-Net is better on relatively simpler datasets such as CVA, CFA, and more. Similarly, Figure 15 shows the comparison of Latent U-Net (Large) model with other baseline models. As expected, we observe that our model is able to generalize much better than all the baseline models.

### E.6 FORWARD PROBLEM

Similar to the inverse problem, we analyze out-of-distribution generalization of our models against baseline models across all evaluation metrics.

Figures 16 and 17 compares the performance of our models Invertible X-Net and Latent U-Net (Large) against baselines. In Figure 16, we observe strong generalization of Invertible X-Net over all baselines - AutoLinear, FNO, and WaveformNet. Figure 16 (d) shows the comparative performance of Invertible X-Net against Latent U-Net (Large) where we see that Invertible X-Net dominates overall across all metrics. In Figure 17, we compare the Latent U-Net (Large) model against same baselines as above. The figure indicates Latent U-Net (Large) model has much stronger generalizability than baseline models consistently.

## F  ABLATION STUDIES

### F.1 EFFECT OF VARYING LATENT SPACE SIZES

Figure 18a (a) shows how the performance of Latent U-Net model is affected with change in the latent space size. As the latent space size is reduced, the MAE and MSE metrics for mapping seismic waveform to velocity is increasing, while the SSIM is decreasing. The impact is more pronounced on complex datasets such as B group of datasets. These datasets represent geologically complex datasets and therefore may require larger latent space to encode geological heterogeneity. In Figure 18b, we compare the large Latent U-Net with the small Latent U-Net model, while latent space size is also reduced. We observe that the performance gap between the two models reduces as the latent space size is also reduced. This underscore the importance of latent space size for encoding the geological features for an effective translation.

Further, we provide visualizations of the first two primary PCA and t-SNE components of the velocity latent space in Figure 19 and 20 respectively. We take the encoder trained on a dataset and get the latent space encoding on several datasets. This visualization shows the in-distribution and out-of-distribution generality of our encoders and highlights the importance of the manifold for latent space translation. Overall, we observe that the larger latent space $70 \times 70$ have better structure than $8 \times 8$, indicating that the ideal size of latent space should be decided based on the complexity of dataset and problem being solved.

### F.2 EFFECT OF SKIP CONNECTIONS FOR LATENT U-NET

In this section, we study the impact of skip connections of Latent U-Net model for the inverse problem. Figure 21 shows how the MAE and MSE are increasing and SSIM is decreasing when the latent space size is decreasing from $70 \times 70$ to $8 \times 8$. We observe that the impact of skip connections is more pronounced at smaller latent space size as opposed to larger space.

### F.3   EFFECT OF MANIFOLD LEARNING

Figure 22 for Latent U-Net Small further elaborates on our finding that direct translation learning consistently outperforms the two-stage approach, where translation follows reconstruction.

## G   NOISE EXPERIMENTS

Table 11: Quantitative results on CurveFault-B with Gaussian noise of varying variance $\sigma^2$ added during testing for the inverse problem.

| Model | $\sigma^2 = 0$ | | | $\sigma^2 = $ 1e-5 PSNR=84.07dB | | | $\sigma^2 = $ 5e-5 PSNR=77..08dB | | | $\sigma^2 = $ 1e-4 PSNR=74.07dB | | | $\sigma^2 = $ 5e-4 PSNR=67.08dB | | |
|---|---|---|---|---|---|---|---|---|---|---|---|---|---|---|---|
| | MAE | MSE | SSIM | MAE | MSE | SSIM | MAE | MSE | SSIM | MAE | MSE | SSIM | MAE | MSE | SSIM |
| Latnet U-Net | 185.22 | 93248.64 | 0.6930 | 185.54 | 93538.78 | 0.6926 | 187.03 | 94674.546 | 0.6917 | 188.56 | 95777.95 | 0.6906 | 195.94 | 101444.7 | 0.6844 |
| Degradation (%) | (-) | (-) | (-) | 0.17% | 0.31% | 0.05% | 0.97% | 1.52% | 0.20% | 1.80% | 2.71% | 0.35% | 5.78% | 8.78% | 1.24% |
| Invertible X-Net | 180.97 | 86465.25 | 0.7117 | 181.32 | 86673.38 | 0.7115 | 182.83 | 87674.14 | 0.7106 | 184.17 | 88585.86 | 0.7097 | 192.22 | 94728.94 | 0.7036 |
| Degradation (%) | (-) | (-) | (-) | 0.19% | 0.24% | 0.02% | 1.02% | 1.39% | 0.15% | 1.76% | 2.45% | 0.27% | 6.21% | 9.55% | 1.13% |
| Auto-Linear | 268.47 | 154713.10 | 0.5695 | 270.29 | 156201.81 | 0.5682 | 277.05 | 162156.50 | 0.5632 | 285.34 | 169892.26 | 0.5572 | 325.02 | 211616.29 | 0.5190 |
| Degradation (%) | (-) | (-) | (-) | 0.67% | 0.96% | 0.21% | 3.19% | 4.81% | 1.10% | 6.28% | 9.81% | 2.15% | 21.06% | 36.77% | 8.86% |

Table 12: Quantitative results on FlatFault-B with Gaussian noise of varying variance $\sigma^2$ added during testing for the inverse problem.

| Model | $\sigma^2 = 0$ | | | $\sigma^2 = $ 1e-5 PSNR=84.07dB | | | $\sigma^2 = $ 5e-5 PSNR=77..08dB | | | $\sigma^2 = $ 1e-4 PSNR=74.07dB | | | $\sigma^2 = $ 5e-4 PSNR=67.08dB | | |
|---|---|---|---|---|---|---|---|---|---|---|---|---|---|---|---|
| | MAE | MSE | SSIM | MAE | MSE | SSIM | MAE | MSE | SSIM | MAE | MSE | SSIM | MAE | MSE | SSIM |
| Latnet U-Net | 92.93 | 33925.46 | 0.8515 | 93.47 | 34137.49 | 0.8511 | 95.43 | 34907.52 | 0.8499 | 98.19 | 36002.58 | 0.8482 | 108.83 | 40770.50 | 0.8417 |
| Degradation (%) | (-) | (-) | (-) | 0.58% | 0.62% | 0.05% | 2.68% | 2.89% | 0.18% | 5.66% | 6.12% | 0.38% | 17.10% | 20.17% | 1.15% |
| Invertible X-Net | 94.05 | 32548.49 | 0.8532 | 96.23 | 33283.08 | 0.8529 | 104.75 | 36522.60 | 0.85 | 110.30 | 39016.04 | 0.8474 | 122.66 | 45509.88 | 0.8382 |
| Degradation (%) | (-) | (-) | (-) | 2.31% | 2.25% | 0.03% | 11.37% | 12.20% | 0.37% | 17.27% | 19.87% | 0.67% | 30.41% | 39.82% | 1.76% |
| Auto-Linear | 181.39 | 81789.80 | 0.6865 | 187.40 | 85327.79 | 0.6821 | 206.58 | 98616.76 | 0.6664 | 221.79 | 110396.50 | 0.6511 | 283.50 | 167482.73 | 0.5796 |
| Degradation (%) | (-) | (-) | (-) | 3.31% | 4.32% | 0.64% | 13.88% | 20.57% | 2.93% | 22.26% | 34.97% | 5.16% | 56.28% | 104.77% | 15.57% |

Table 13: Quantitative results on CurveFault-B with Gaussian noise of varying variance $\sigma^2$ added during testing for the forward problem.

| Model | $\sigma^2 = 0$ | | | $\sigma^2 = $ 1e-5 PSNR=56.02dB | | | $\sigma^2 = $ 5e-5 PSNR=49.03dB | | | $\sigma^2 = $ 1e-4 PSNR=46.02dB | | | $\sigma^2 = $ 5e-4 PSNR=39.03dB | | |
|---|---|---|---|---|---|---|---|---|---|---|---|---|---|---|---|
| | MAE | MSE | SSIM | MAE | MSE | SSIM | MAE | MSE | SSIM | MAE | MSE | SSIM | MAE | MSE | SSIM |
| Latnet U-Net | 0.0891 | 0.0297 | 0.8636 | 0.0891 | 0.0297 | 0.8635 | 0.0892 | 0.0297 | 0.8634 | 0.0897 | 0.0301 | 0.8630 | 0.0966 | 0.0396 | 0.8565 |
| Degradation (%) | (-) | (-) | (-) | 0% | 0% | 0.01% | 0.13% | 0.30% | 0.01% | 0.60% | 1.40% | 0.06% | 8.37% | 33.47% | 0.82% |
| Invertible X-Net | 0.0671 | 0.0163 | 0.9157 | 0.0672 | 0.0164 | 0.9156 | 0.0674 | 0.0164 | 0.9152 | 0.0678 | 0.0166 | 0.9147 | 0.0719 | 0.0193 | 0.9090 |
| Degradation (%) | (-) | (-) | (-) | 0.07% | 0.07% | 0.009% | 0.42% | 0.54% | 0.04% | 1.00% | 1.68% | 0.10% | 7.08% | 18.18% | 0.73% |
| Auto-Linear | 0.1356 | 0.0866 | 0.8188 | 0.1356 | 0.0868 | 0.8187 | 0.1360 | 0.0873 | 0.8183 | 0.1365 | 0.088 | 0.8179 | 0.139 | 0.0919 | 0.8159 |
| Degradation (%) | (-) | (-) | (-) | 0.06% | 0.14% | 0.01% | 0.34% | 0.78% | 0.05% | 0.70% | 1.61% | 0.09% | 2.53% | 6.03% | 0.35% |

## H   ADDITIONAL VISUALIZATIONS

Here, we provide additional visualization of the waveform and velocity predictions for baselines and our models, namely Latent-UNet (small), Latent-UNet (large), Invertible-XNet, and Invertible-XNet (cycle), for both forward and inverse problems. Please note that we show the prediction of seismic waveform and velocity in the original space by unnormalizing the predictions for every model (Figures 23 - 34).

Table 14: Quantitative results on FlatFault-B with Gaussian noise of varying variance $\sigma^2$ added during testing for the forward problem.

| Model | $\sigma^2 = 0$ | | | $\sigma^2 = 1\text{e-}5$ PSNR=56.02dB | | | $\sigma^2 = 5\text{e-}5$ PSNR=49.03dB | | | $\sigma^2 = 1\text{e-}4$ PSNR=46.02dB | | | $\sigma^2 = 5\text{e-}4$ PSNR=39.03dB | | |
|---|---|---|---|---|---|---|---|---|---|---|---|---|---|---|---|
| | MAE | MSE | SSIM | MAE | MSE | SSIM | MAE | MSE | SSIM | MAE | MSE | SSIM | MAE | MSE | SSIM |
| Latent U-Net | 0.0594 | 0.0175 | 0.9283 | 0.0594 | 0.0175 | 0.9283 | 0.0598 | 0.0179 | 0.9279 | 0.0606 | 0.0187 | 0.9271 | 0.0694 | 0.0325 | 0.9178 |
| Degradation (%) | (-) | (-) | (-) | 0% | 0% | 0% | 0.66% | 2.16% | 0.04% | 1.99% | 6.99% | 0.12% | 16.87% | 85.99% | 1.12% |
| Invertible X-Net | 0.0477 | 0.0117 | 0.9540 | 0.0478 | 0.0117 | 0.9540 | 0.0482 | 0.0119 | 0.9536 | 0.0487 | 0.0122 | 0.9531 | 0.0535 | 0.016 | 0.9463 |
| Degradation (%) | (-) | (-) | (-) | 0.17% | 0.33% | 0% | 0.94% | 1.88% | 0.04% | 1.93% | 4.26% | 0.10% | 12.16% | 36.39% | 0.80% |
| Auto-Linear | 0.1048 | 0.0625 | 0.8736 | 0.1051 | 0.0628 | 0.8734 | 0.1061 | 0.0641 | 0.8726 | 0.1072 | 0.0656 | 0.8717 | 0.112 | 0.0727 | 0.868 |
| Degradation (%) | (-) | (-) | (-) | 0.25% | 0.54% | 0.02% | 1.19% | 2.59% | 0.11% | 2.24% | 5.06% | 0.21% | 6.86% | 16.37% | 0.64% |

# I    CODE APPENDIX

All the code required to train and evaluate the proposed methods, as well as the baselines, has been uploaded to an anonymous GitHub repository: `https://github.com/KGML-lab/Generalized-Forward-Inverse-Framework-for-DL4SI`. The data and corresponding processing code used in this work are sourced from the OpenFWI website: `https://openfwi-lanl.github.io/docs/data.html`.

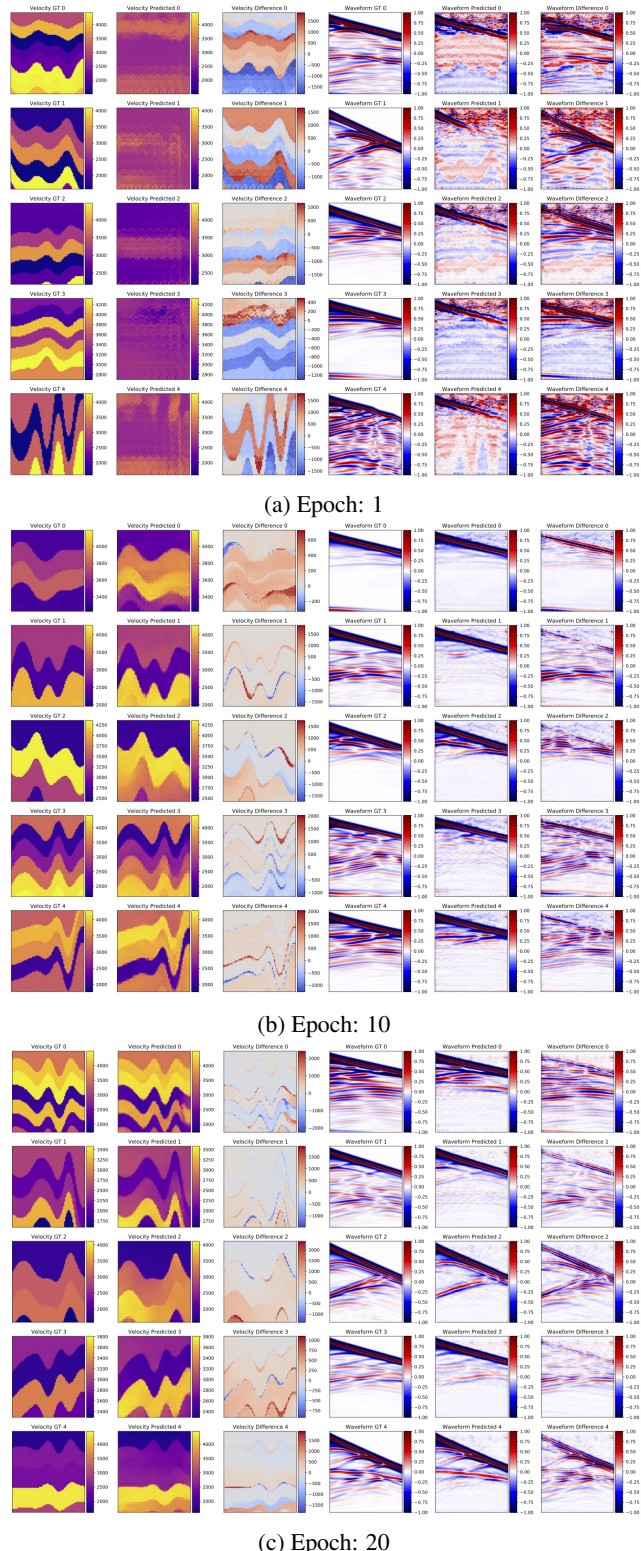

(a) Epoch: 1

(b) Epoch: 10

(c) Epoch: 20

Figure 12: Training of Invertible X-Net model on the CVB dataset illustrating velocity and seismic waveform learning with epochs.

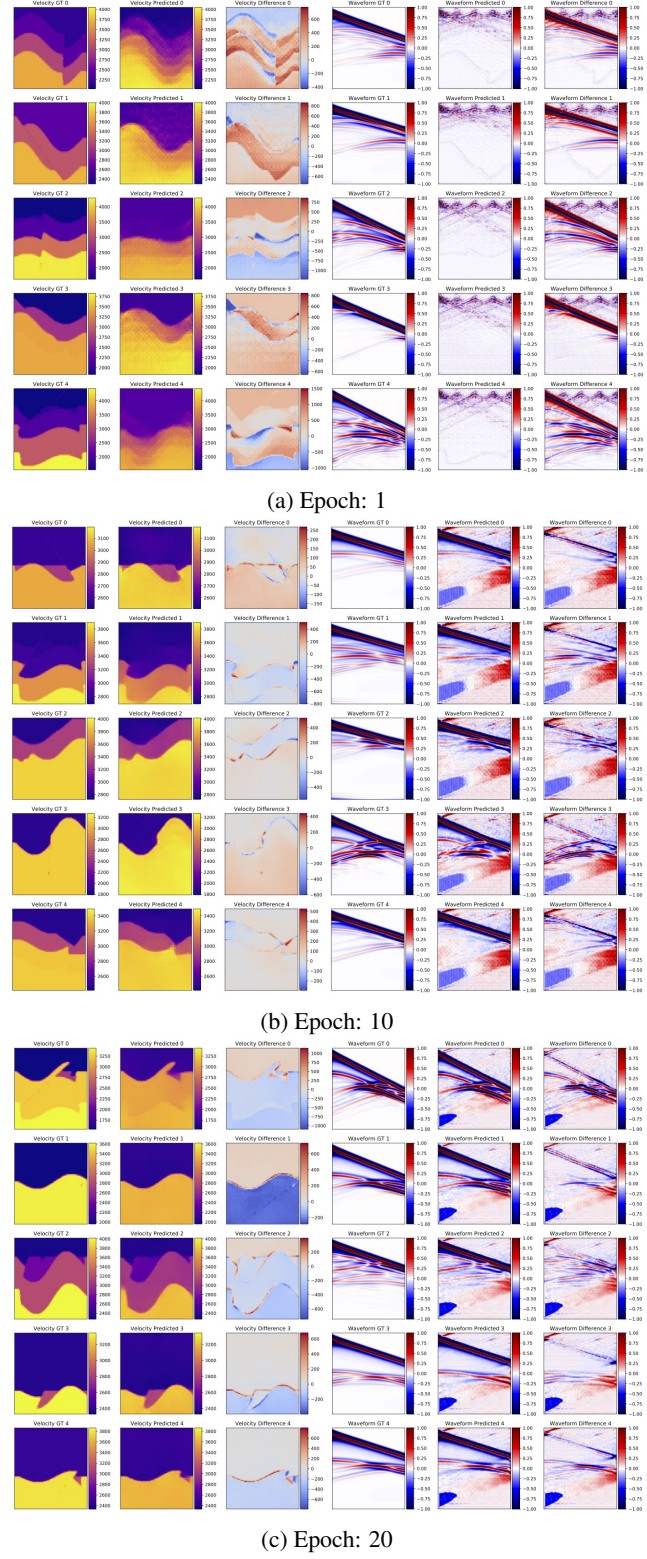

Figure 13: Training of Invertible X-Net model on the CFA dataset illustrating velocity and seismic waveform learning with epochs.

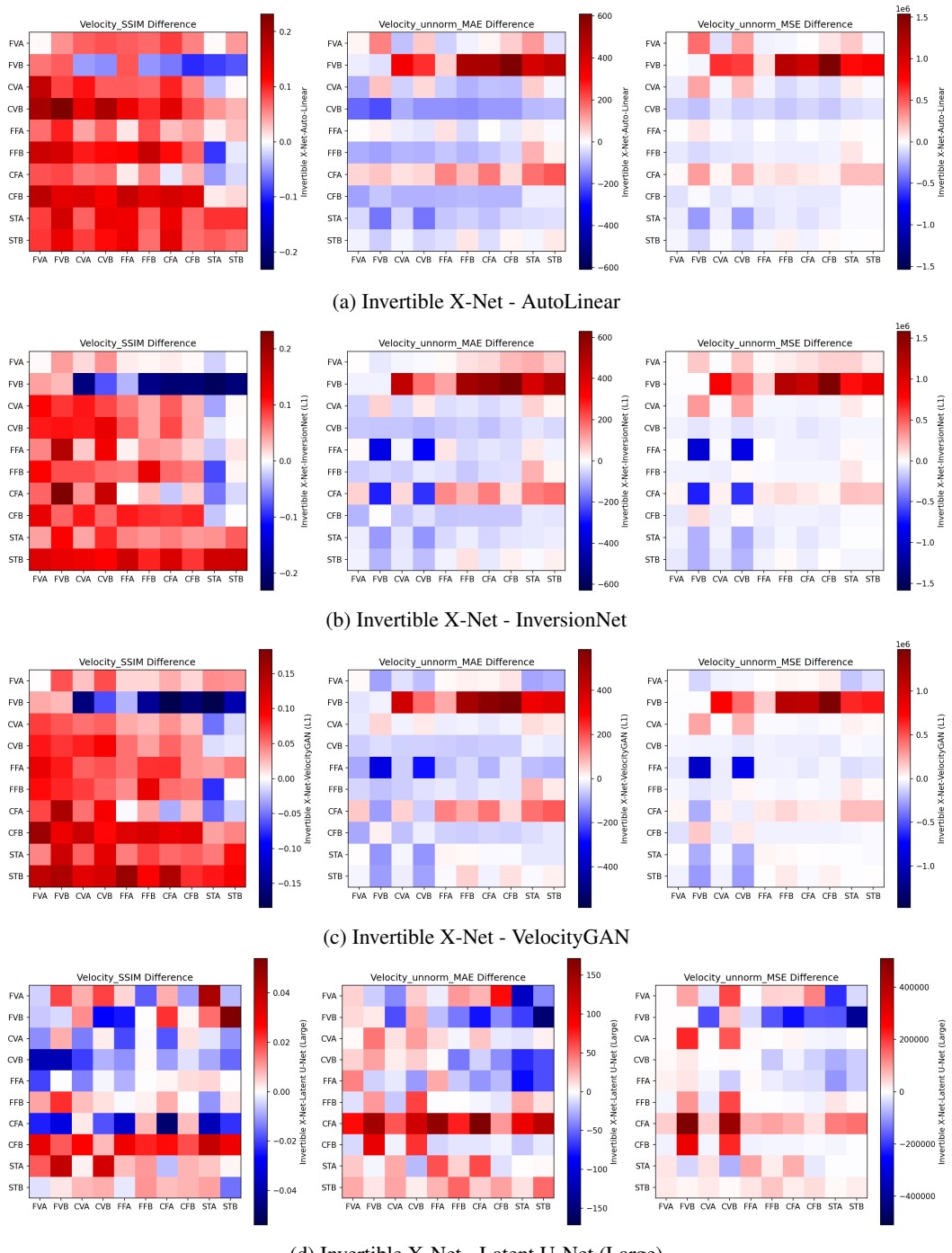

Figure 14: Out-of-distribution zero shot generalizations for the inverse problem of Invertible X-Net with AutoLinear, InversionNet, VelocityGAN, Latent U-Net (Large).

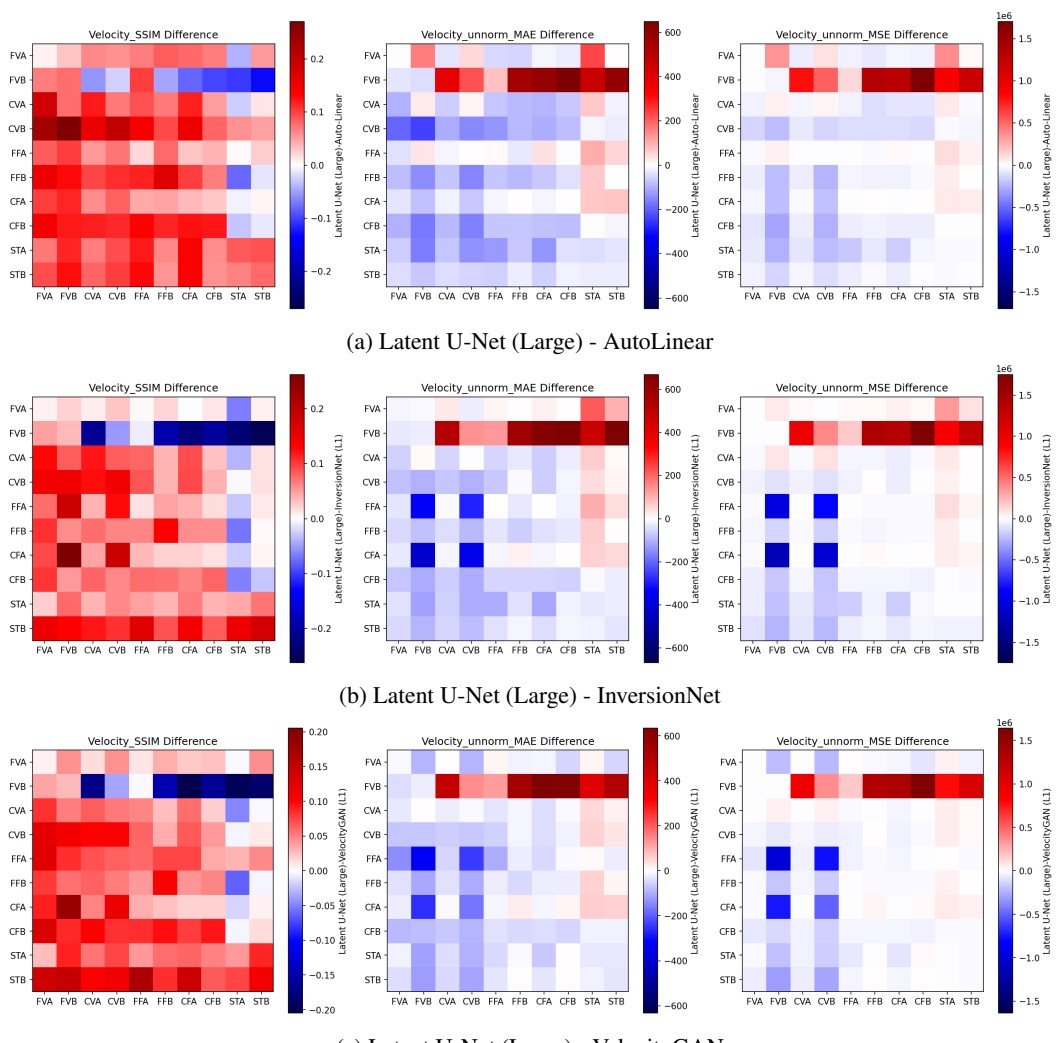

(a) Latent U-Net (Large) - AutoLinear

(b) Latent U-Net (Large) - InversionNet

(c) Latent U-Net (Large) - VelocityGAN

Figure 15: Out-of-distribution zero shot generalizations for the inverse problem of Latent U-Net (Large) with AutoLinear, InversionNet, VelocityGAN.

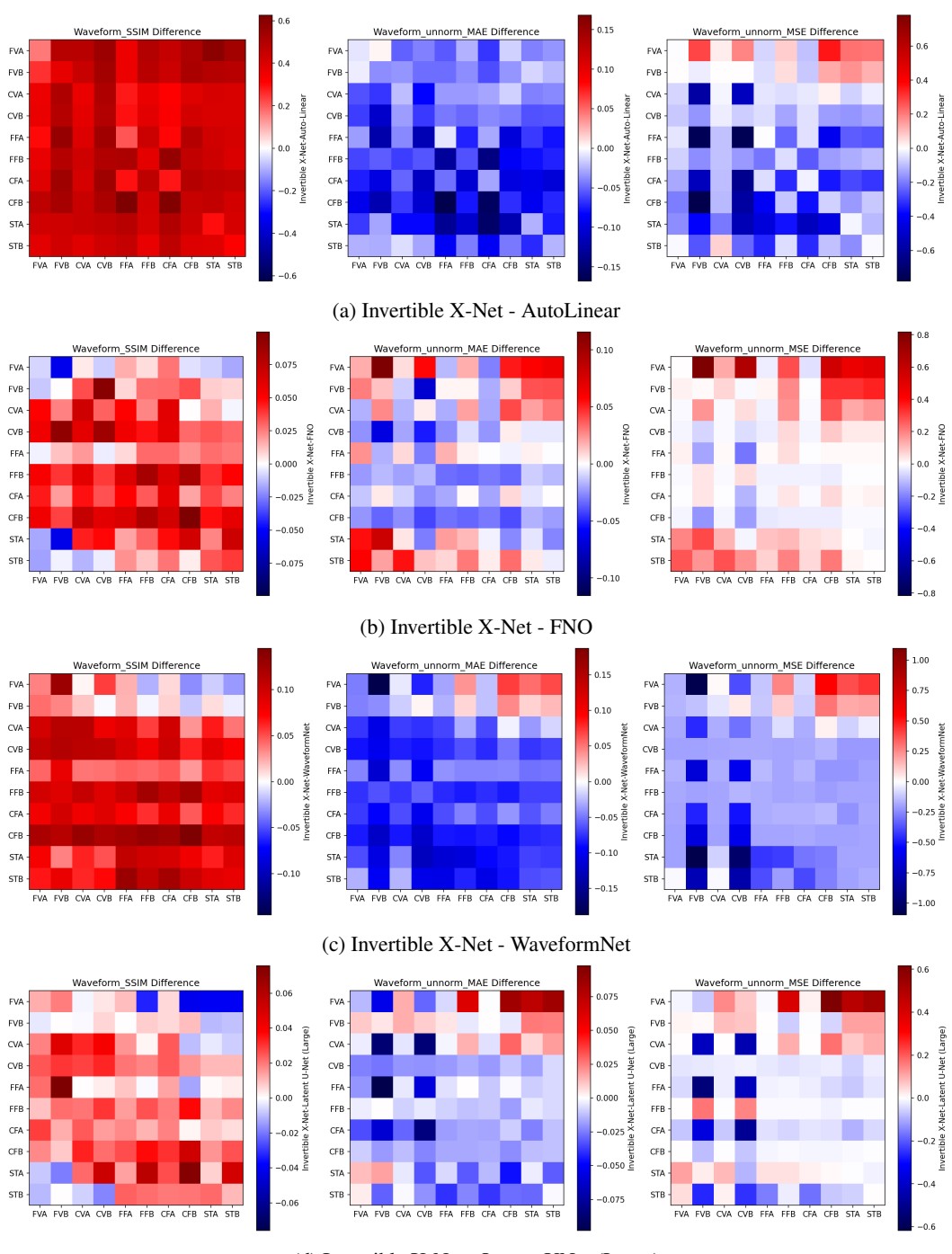

Figure 16: Out-of-distribution zero shot generalizations for the forward problem of Invertible X-Net with AutoLinear, FNO, WaveformNet (U-Net like model), and Latent U-Net (Large).

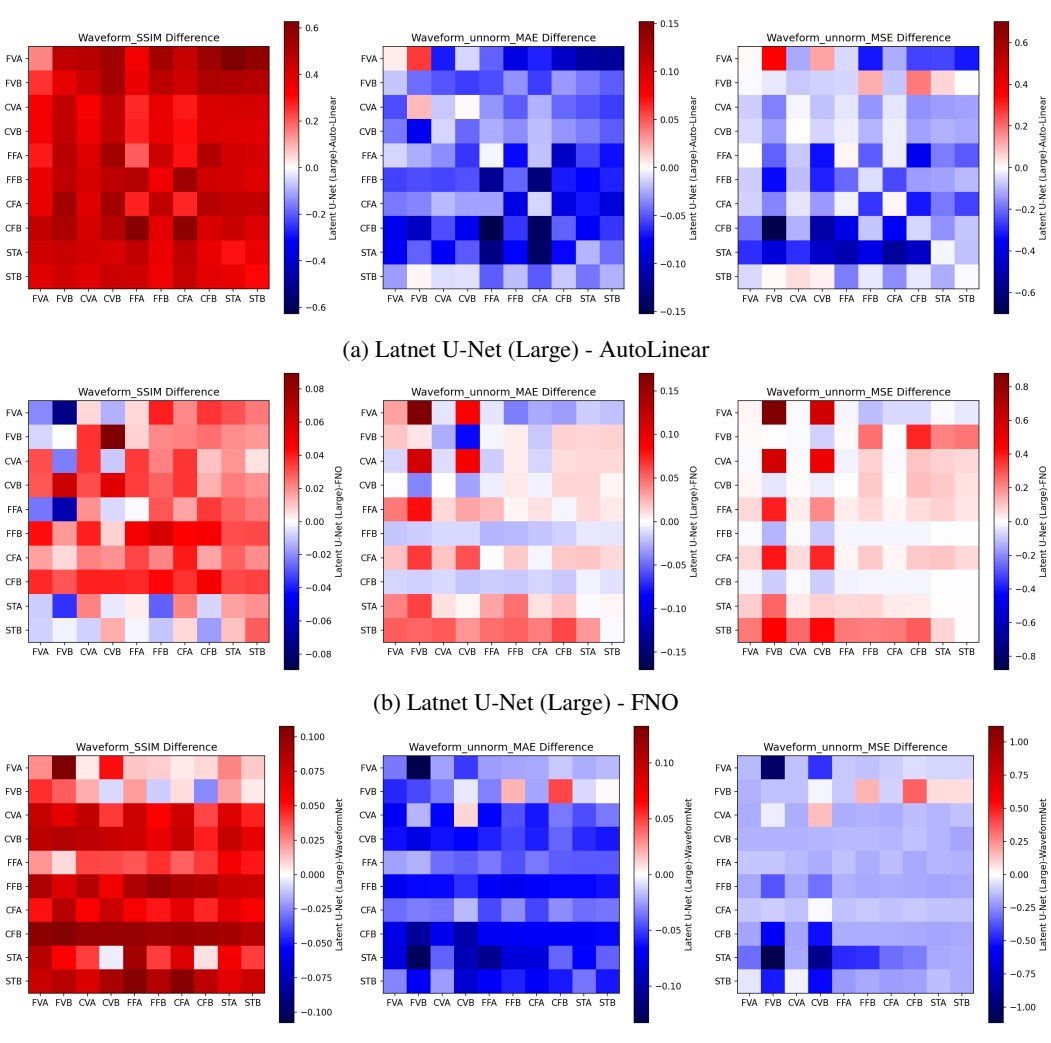

(a) Latnet U-Net (Large) - AutoLinear

(b) Latnet U-Net (Large) - FNO

(c) Latnet U-Net (Large) - WaveformNet

Figure 17: Out-of-distribution zero shot generalizations for the forward problem of Latent U-Net with AutoLinear, FNO, and WaveformNet (U-Net like model).

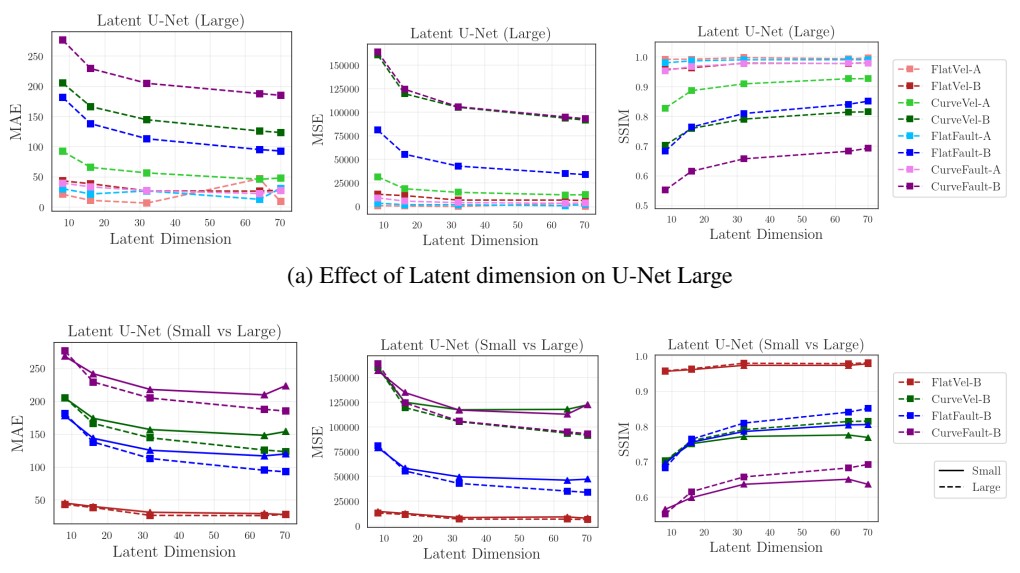

(a) Effect of Latent dimension on U-Net Large

(b) Effect of Latent dimension on U-Net Small in comparison to U-Net Small

Figure 18: Effect of the size of latent sizes on the performance of large and small Latent U-Net (small and large) on the OpenFWI datasets.

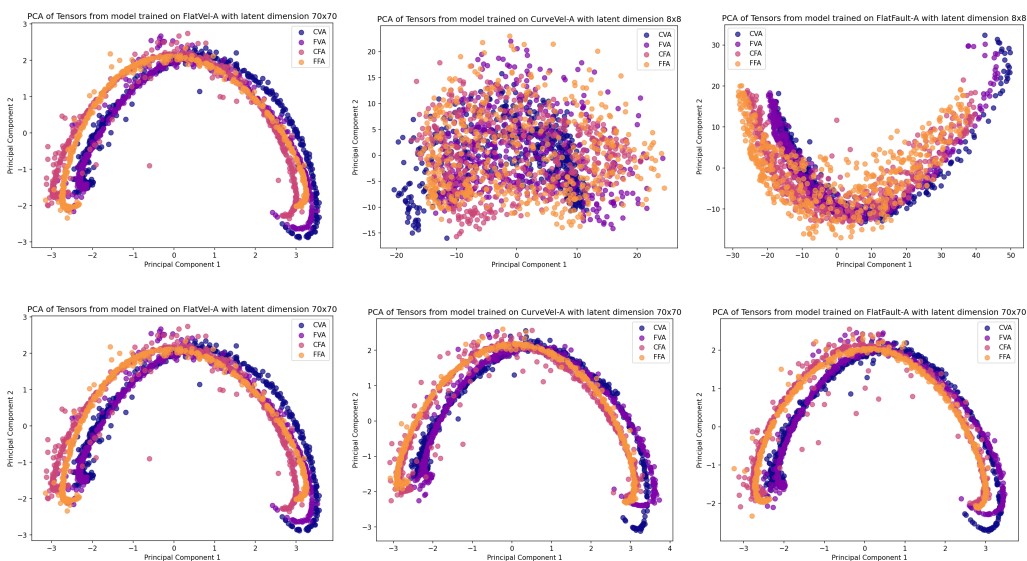

Figure 19: Visualizing the PCA projection of Velocity latent space for $8 \times 8$ and $70 \times 70$.

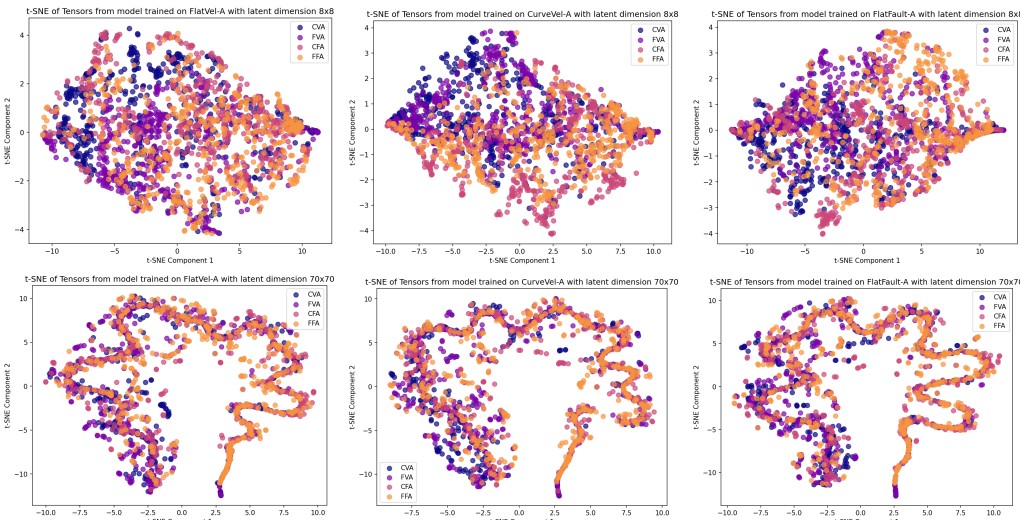

Figure 20: Comparing the t-SNE plot of latent space for velocity encoder.

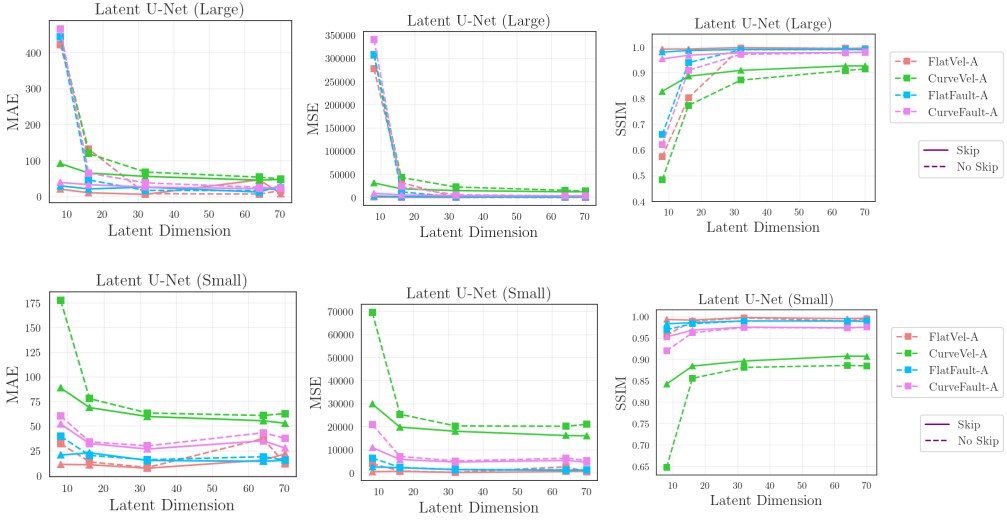

Figure 21: Effect of the size of latent space and skip vs no skip connections on the performance of large and small Latent U-Net models across OpenFWI datasets.

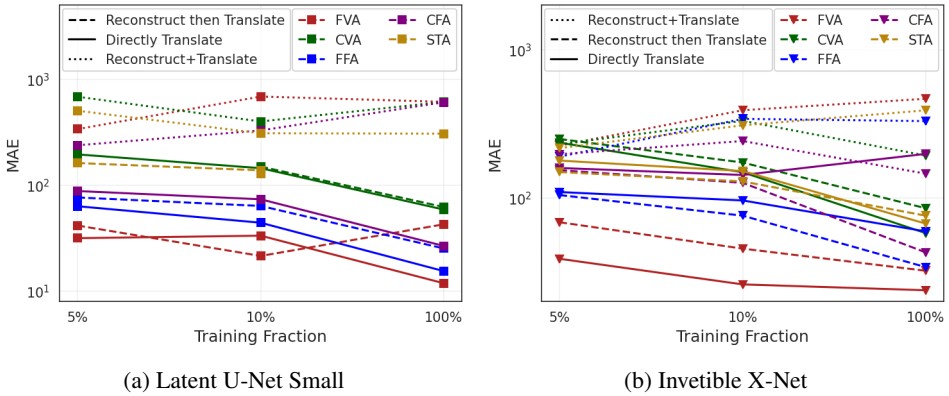

(a) Latent U-Net Small                    (b) Invetible X-Net

Figure 22: Comparison of Latent U-Net's and Invertible X-Net's performance across three learning objectives: translation directly, reconstruction followed by translation, and combined learning of both, evaluated at different training fractions.

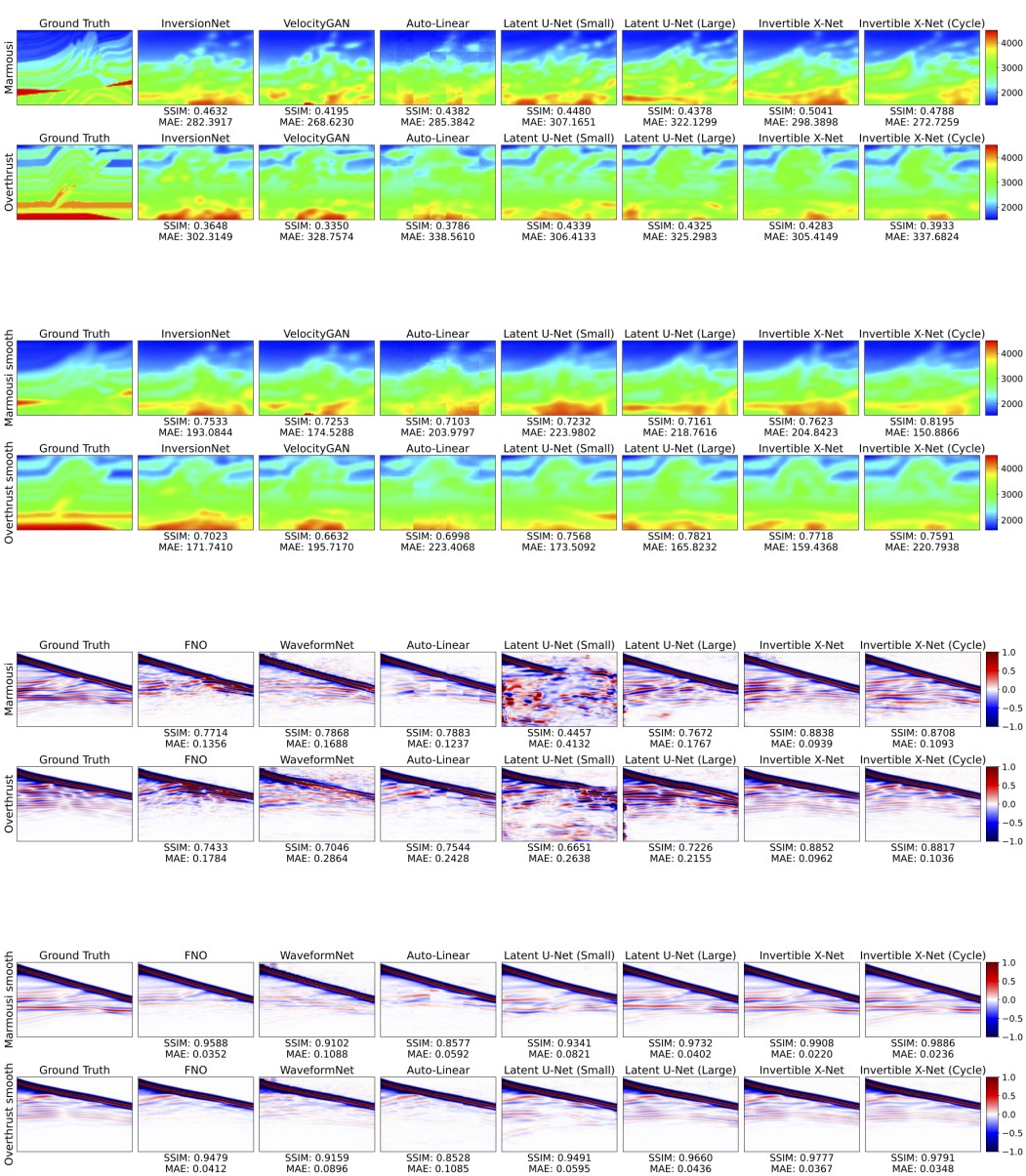

Figure 23: Zero shot generalization results of model trained on Style-A dataset on Marmousi and Overthrust dataset samples and their smoothened versions.

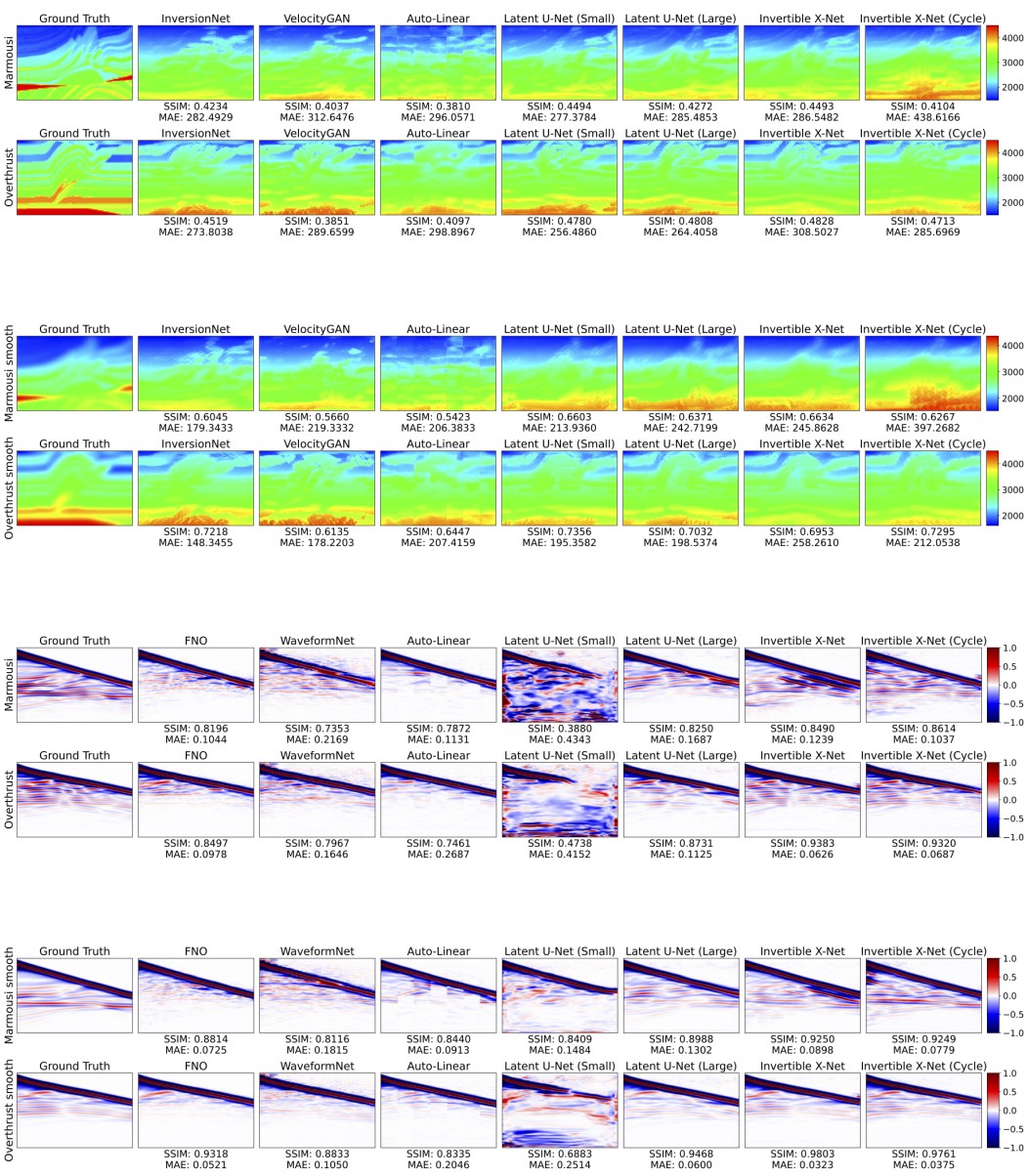

Figure 24: Zero shot generalization results of model trained on Style-B dataset on Marmousi and Overthrust dataset samples and their smoothened versions.

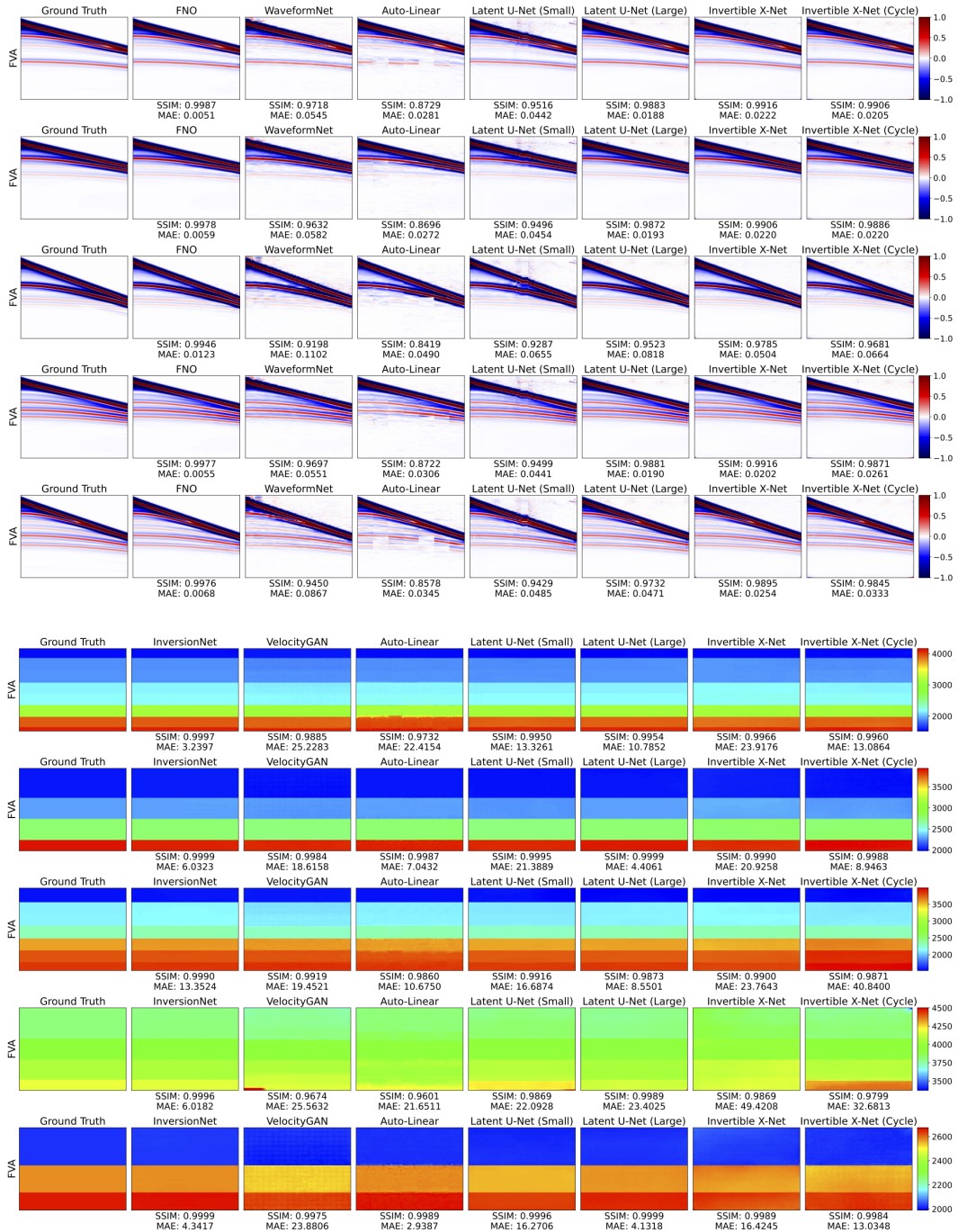

Figure 25: Visualization of predictions for forward and inverse problems on FVA dataset.

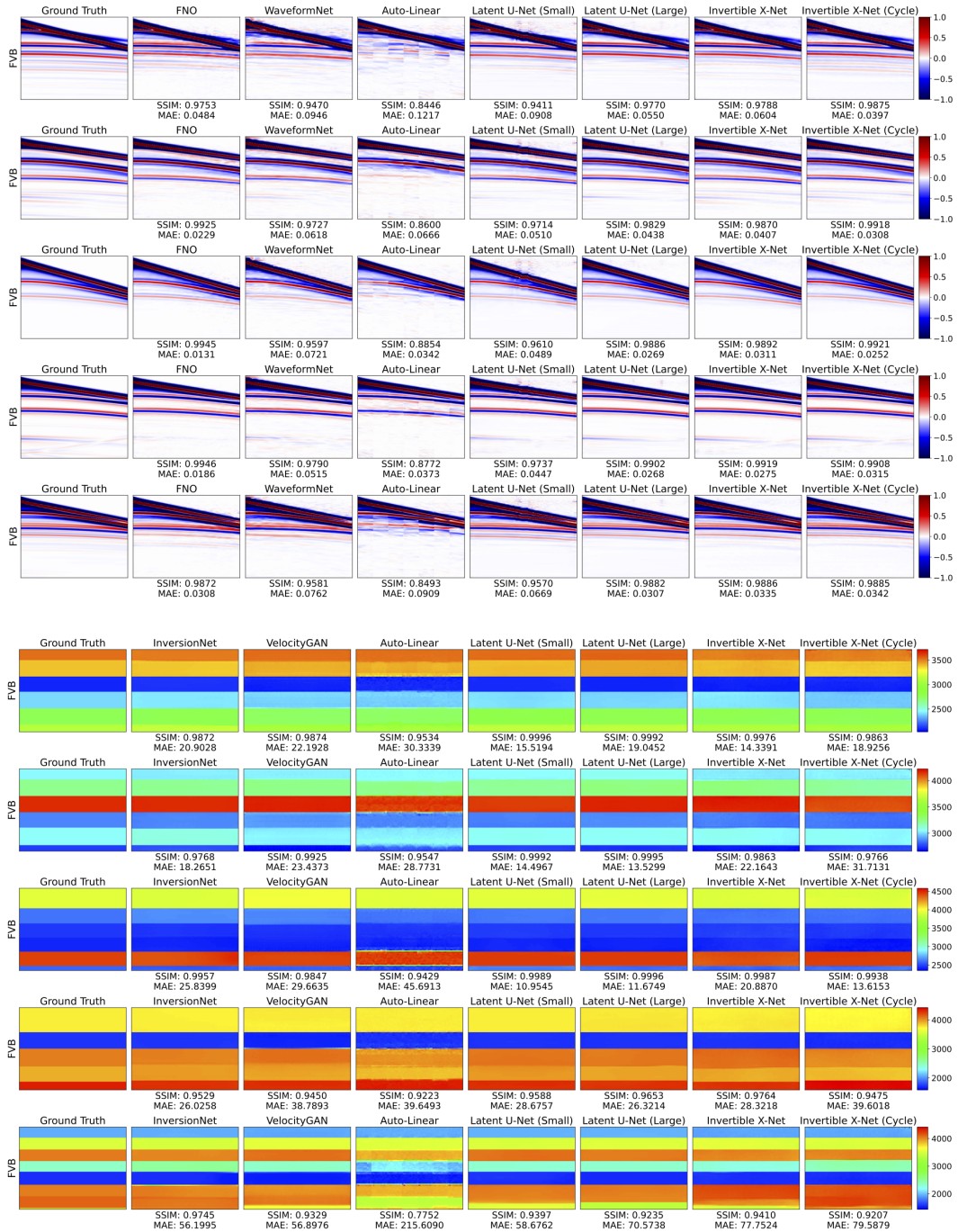

Figure 26: Visualization of predictions for forward and inverse problems on FVB dataset.

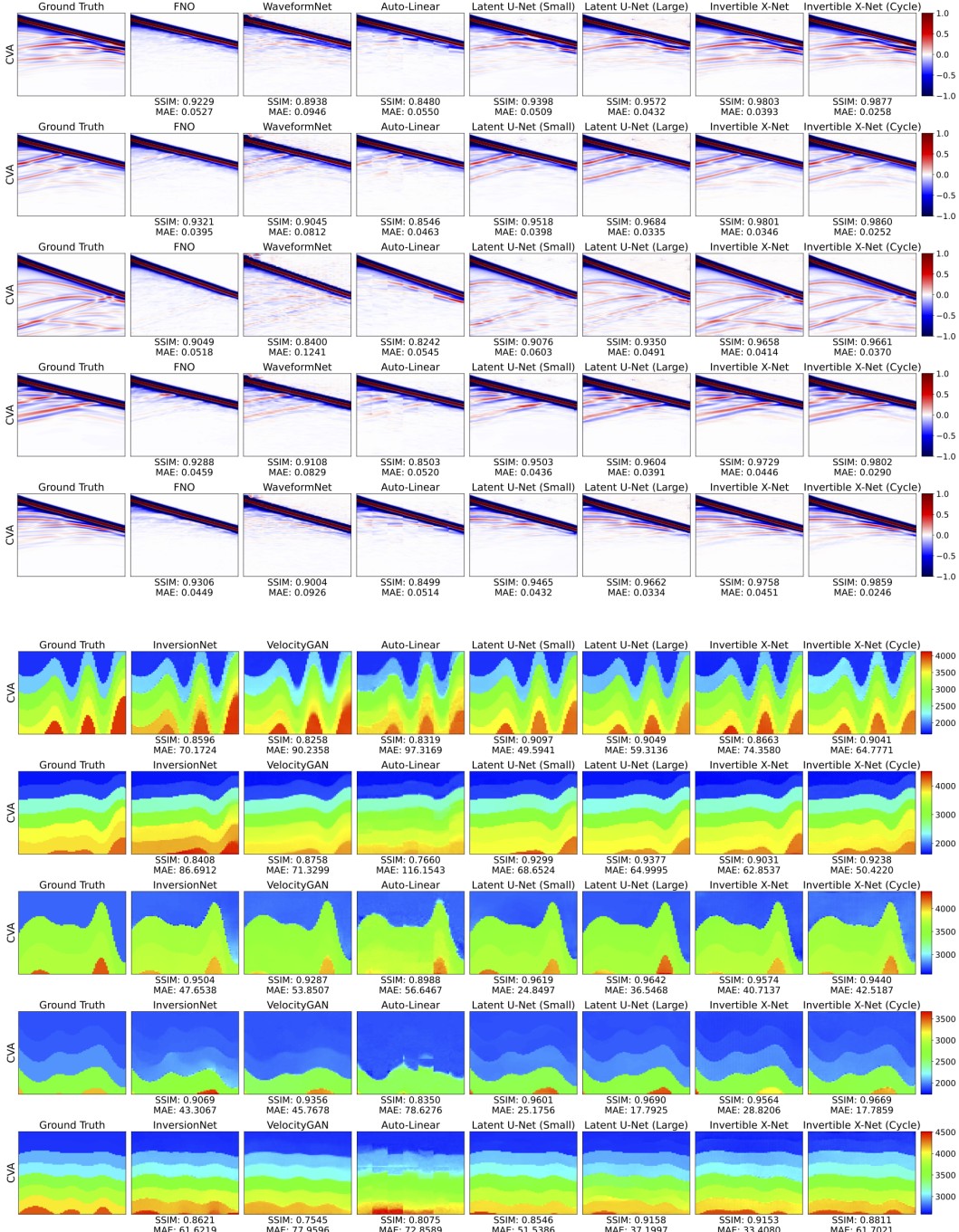

Figure 27: Visualization of predictions for forward and inverse problems on CVA dataset.

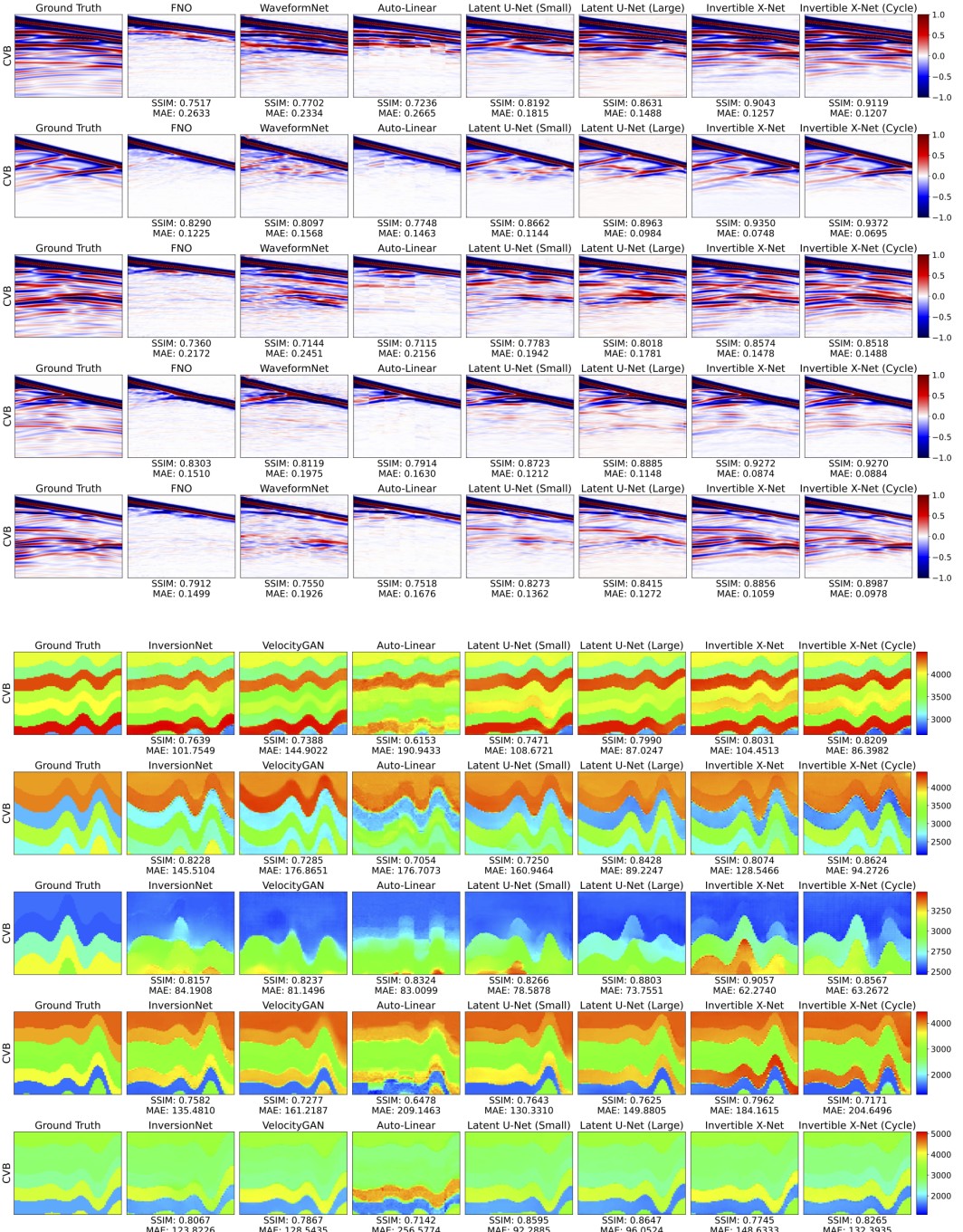

Figure 28: Visualization of predictions for forward and inverse problems on CVB dataset.

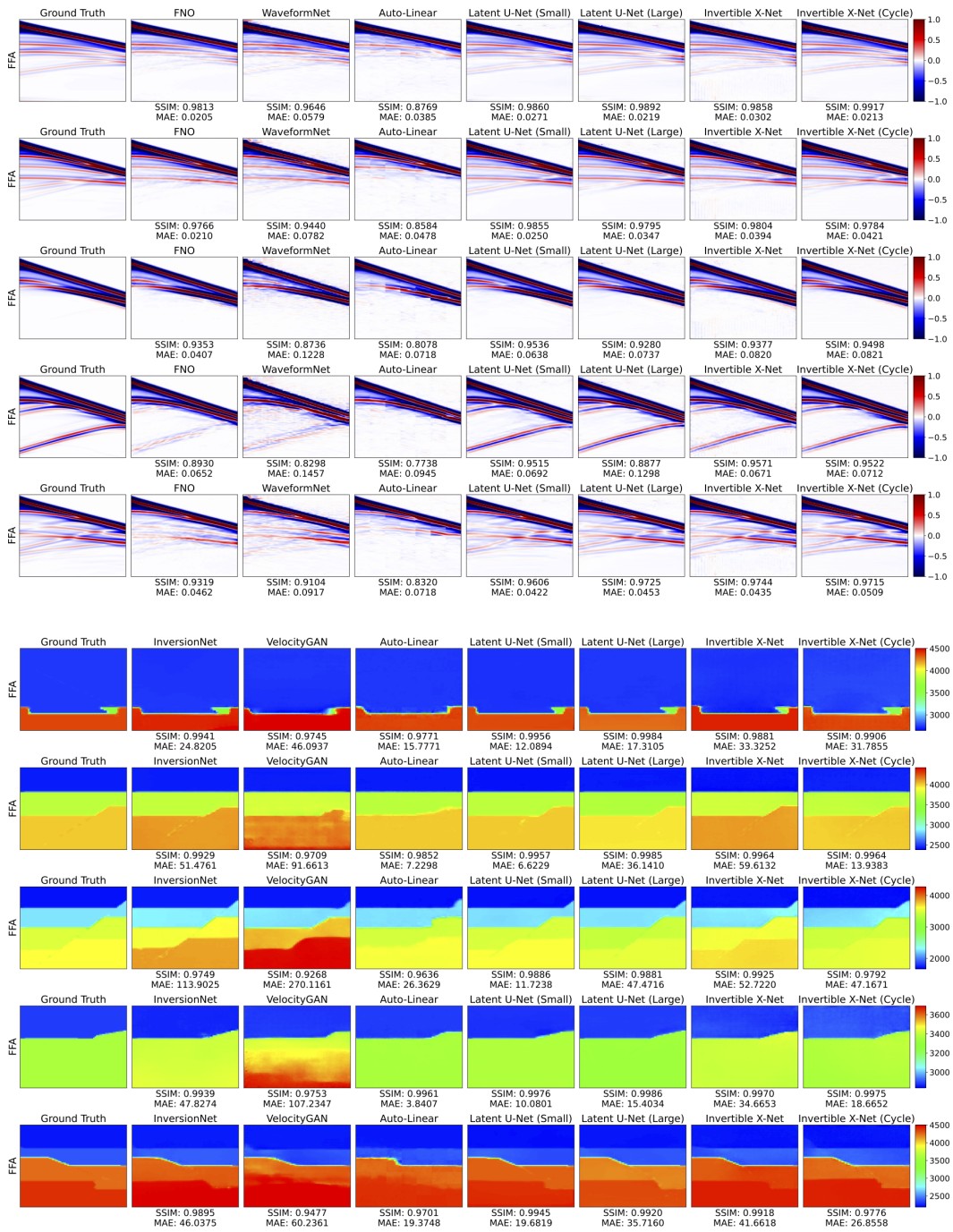

Figure 29: Visualization of predictions for forward and inverse problems on FFA dataset.

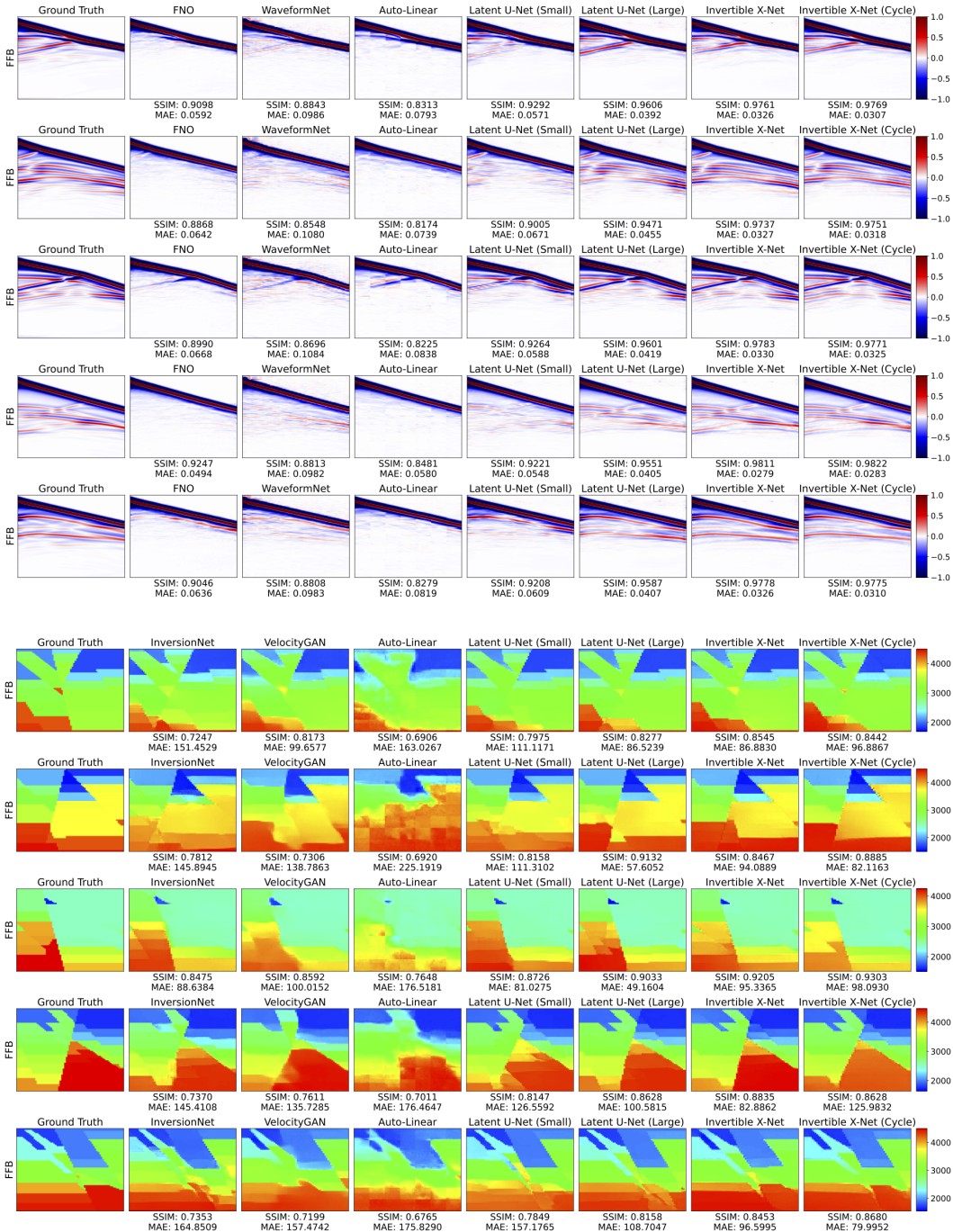

Figure 30: Visualization of predictions for forward and inverse problems on FFB dataset.

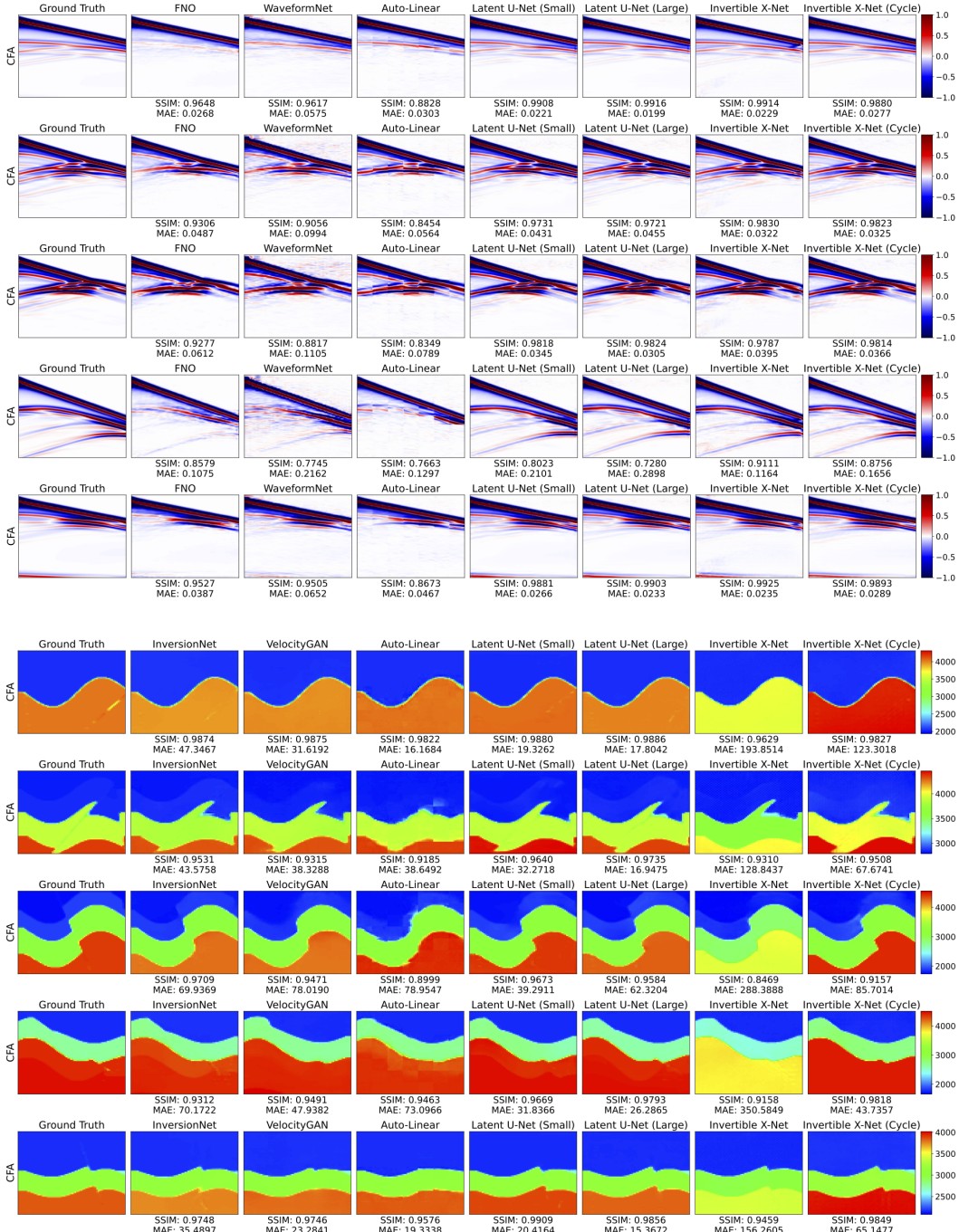

Figure 31: Visualization of predictions for forward and inverse problems on CFA dataset.

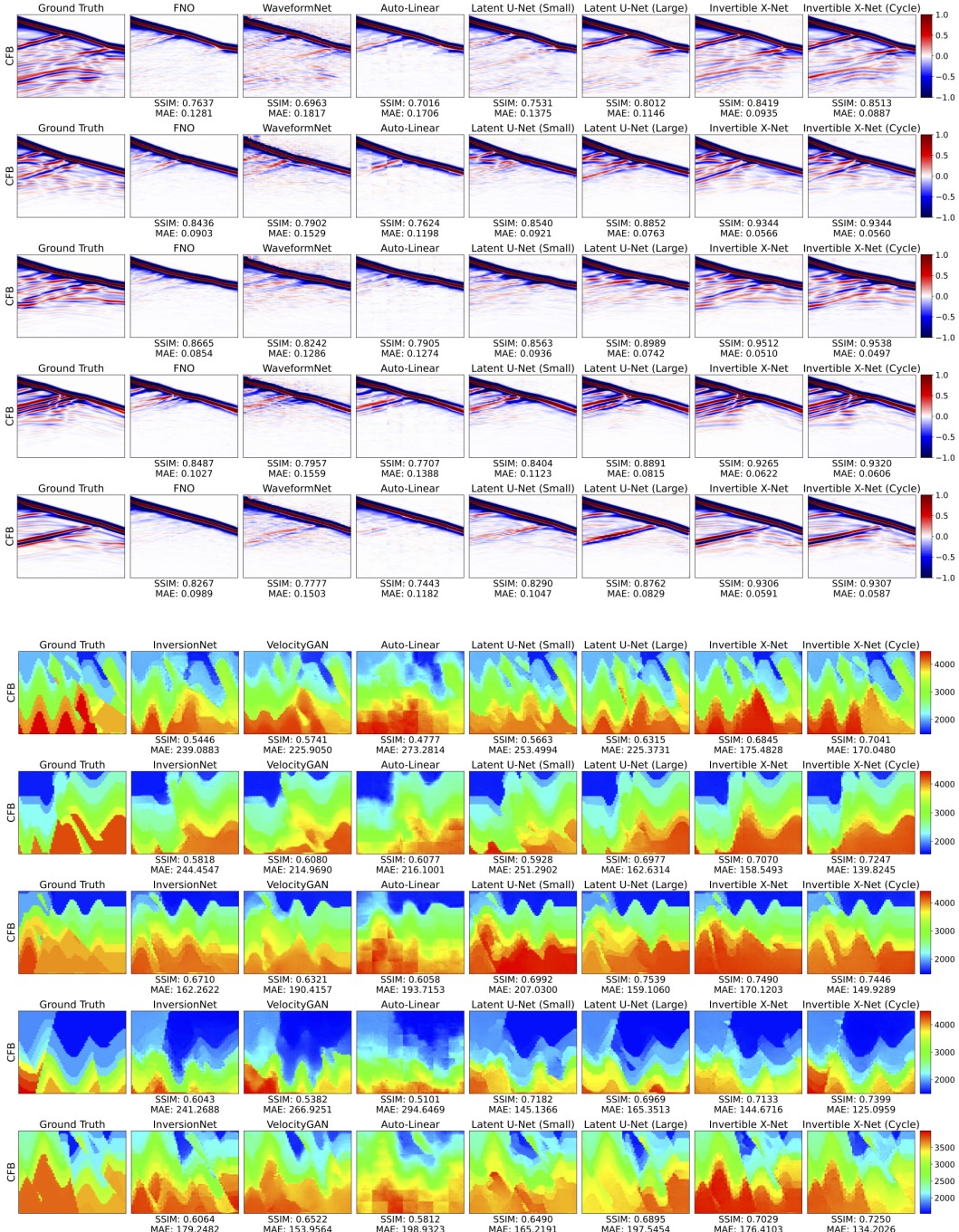

Figure 32: Visualization of predictions for forward and inverse problems on CFB dataset.

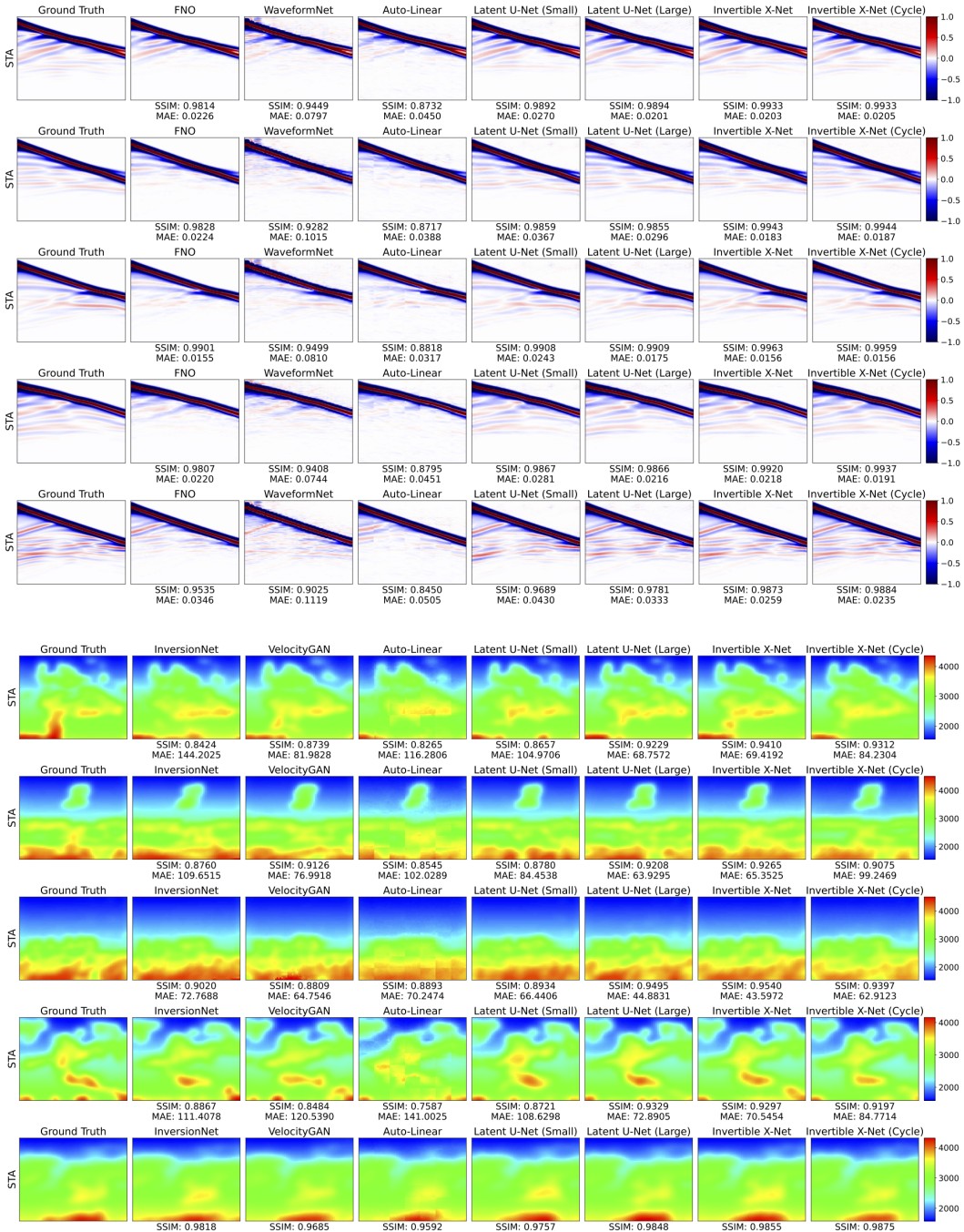

Figure 33: Visualization of predictions for forward and inverse problems on STA dataset.

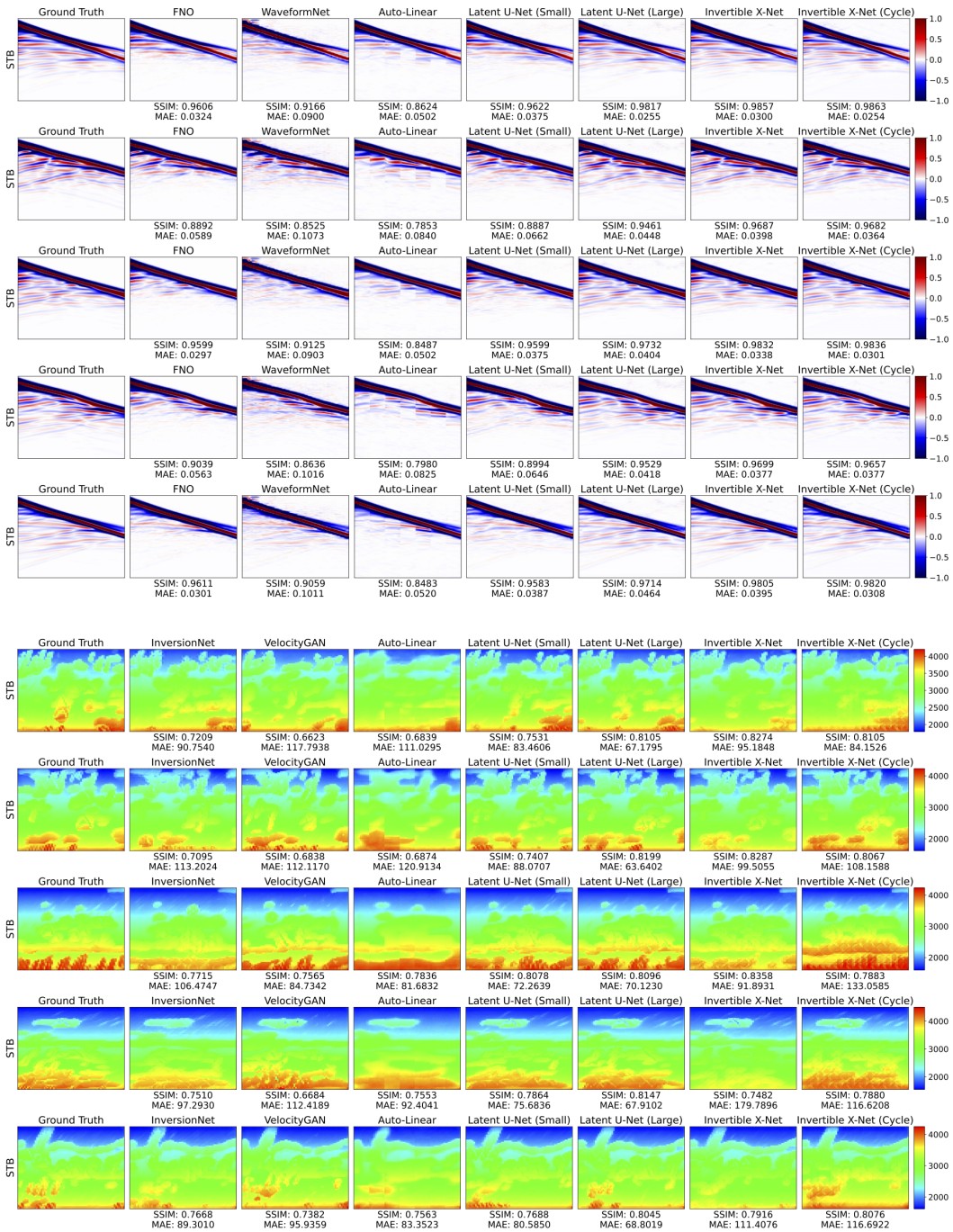

Figure 34: Visualization of predictions for forward and inverse problems on STB dataset.

