# OpenReview forum: "A Unified Framework for Forward and Inverse Problems in Subsurface Imaging using Latent Space Translations"
_ICLR.cc/2025/Conference — ICLR 2025 Poster_

### Official Review · Reviewer_nmvk · 2024-10-31

**Soundness:** 3
**Presentation:** 3
**Contribution:** 2
**Rating:** 6
**Confidence:** 5

**Summary:**

This work summarizes recent advances in deep learning-based mapping between velocity maps and seismic waveforms, introducing a unified framework (GFI) to systematically characterize prior research. Using GFI, key factors affecting mapping performance are investigated, alongside the proposal of two novel model architectures. Comprehensive experiments reveal the impact of these factors, with the proposed models achieving state-of-the-art (SOTA) performance in both forward and inverse problems.

**Strengths:**

1. This work is the first to investigate key factors influencing mapping performance.
2. Extensive experiments were conducted to conclude the influence of these factors.
3. Two novel networks are introduced for mapping between velocity maps and seismic waveforms, achieving SOTA performance.

**Weaknesses:**

1. This paper resembles a survey and might be more suitable for journal submission.
2. A significant portion of the content is placed in the appendices, making it less reader-friendly.
3. Aside from extensive experiments on factor influences, the work offers limited novelty and lacks theoretical analysis.
4. Additional factors should be investigated, such as the robustness of data noise across different network architectures.

**Questions:**

Invertible Neural Networks (INNs) often have limited flexibility in adjusting the dimensions of input and output, the number of layers, filters, and convolutional kernel sizes. These constraints can impede performance for complex tasks.

When mapping between velocity maps and seismic waveforms—data from two distinct domains (one spatial and the other spatio-temporal), why Invertible X-Net could demonstrate superior performance?

---

> ### Author Response · Authors · 2024-11-23
>
> **Comment 1: This paper resembles a survey and might be more suitable for journal submission. Aside from extensive experiments on factor influences, the work offers limited novelty and lacks theoretical analysis.**
>
> > One of the key innovations of our work is the novel Generalized Forward-Inverse (GFI) framework, which unifies forward and inverse problems in subsurface imaging. Within this framework, we propose two new architectures—Latent U-Net and Invertible X-Net—that achieve state-of-the-art performance on multiple datasets, including OpenFWI benchmarks. By systematically addressing forward and inverse problems, GFI bridges existing methods and provides a systematic approach to enhancing performance in scientific machine learning tasks.
>
> > Similar to recent works such as *"A Unified and General Framework for Continual Learning"* (ICLR 2024) [1] and *"Mirror Learning: A Unifying Framework of Policy Optimisation"* (ICML 2022) [2], which highlight the importance of unifying diverse methodologies under a single framework, our GFI framework provides a comprehensive lens to interpret, compare, and extend approaches to subsurface imaging. These frameworks demonstrate the value of unifying state-of-the-art approaches while introducing novel components, just as we do with Latent U-Net and Invertible X-Net. Moreover, our work complements practical advancements with theoretical insights into invertibility and latent space translation. This dual focus aligns with the trends seen in leading ML venues, reinforcing the relevance and novelty of our contributions.
>
> > Along with introducing GFI, we also propose a novel Invertible X-Net architecture that is the first state-of-the-art model designed to jointly solve forward and inverse problems using a single-step unified approach. Unlike prior methods such as InversionNet or AutoLinear, which treat these tasks independently, X-Net leverages a shared invertible latent space that allows for consistent and efficient bidirectional mapping between the spatial (velocity maps) and spatio-temporal (seismic waveforms) domains. This joint optimization is a key innovation, as it ensures that the learning for one task benefits the other, leading to more robust and accurate results across both tasks.
>
> > Regarding the lack of theoretical analysis, we have detailed the theoretical foundations of GFI in Appendix Section C, where we explain how invertibility and latent space translation enable this unified approach.
>
> > [1] Wang, Z., Li, Y., Shen, L., & Huang, H. A Unified and General Framework for Continual Learning. In The Twelfth International Conference on Learning Representations, ICLR 2024.
>
> > [2] Grudzien, Jakub, Christian A. Schroeder De Witt, and Jakob Foerster. "Mirror learning: A unifying framework of policy optimisation." International Conference on Machine Learning. PMLR, 2022.
>
> **Comment 2: Additional factors should be investigated, such as the robustness of data noise across different network architectures.**
>
>  > We thank the reviewer for this valuable suggestion. In response, we conducted comprehensive noise evaluation experiments to assess the robustness of Latent U-Net and Invertible X-Net under varying levels of Gaussian noise in the input data. For these experiments, Gaussian noise with varying variances was added to the input velocity or waveform, and the models’ predictions were evaluated based on these noisy inputs.
>
> > Detailed results for the CurveFault-B and FlatFault-B datasets are provided in Appendix Tables 11–14, covering both forward and inverse tasks. These results show that Latent U-Net and Invertible X-Net maintain strong performance even with low PSNR of input signals, exhibiting minimal degradation in accuracy compared to contemporary methods. The performance of Latent U-Net and Invertible X-Net for both forward and inverse problems at the noisiest signal level is better than the Auto-Linear predictions at the zero noise level.
>
> > For the inverse problem, the noise levels correspond to PSNR values of 84.07 dB, 77.08 dB, 74.07 dB, and 67.08 dB. Both Latent U-Net and Invertible X-Net demonstrate robust performance under varying noisy conditions. For instance, on CurveFault-B, the Latent U-Net and Invertible X-Net show degradation in SSIM of about 1.24% and 1.13% respectively at the noisiest signal level (PSNR = 67.08 dB), outperforming Auto-Linear with significant margin.
>
> > For the forward problem, the noise levels correspond to PSNR values of 56.02 dB, 49.03 dB, 46.02 dB, and 39.03 dB. Both Latent U-Net and Invertible X-Net demonstrate strong robustness at all noise levels. For example, on CurveFault-B, both Latent U-Net and Invertible X-Net show similar degradation in SSIM of about 0.82% and 0.73% respectively at highest noise level (PSNR = 39.03 dB). The performance of these models across all noise levels exceeds that of the Auto-Linear at the zero noise level for all metrics, indicating significant improvement for forward problem as well.

---

> > ### Author Response · Authors · 2024-11-23
> >
> > **Comment 3: Invertible Neural Networks (INNs) often have limited flexibility in adjusting the dimensions of input and output, the number of layers, filters, and convolutional kernel sizes. These constraints can impede performance for complex tasks.**
> >
> > > We appreciate the reviewer’s comment regarding the flexibility of INNs. While INNs are sometimes perceived as having limited adaptability, we are able to flexibly use iUNets by pairing them with encoder-decoder architectures in Invertible X-Net that can be adapted easily to the required input-output settings. We agree that the use of coupling blocks in iUNets can make the scaling of iUNets challenging compared to standard U-Nets. However, the other architectural components of iUNets are still flexible such as the number of layers, filters, and convolutional kernel sizes.
> >
> > > Also note that Invertible X-Net achieved best or second-best performance consistently across all datasets and experiments using a single architectural configuration. This suggests that its setup is not highly sensitive to hyperparameter variations, indicating robustness to diverse data characteristics.
> >
> > **Comment 4: When mapping between velocity maps and seismic waveforms—data from two distinct domains (one spatial and the other spatio-temporal), why Invertible X-Net could demonstrate superior performance?**
> >
> >  > We appreciate the reviewer’s question regarding the rationale behind Invertible X-Net’s performance in mapping between velocity maps (spatial domain) and seismic waveforms (spatio-temporal domain). Invertible X-Net’s latent space design enables it to learn a shared representation that facilitates bidirectional mapping across these distinct domains. This shared latent space captures structural similarities and efficiently models cross-domain relationships, allowing the network to leverage information from both domains effectively. This design makes Invertible X-Net particularly suitable for applications requiring joint optimization of forward and inverse tasks.
> >
> > > Furthermore, we note that existing baselines in full waveform inversion, such as InversionNet (a U-Net-based architecture) and AutoLinear are also learning mappings between spatial and spatio-temporal domains. We build on these prior works to introduce the unified GFI perspective for mapping between the two domains using a richer set of model architectures.
> >
> > **Comment 5: A significant portion of the content is placed in the appendices, making it less reader-friendly.**
> >
> > > As stated in our global response, to improve the readability of the Appendix section from the main paper, we have now added cross-references in the main paper to specific Appendix sections wherever appropriate. Additionally, key findings from the Appendix Tables and Figures,  summarized through average ranks, have been included in the main paper and will be used throughout the discussions of results in the main paper.

---

> > > ### Comment · Reviewer_nmvk · 2024-11-25
> > >
> > > Thanks. Most of my concerns have been resolved, and I have updated my rating accordingly.

---

> > > > ### Author Response · Authors · 2024-11-25
> > > > **Thank you!**
> > > >
> > > > Thank you very much for your feedback and support!

---

### Official Review · Reviewer_74KD · 2024-10-31

**Soundness:** 3
**Presentation:** 3
**Contribution:** 2
**Rating:** 6
**Confidence:** 3

**Summary:**

The paper proposes a new training scheme combined with two new model architectures for the problem of learning the forward and inverse map in deep learning for subsurface imaging.
The training scheme consists of jointly learning the encoders and decoders for the two domains (velocity maps and seismic waveforms) with the translation network transforming the latent space representations from one domain to another.
The two architectures differ in the characteristics of the translation network with one consisting of two separate unidirectional U-nets (latent U-net) and the other of a single bidirectional invertible network (invertible X-net).
Both architectures achieve a what seems to be significant improvement in performance over existing works.
Finally the paper formulates a unifying framework in which existing approaches can be categorized and distinguished.

**Strengths:**

1.	(Quality) High quality of experimental evaluation: Experiments are extensive over a range of datasets including an extensive evaluation of out-of-distribution performance and qualitative results are provided in addition to quantitative results. Hence, the conclusion of generally improved performance seems sound.
2.	(Significance) The improvements in performance seem to be significant, however interpretability of how significant could be improved (see weaknesses point 5)
3.	(Significance) The finding that jointly solving forward and inverse problems is helpful for the forward problem but not for the inverse problem is interesting.
4.	(Originality) The proposed unifying framework helps to quickly understand the different existing approaches and will be helpful for future works to expand the framework or propose improvements within it.
5.	(Clarity) The paper is overall well written and easy to follow also for researches not actively involved in the field of subsurface imaging.

**Weaknesses:**

Major concerns

1. (Clarity) The experiment 5.1.3 addresses the question ‘should we train latent spaces solely for reconstruction?’, which I assume corresponds to the question 3) formulated in line 083 ‘what is the role of manifold learning?’.
1.1 For increased clarity I’d suggest not changing the phrasing of the questions.
1.2 The setup is unclear to me. What are training fractions? A fraction of the training set or a fraction of the training time? And why is it instructive to look at 5% and 10% training fraction?
1.3 So, is the conclusion form the experiment that learning manifolds is not important? It is difficult to grasp the conclusion if the wording is not consistent. Is “influence the training of the latent spaces based on the translation objectives” identical to “not learning manifolds”?
1.4 To better answer the question I think the following baseline is missing that combines the two alternatives (reconstruct then translate and directly translate) into directly learning the translation networks while having at the same time reconstruction losses (basically what is described in lines 243-245). Seeing how this combination performs would help answering the question if manifold learning helps.
2. (Originality) The idea of the Latent U-net seems to be very similar to the InversionNet following the comparison in Table 1.
2.1 The Modelling Mode is the same because the forward and inverse part of the Latent U-net are completely disjoint and hence correspond to an InversionNet and a ForwardNet (as long as no manifold learning is performed).
2.2 While the latent U-net allows manifold learning, the experimental results pertain to the case without manifold learning same as the InversionNet.
2.3 Latent space  translation is said to be identity for the InversionNet and U-net for the Latent U-net, but couldn’t we also just claim some middle layers of the InversionNet to perform latent space translation?
2.4 The size of the latent space is low in both cases.
--> So to my understanding the difference between the two boils solely down to the architectural design and size of the network but not to any conceptual differences. If that is the case, it should be stated more directly in the paper and the exact architectural differences should be explained in more detail.
3. (Clarity) From Section 5.1.4 it is not entirely clear what the answer to the question raised in the paragraph (Is it useful to jointly solve forward and inverse problem) is. Is the answer that no it is not useful for learning the inverse map but yes it is useful for learning the forward map (both answers based on the findings in Figure 3)?

Minor concerns

4. (Clarity) Please provide links to the exact Section in the Appendix rather than just referencing the entire appendix (e.g. lines 308, 311, 314, 377, 388).
5. (Significance) Include quantitative scores in the qualitative examples (e.g. in Figures 4 and 8) to give a better understanding of how differences in scores translate to significantly perceptible differences.

**Questions:**

See weaknesses.

---

> ### Author Response · Authors · 2024-11-23
>
> **Comment 1: The experiment 5.1.3 addresses the question ‘should we train latent spaces solely for reconstruction?’, which I assume corresponds to the question 3) formulated in line 083 ‘what is the role of manifold learning?’.
> 1.1 For increased clarity I’d suggest not changing the phrasing of the questions**
>
> > We apologize for the inconsistency in our section titles. We have now revised the phrasing of questions to ensure consistency throughout the paper. Section 5.1.3 is now renamed as “What is the role of manifold learning in DL4SI?”
>
>
> **Comment 2: What are training fractions? A fraction of the training set or a fraction of the training time? And why is it instructive to look at 5% and 10% training fraction?**
>
> > Training fraction refers to the subset of the dataset used for training, where the remaining data is masked and not utilized. In our experiments, we followed the same precedent for training with lesser data as followed in AutoLinear, which evaluated model performance using 5%, 10%, and 100% training fractions on the same dataset. Evaluating model performance across different training fractions provides insights into model robustness and efficiency, particularly in data-constrained scenarios that are often encountered in real-world applications.
>
>
> **Comment 3:  Importance of manifold learning in the GFI framework and inclusion of reconstruction loss for evaluation.**
>
>
> > We appreciate the reviewer’s suggestion to explore a third learning objective combining reconstruction and translation losses. Following this, we trained all three models—Latent U-Net (Large), Latent U-Net (Small), and Invertible X-Net—on this combined objective across various datasets. These results have been included in the updated Figure 6 in the main paper and Figure 22 in the Appendix.
> > Our findings indicate that directly learning to translate yields better performance for translation tasks compared to a two-step approach of first learning a latent space optimized for reconstruction followed by translation. Directly learning to translate (without using reconstruction loss) is also better than simultaneously optimizing both reconstruction and translation loss terms. This suggests that translation and reconstruction can have competing objectives, where a manifold optimized for reconstruction may not be best-tailored for translation. Hence, manifold learning, while beneficial for reconstruction, may lead to suboptimal performance when translation is the primary goal. We have now included this discussion in the main paper.
>
> **Comment 4: The idea of the Latent U-net seems to be very similar to the InversionNet following the comparison in Table 1. So to my understanding the difference between the two boils solely down to the architectural design and size of the network but not to any conceptual differences. If that is the case, it should be stated more directly in the paper and the exact architectural differences should be explained in more detail.**
>
> > We agree with the reviewer that the differences between Latent U-Net and InversionNet primarily stem from architectural design. As suggested by the reviewer, InversionNet can be viewed as a special case of Latent U-Net, lacking skip connections and incorporating a smaller bottleneck layer. Both InversionNet and Latent U-Net (Large) have similar numbers of parameters. We also agree that some of the middle layers of InversionNet are performing latent space translation. However, the Latent U-Net (Large) has more layers in its translation model (U-Net) while having lighter encoder-decoder models compared to InversionNet. Architectural differences, such as skip connections, encoder/decoder size, latent dimension size, and the choice of translation architecture (e.g., U-Net vs. iUNet) play significant roles on model performance, as explored through the four salient questions of our paper. Note that GFI represents a spectrum of methods where InversionNet is just one point in the spectrum. By drawing connections between InversionNet and Latent U-net, we are expanding the space of model choices available for DL4SI.

---

> > ### Author Response · Authors · 2024-11-23
> >
> > **Comment 5: From Section 5.1.4 it is not entirely clear what is the answer to the question: Is it useful to jointly solve forward and inverse problems? Is the answer that no it is not useful for learning the inverse map but yes it is useful for learning the forward map (both answers based on the findings in Figure 3)?**
> >
> > > As stated in our global response, Invertible X-Net shows best performance for forward problems and second-best performance for inverse problems across all datasets. This shows that jointly solving forward and inverse problems via Invertible X-Net is better than only solving the forward problem via Latent U-Net. However, if the goal is to only solve the inverse problem, we find that jointly solving forward and inverse via Invertible X-Net is not as optimal as using Latent U-Net. Hence, we recommend using Invertible X-Net if the goal is to solve both forward and inverse problems, or just the forward problem. Additionally, Invertible X-Net is highly parameter-efficient when it comes to jointly solving forward and inverse problems, requiring only 26M parameters compared to the 70M parameters of Latent U-Net (Large).
> >
> > **Comment 6: Please provide links to the exact Section in the Appendix rather than just referencing the entire appendix (e.g. lines 308, 311, 314, 377, 388).**
> >
> >  > We apologize for any inconvenience caused by the lack of specific cross-references to Appendix Sections in the earlier version. We have now added all references to the relevant sections in the Appendix that are easy to navigate from the main paper, for improved clarity and accessibility.
> >
> > **Comment 7: Include quantitative scores in the qualitative examples (e.g. in Figures 4 and 8) to give a better understanding of how differences in scores translate to significantly perceptible differences.**
> >
> >  > As stated in the global response, we have updated Figures 4 and 8 as well as other visualizations in the Appendix to include both MAE and SSIM metrics.

---

> ### Comment · Reviewer_74KD · 2024-11-25
> **Response to official comment by authors**
>
> Thank you for your responses.
>
> I went through the revised versions and find that the clarity of certain statements is improved as asked for in my review.
> Further, I think the additional experiments with the joint reconstruction and translation loss are interesting and improve the discussion of the question on training latent spaces for reconstruction.
>
> I adjusted my score accordingy. In summary I would recommend the paper for acceptance as I think the unifying framework and the advancement of the state of the art is interesting for the audience. The reason for not further raising the score is that I find the novelty of the best performing method (latent U-net) to be somewhat limited to equipping the idea behind InversionNet with a better network architecture (which I still think should be communicated and compared more clearly in the paper).

---

> > ### Author Response · Authors · 2024-11-25
> > **Thank you!**
> >
> > Thank you very much for your feedback and support! We will revise our discussion of InversionNet to make its connection with Latent U-Net more apparent in the revised version, as described in our response.

---

### Official Review · Reviewer_VWh3 · 2024-11-02

**Soundness:** 3
**Presentation:** 2
**Contribution:** 2
**Rating:** 5
**Confidence:** 3

**Summary:**

This paper explores deep learning methods to address the forward and inverse problems in subsurface imaging by mapping between velocity maps and seismic waveforms. The authors introduce the Generalized Forward-Inverse (GFI) framework, which unifies past approaches and addresses open questions around latent spaces, manifold learning, and model complexity. They propose two novel architectures, Latent U-Net and Invertible X-Net, which achieve state-of-the-art results on synthetic datasets and show promising zero-shot performance on real-world-like datasets.

**Strengths:**

The paper presents a unified architecture for addressing both forward and inverse subsurface imaging problems, while previous studies typically focus only on inversions. Modeling these problems together within a single framework could offer valuable insights, though it may also introduce potential challenges.

**Weaknesses:**

1. The paper’s narrow focus on subsurface imaging may limit its relevance for a broader audience at ICLR. If there are analogous problems in other fields where their framework could be applied?
2. Comparisons to prior work are outdated, with main competing methods from 2019, like InversionNet and VelocityGAN. The authors are encouraged to consider more recent advances, such as “Physics-Informed Robust and Implicit Full Waveform Inversion Without Prior and Low-Frequency Information (2024)” and other state-of-the-art methods from 2023, for a fairer and more relevant assessment.
3. While the authors aim to model the forward problem using neural networks, it is uncertain if the learned mapping can capture complex physical dynamics, as seen in the governing equations of seismic data. Extensive out-of-domain experiments are recommended to assess the model’s robustness and generalizability to real-world data.

4. The paper's structure could be streamlined to highlight the proposed methodology. Currently, a substantial background section precedes a brief description of the methods. Specific sections of the introduction and related work and preliminary knowledge could be condensed.

5. The model specifications lack detail; adaptations made to standard UNet for subsurface imaging and the mechanisms enabling X-Net’s invertibility are unclear. Clarifying these aspects and specifying when each model might be preferred would improve understanding.
6. Experimental comparisons with recent baselines (published within the last 2-3 years) would add rigor to the evaluation, as several recent methods could serve as relevant benchmarks. To name a few, “Physics-Informed Robust and Implicit Full Waveform Inversion Without Prior and Low-Frequency Information 2024”; “Full-waveform inversion using a learned regularization 2023”; “Wasserstein distance-based full-waveform inversion with a regularizer powered by learned gradient 2023”; “Full-waveform inversion using a learned regularization 2023”, and more.
7. Key numerical results in Section 5 are deferred to the appendix, which reduces clarity. Summarizing key findings directly in the main text would enhance readability.
8. The Invertible X-Net significantly underperforms other models, raising questions about its ideal use cases. Further context on scenarios where the X-Net's design offers advantages would be valuable.

**Questions:**

please see my concerns in the Weakness

---

> ### Author Response · Authors · 2024-11-23
>
> **Comment 1: The paper’s narrow focus on subsurface imaging may limit its relevance for a broader audience at ICLR. If there are analogous problems in other fields where their framework could be applied?**
> > Thank you for the suggestion to discuss our work’s broader applicability outside of the domain of subsurface imaging. Our proposed GFI framework is generic enough to be applied in any domain-to-domain translation problem, especially where the input and output domains are from two different spaces with varying dimensionalities. This includes applications in medical imaging, climate modeling, and material property predictions. For example, full-waveform inversion has been successfully applied to neuroimaging, enabling non-invasive 3D imaging of the human brain with sub-millimeter resolution [1]. Our work provides a generalized perspective of solving forward and inverse problems for domain translation, using encoder-decoder blocks coupled with a translation model. While our work analyzes different variations of GFI for applications in seismic imaging, similar analyses can be performed for other domain translation problems. We will include this broader motivation in the revised manuscript.
>
> > We would also like to highlight that research in deep learning for subsurface imaging (DL4SI) is gaining growing attention at top-tier venues in machine learning, with influential papers such as AutoLinear [2], OpenFWI [3], UPFWI [4], and InvLINT [5] being published at ICML, NeurIPS, and ICLR. This reflects the broader scope of research in DL4SI beyond the application of seismic imaging.
>
> > [1] Guasch, L., Calderón Agudo, O., Tang, M. X., Nachev, P., & Warner, M. (2020). Full-waveform inversion imaging of the human brain. NPJ digital medicine, 3(1), 28.
>
> > [2] Y. Feng, Y. Chen, P. Jin, S. Feng, Y. Lin, "Auto-Linear Phenomenon in Subsurface Imaging ." ICML, PMLR:235, 2024, pp. 13153-13174.
>
> >[3] C. Deng, S. Feng, H. Wang, X. Zhang, P. Jin, Y. Feng, Q. Zeng, Y. Chen, Y. Lin, "OpenFWI: Large-scale Multi-structural Benchmark Datasets for Full Waveform Inversion," Thirty-sixth Conference on Neural Information Processing Systems (NeurIPS), Datasets and Benchmarks Track, 2022.
>
> >[4] P. Jin, X. Zhang, Y. Chen, S. X. Huang, Z. Liu, Y. Lin, "Unsupervised Learning of Full-Waveform Inversion: Connecting CNN and Partial Differential Equation in a Loop," ICLR, 2022
>
> >[5] Y. Feng, Y. Chen, S. Feng, P. Jin, Z. Liu, Y. Lin, "An Intriguing Property of Geophysics Inversion," Proceedings of the 39th International Conference on Machine Learning (ICML), PMLR:162, 2022, pp. 6434–6446

---

> ### Author Response · Authors · 2024-11-23
>
> **Comment 2: Comparisons to prior work are outdated, with main competing methods from 2019, like InversionNet and VelocityGAN. Experimental comparisons with recent baselines (published within the last 2-3 years) would add rigor to the evaluation, as several recent methods could serve as relevant benchmarks. To name a few, “Physics-Informed Robust and Implicit Full Waveform Inversion Without Prior and Low-Frequency Information 2024”; “Full-waveform inversion using a learned regularization 2023”; “Wasserstein distance-based full-waveform inversion with a regularizer powered by learned gradient 2023” and more.**
>
> > We thank the reviewer for highlighting these impactful works [2,3,4], which we are happy to include in our related works discussion. While these methods focus on solving inverse problems using physics-informed priors, they can be categorized as *per-sample* methods according to the taxonomy introduced in [1]. In particular, a different model is optimized in these methods to solve a single sample of the inverse problem (e.g., a single waveform) independently of the other samples. In contrast, our GFI framework belongs to a completely different category of *per-domain* approaches, where the trained model can solve forward and inverse problems for any input sample. This offers generalizability across the entire domain of input and output samples, while enabling fast “online” predictions without per-instance optimization.
>
> > We did not compare our results with these prior works as per-sample methods are outside the scope of GFI, where we are interested in analyzing shared latent spaces for jointly solving forward and inverse problems. Direct comparisons were also not feasible as their code and model checkpoints are not publicly available.
>
> > Regarding the comment on using recent baselines, we would like to emphasize that we compare our results with AutoLinear [5], a 2024 ICML publication that is the most recent state-of-the-art baseline in the field.
>
> > [1] Lin, Youzuo, et al. "Physics and Deep Learning in Computational Wave Imaging." arXiv preprint arXiv:2410.08329 (2024).
>
> > [2] P. Sun, F. Yang, H. Liang and J. Ma, "Full-Waveform Inversion Using a Learned Regularization," in IEEE Transactions on Geoscience and Remote Sensing, vol. 61, pp. 1-15, 2023, Art no. 5920715, doi: 10.1109/TGRS.2023.3322964.
>
> > [3] B. Du, J. Sun, A. Jia, N. Wang and H. Liu, "Physics-Informed Robust and Implicit Full Waveform Inversion Without Prior and Low-Frequency Information," in IEEE Transactions on Geoscience and Remote Sensing, vol. 62, pp. 1-12, 2024, Art no. 5918712, doi: 10.1109/TGRS.2024.3416547.
>
> > [4] F. Yang and J. Ma, "Wasserstein Distance-Based Full-Waveform Inversion With a Regularizer Powered by Learned Gradient," in IEEE Transactions on Geoscience and Remote Sensing, vol. 61, pp. 1-13, 2023, Art no. 5904813, doi: 10.1109/TGRS.2023.3241723.
>
> > [5] Y. Feng, Y. Chen, P. Jin, S. Feng, Y. Lin, "Auto-Linear Phenomenon in Subsurface Imaging ." ICML, PMLR:235, 2024, pp. 13153-13174.

---

> ### Author Response · Authors · 2024-11-23
>
> **Comment 3: While the authors aim to model the forward problem using neural networks, it is uncertain if the learned mapping can capture complex physical dynamics, as seen in the governing equations of seismic data. Extensive out-of-domain experiments are recommended to assess the model’s robustness and generalizability to real-world data.**
> > We agree with the reviewer that extensive out-of-domain experiments are required to validate our performance on forward problems. Toward this goal, we have devoted an entire Section 5.2 to evaluate zero-shot generalization capabilities of our models on both forward and inverse problems. First, we demonstrate out-of-distribution generalization across all OpenFWI datasets, where a model trained on a certain dataset is tested on all remaining datasets (see Appendix E.4 to E.6 for additional visualizations). Second, we perform zero-shot evaluation on two real-world benchmarks widely used in seismic imaging studies, Marmousi and Overthrust. Quantitative results for these datasets are presented in Appendix Section E.2, with visualizations included both in the main paper and the Appendix. Similar out-of-domain evaluations have been performed in previous state-of-the-art works as well [1].
>
> > To further assess the robustness of our models, we have conducted new experiments analyzing the effects of adding noise to the inputs of forward and inverse problems. In these experiments, Gaussian noise with varying variances is added to the input velocity or waveform, and the models' predictions are evaluated based on the noisy inputs. We observe that for the inverse problem, both Latent U-Net and Invertible X-Net demonstrate robust performance even under noisy conditions. In the forward case, Invertible X-Net has the highest SSIM and the lowest MAE even at the lowest PSNR signal. Detailed results are provided in Appendix Tables 11–14 for CurveFault-B and FlatFault-B datasets, covering both forward and inverse problems.
>
> > [1] Y. Feng, Y. Chen, P. Jin, S. Feng, Y. Lin, "Auto-Linear Phenomenon in Subsurface Imaging ." ICML, PMLR:235, 2024, pp. 13153-13174.
>
> **Comment 4: The paper's structure could be streamlined to highlight the proposed methodology. Currently, a substantial background section precedes a brief description of the methods. Specific sections of the introduction and related work and preliminary knowledge could be condensed.**
> > We appreciate the reviewer’s suggestion and will try to streamline the introduction and related works sections to reduce redundancy and increase focus on our contributions. However, we would like to emphasize that the main contribution of our work is introducing the unified framework of GFI, which subsumes previous works in the field that can be viewed as special cases of GFI. As a result, we begin Section 3 on “Proposed Methodologies” by first introducing GFI and then describing how previous works are different instantiations of GFI (which is also a novel contribution), before discussing the proposed architectures of Latent U-Net and Invertible X-Net. We will revise Section 3 to bring more prominence to our contributions and emphasize the connections of prior works to GFI.
>
> **Comment 5: The model specifications lack detail; adaptations made to standard UNet for subsurface imaging and the mechanisms enabling X-Net’s invertibility are unclear. Clarifying these aspects and specifying when each model might be preferred would improve understanding.**
> > To provide more details of our model specifications, we have added two new Sections in the Appendix (D.3.1 and D.3.2) that explain how U-Net adaptations are tailored for subsurface imaging and summarize the architecture of U-Nets and iUNets used as backbones in our proposed models, including the type and number of blocks and parameter counts (also described in Appendix Tables 3 to 6). Additionally, the code has been made available to ensure full transparency and reproducibility. Our approach leverages the strengths of U-Nets, widely successful in computer vision, to be applied in the domain of subsurface imaging.
>
> > Regarding the comment on when we should choose one model over another (e.g., Invertible X-Net over Latent U-Net), we would like to refer to our global response on this comment. In summary, we recommend using Invertible X-Net if the goal is to solve both forward and inverse problems, or just the forward problem. We recommend Latent U-Net (Large) if the goal is to only solve the inverse problem. We have included this explicit recommendation in the main paper.

---

> > ### Author Response · Authors · 2024-11-23
> >
> > **Comment 6: The Invertible X-Net significantly underperforms other models, raising questions about its ideal use cases. Further context on scenarios where the X-Net's design offers advantages would be valuable.**
> > > As stated in our global response, Invertible X-Net shows best performance for forward problems and second-best performance for inverse problems across all 10 OpenFWI datasets based on the average ranks of metrics (see Appendix Tables 7 and 9 for more details). Additionally, Invertible X-Net is highly parameter-efficient when it comes to jointly solving forward and inverse problems, requiring only 26M parameters compared to the 70M parameters of Latent U-Net (Large).
> >
> > **Comment 7: Key numerical results in Section 5 are deferred to the appendix, which reduces clarity. Summarizing key findings directly in the main text would enhance readability.**
> > > To improve readability, we summarize key numerical findings in the main text and retain detailed tables in the appendix to save space. Additionally, we have included average rank metrics in the tables and have discussed those in the main paper. This ensures that the main paper highlights essential insights without overwhelming the reader with detailed tables.
> >
> > > As stated in our global response, we have now included the average rank of every model across all 10 OpenFWI datasets to summarize key findings. We will use these average ranks to discuss results throughout the main paper, as described below:
> >
> > > For the inverse problem Latent U-Net (Large) achieves the best SSIM performance with average rank 1.3 and Invertible X-Net ranks second with average SSIM rank 1.9. For the forward problem Invertible X-Net performs the best with average SSIM average rank 1.1, followed by Latent U-Net with average SSIM rank 2.3.

---

> > > ### Comment · Reviewer_VWh3 · 2024-11-25
> > > **Thank you for the substantial revisions.**
> > >
> > > I appreciate the authors' responses, which address my previous concerns about the scope of this research. Additionally, section 5.2 demonstrates the model's zero-shot capability, providing strong evidence of generalization. While I acknowledge the broad value of this study in other inverse problem applications, I believe the transfer adaptations may prove non-trivial. From my perspective as a medical imaging researcher, I did not find direct insights applicable to my field. Another concern is that comments 4 and 5 regarding the writing have not been sufficiently addressed. Nonetheless, this is a solid study overall, albeit with a somewhat limited audience. I will raise my score and thank the authors for their discussions.

---

> > > > ### Author Response · Authors · 2024-11-25
> > > > **Thank you!**
> > > >
> > > > Thank you very much for your feedback and support! We will work on improving the readability of the paper by addressing comments 4 and 5 suggested by the reviewer in the revised version.

---

### Official Review · Reviewer_Lx2Y · 2024-11-02

**Soundness:** 3
**Presentation:** 4
**Contribution:** 3
**Rating:** 8
**Confidence:** 4

**Summary:**

Subsurface imaging is a technique for identifying the geophysical properties of layers underneath the Earth's surface. There are two directions: learning the mapping from velocity maps to seismic waveforms, called the forward problem, and the inverse problem which is the mapping from seismic data to velocity maps. Most of the previous deep learning-based research in the field focuses only on the inverse problem also referred to as full waveform inversion (FWI). This paper proposes a generalized forward-inverse (GFI) framework that unifies both directions, i.e., FWI and the forward problem. The framework builds on the assumption of manifolds and latent space translations, proposing two model architectures for the latter, namely Latent U-Net and Invertible X-Net. Latent U-Net architecture employs two U-Nets for translation between the velocity and waveform latent representations and vice versa, while Invertible X-Net uses a single IU-Net that simultaneously learns forward and inverse translations. The GFI framework encompasses previous works in deep learning for subsurface imaging, at the same time trying to answer questions such as the effect of latent space size, the importance of manifold learning and the value of jointly solving forward and inverse problems. The models were evaluated on the synthetic OpenFWI dataset and their generalization ability was tested on two real-world-like datasets.

**Strengths:**

- The paper is clearly presented and well-organized.
- Even though the GFI framework is not the first idea that unifies seismic imaging forward and inverse problems (e.g., Auto-Linear [1]), it tries to systematically characterize and unify prior research in this area. The paper answers a few questions that were open in the research field. The paper concludes the following: the size of the latent space should be decided based on the complexity of the problem, the training of the latent spaces should be influenced based on the translation objectives and jointly solving forward and inverse problems is helping the model to achieve better forward solutions. In my opinion, these are answers to important questions that were backed up by the experiments in the paper.
- The paper introduces U-Net and IU-Net architectures for learning translations between velocity and seismic waveforms in the latent space. This leads to two new architectures for unified seismic imaging, namely Latent U-Net and Invertible X-Net. In comparison, the Auto-Linear framework's architecture comprises two separate linear layers trained for the forward and inverse translations in the latent space.
- Extensive and systematic experiments. The proposed approach was compared to the existing state-of-the-art methods both for the forward and inverse seismic imaging problems. The proposed method employing Latent U-Net architecture outperforms the existing methods in most of the datasets in both forward and inverse directions.

---
[1] Y. Feng, Y. Chen, P. Jin, S. Feng, Y. Lin, "Auto-Linear Phenomenon in Subsurface Imaging
." ICML, PMLR:235, 2024, pp. 13153-13174.

**Weaknesses:**

- Section 5.1.1. "What is the effect of latent space sizes on translation performance?" is kind of vague. It states that the ideal size of the latent space should be decided based on the complexity of the problem being solved. It is not clear how to determine the complexity in real-world applications. I think the authors should be more specific in this section.
- In Section 5.1.2. "Do we need complex architectures for translations?", there is a comparison between large and small Latent U-Net. The only difference was the complexity (size) of the U-Nets for the latent space translation and the influence of having skip connections. What is the relation between the number of parameters in encoders and decoders and the U-Net and how does it influence performance? The Invertible X-Net was not included in the experiments and only one size was used throughout the paper. It would be useful to see how its size influences the performance.
- The Invertible X-Net model achieves worse results than the Latent U-Net model for the inversion problem and often fails to outperform other state-of-the-art methods. In contrast, it outperforms most of the methods for the forward problem. It is not clear from the paper what are the benefits of having such a unified framework. I think the limitations should be stated more clearly in the paper.
- In the results section, in qualitative comparison, Figure 4. and Figure 8. are missing a measure of error (at least MAE). The picked images should have approximately the same MAE as the reported results in the tables presenting a realistic case and fair comparison. It is not clear whether the presented results were in a sense cherry-picked, e.g., the proposed method achieved better results than the average and other methods below the average for the illustrated example.
- Instead of having a qualitative comparison of the methods on the two real-world-like datasets, I think it would be much more representative to have a quantitative comparison. Here, the presented examples might also be cherry-picked.

**Questions:**

1. Could you comment on the complexity of the problems and how to "calculate" it when choosing the latent space size? How to choose this size for real-world applications and are there any limitations in real-world scenarios?
2. Why have you decided to remove Invertible X-Net from the experiments in Section 5.1.2.? Have you tried different model sizes for this architecture?  What is the relation between the number of parameters in encoders and decoders and the U-Net and how does it influence performance?
3. It is not clear from the paper what are the benefits of having such a unified framework. Could you comment on the limitations? What would be your recommendation on which architecture to use and when?
4. Could you comment on the examples in Figure 4. and Figure 8. and the performance of the models achieved in terms of MAE?

---

> ### Author Response · Authors · 2024-11-23
>
> **Comment 1:  Section 5.1.1 states that the ideal size of the latent space should be decided based on the “complexity” of the problem being solved. It is not clear how to determine the complexity in real-world applications.**
>
> > We apologize for the lack of clarity in our wording. We meant to say that the latent space size is a hyper-parameter that needs be determined based on the *needs” of the problem. Intuitively, problems with simpler geological structures and velocity variations (e.g., A-family of datasets in OpenFWI) can be solved with smaller latent dimensions, since the amount of information that we need to encode is also small. On the other hand, problems with more complex structures (e.g., B-family of datasets) require large latent space sizes to encode more sophistical patterns in the waveform and velocity fields. We will revise Section 5.1.1 to clarify this point.
>
> **Comment 2:  What is the relation between the number of parameters in encoders and decoders and the U-Net and how does it influence performance?**
>
> > Appendix Table 6 provides a detailed comparison of the number of parameters in the encoder-decoder and translation models across different baselines. While Latent U-Net (Large), Latent U-Net (Small), and Invertible X-Net maintain a similar ratio between these components, AutoLinear represents a contrasting case with a more complex encoder-decoder and a lightweight translation model. Despite having a similar total number of parameters to Latent U-Net (Large), AutoLinear demonstrates different performance characteristics, indicating that the parameter ratio is not the sole factor influencing performance. Architectural differences, such as skip connections, latent dimension size, and the choice of translation architecture (e.g., U-Net vs. iUNet), also play significant roles, as evaluated in Figure 5.
>
> > We also agree that conducting more ablations of Latent U-net with different ratios of encoder/decoder to translation model parameters is an important direction of research, which we could not complete in the time constraints of the discussion period but are happy to explore in future works.
>
> **Comment 3: Invertible X-Net was not included in the experiments on varying model size and only one size was used throughout the paper. It would be useful to see how its size influences the performance.**
> > To address this, we have added a new plot in Figure 5(c) of the main paper that shows how Invertible X-Net's performance changes with varying latent dimensions across 8 OpenFWI datasets, similar to the analysis done for Latent U-Net (Large) in Figure 5(a). We observe a similar trend that the performance of Invertible X-Net does not mostly change with latent size for A-family of datasets, while it shows a decreasing trend for B-family.
>
> > To answer why we only chose one size for Invertible X-Net while for Latent U-Net we had two versions (small and large), we would like to remind that we wanted to keep the number of parameters in our proposed models (Latent U-net and Invertible X-Net) similar to the range of parameters used in other baselines like InversionNet. Since the iUNet used in Invertible X-Net requires two coupling blocks for every convolution in a standard U-Net, adjusting the overall model size of Invertible X-Net required a significant change in total parameters. For instance, doubling the number of invertible blocks in the iUNet increases parameters from 25M to 50M, while halving it reduces the parameters to 13M. As a result, a larger version of Invertible X-Net would have significantly more parameters than other models, while a smaller version would have too few parameters leading to underfitting.

---

> > ### Author Response · Authors · 2024-11-23
> >
> > **Comment 4: The Invertible X-Net model achieves worse results than the Latent U-Net model for the inversion problem and often fails to outperform other state-of-the-art methods. In contrast, it outperforms most of the methods for the forward problem. It is not clear from the paper what are the benefits of having such a unified framework. Could you comment on the limitations? What would be your recommendation on which architecture to use and when?**
> > > As stated in our global response, Invertible X-Net shows best performance for forward problems and second-best performance for inverse problems across all 10 OpenFWI datasets based on the average ranks of metrics (see Appendix Tables 7 and 9 for more details). Additionally, Invertible X-Net is highly parameter-efficient when it comes to jointly solving forward and inverse problems, requiring only 26M parameters compared to the 70M parameters of Latent U-Net (Large). Hence, we recommend using Invertible X-Net if the goal is to solve both forward and inverse problems, or just the forward problem. We recommend Latent U-Net (Large) if the goal is to only solve the inverse problem. We have included this explicit recommendation in the main paper.
> >
> > > Regarding the limitations of our work, while the GFI framework currently offers a fully data-driven solution, its inability to directly incorporate physics knowledge represents a limitation. We believe the integration of physics-informed constraints into this framework is the next step in its evolution, enabling greater accuracy, interpretability, and applicability in real-world seismic imaging scenarios. This direction highlights the framework’s potential to bridge the gap between data-driven and physics-informed methods while unifying and extending prior research in this field.
> >
> >
> > **Comment 5:  Figures 4 and 8 are missing a measure of error (at least MAE). The picked images should have approximately the same MAE as the reported results in the tables presenting a realistic case and fair comparison. It is not clear whether the presented results were in a sense cherry-picked, e.g., the proposed method achieved better results than the average and other methods below the average for the illustrated example.**
> >
> >  > As stated in the global response, we have updated Figures 4 and 8 to include both MAE and SSIM metrics. All examples have been randomly generated. Additionally, we provide more visualization examples with MAE/SSIM values for each dataset in Appendix Section G (Additional Visualizations) for completeness. Furthermore, we have submitted the code for generating additional results and plan to release model checkpoints to ensure full reproducibility and transparency of our results.
> >
> >
> > **Comment 6:  Instead of having a qualitative comparison of the methods on the two real-world-like datasets, I think it would be much more representative to have a quantitative comparison. Here, the presented examples might also be cherry-picked.**
> >
> > > We agree. We have included quantitative results for the two real-world datasets: Marmousi and Overthrust in Appendix Section E (Additional Results) in the forms of Tables 8 and 10. We have also updated the visualizations for these datasets in both the main paper and the Appendix to include MAE and SSIM metrics. For these datasets, all available samples were used for testing, as only two were provided by the authors of the AutoLinear paper, and no additional samples are publicly accessible.
> >
> > > Here is a summary of the key observations from the quantitative results on Marmousi and Overthrust datasets:
> >
> > > 1. For the inverse problem, Invertible X-Net achieves the highest SSIM on both Marmousi (0.504) and Overthrust (0.483), with the second-best model being InversionNet for Marmousi (SSIM: 0.46) and Latent U-Net for Overthrust (SSIM: 0.481).
> > > 2. For the forward problem, Latent U-Net (Large) achieves the highest SSIM on both Marmousi (0.834) and Overthrust (0.770), followed by Invertible X-Net on both datasets with SSIMs of 0.827 and 0.786 for Marmousi and Overthrust, respectively.
> >
> > > We can see that these observations are similar to the ones we obtained on the OpenFWI datasets, with Latent U-Net (Large) and Invertible X-Net generally occupying the top two positions.

---

> > > ### Comment · Reviewer_Lx2Y · 2024-11-23
> > > **Thank you for your response**
> > >
> > > Dear authors,
> > >
> > > thank you very much for your responses. Having carefully reviewed both the other reviewers' comments and your responses, I think this is good work overall and worth accepting. I wish to maintain my current score.

---

> > > > ### Author Response · Authors · 2024-11-25
> > > > **Thank you!**
> > > >
> > > > Thank you very much for your feedback and support!

---

### Author Response · Authors · 2024-11-23
**Global Response to Review Comments**

We sincerely thank all the reviewers for their constructive and detailed feedback. We are encouraged that the reviewers appreciated several key aspects of our work and found:

1. Our Generalized Forward-Inverse (GFI) framework to be a significant contribution in unifying forward and inverse seismic imaging problems [*Reviewers Lx2Y, 74KD, VWh3*].
2. Our proposed Latent U-Net and Invertible X-Net architectures to be novel and effective, demonstrating state-of-the-art performance on most datasets [*Reviewers Lx2Y, nmvk*].
3. Our experimental evaluation to be extensive and systematic, including insights into key factors affecting performance on both forward and inverse problems [*Reviewers Lx2Y, 74KD, nmvk*].
4. Our writing to be clear and accessible even to researchers outside the field of subsurface imaging [*Reviewers Lx2Y, 74KD*].

To address the reviewer comments, we have performed additional experiments and added new figures, tables, and text explanations in the revised manuscript (revisions highlighted in red). Here is a **summary of the new experiments:**

1. Analyzed effect of adding noise on forward and inverse problems for Flat Fault-B and Curve Fault-B datasets (Appendix Tables 11-14) (addressing Reviewer nmvk).
2. Performed ablations to study the effect of training with reconstruction loss and translation loss together for Latent U-Net (Large and Small) and Invertible X-Net (Figure 6 and Appendix Figure 22) (addressing Reviewer 74KD).
3. Analyzed effect of varying latent dimensions on Invertible X-Net’s performance for all OpenFWI datasets (Figure 5c) (addressing Reviewer Lx2Y).

Before addressing each of the reviewer’s comments individually, we address some of the shared concerns raised by the reviewers in our general response below:


**Comment: Show more evaluation metrics in visualizations; how are the visualization examples picked? (Lx2Y, 74KD)**

> We have updated the paper to report both MAE and SSIM metrics alongside all visualizations of velocity and waveform predictions in the main paper (Figures 4 and 8) and the appendix (Figures 23-34). We can see that the metrics of the chosen examples are representative of the numbers reported in the Appendix Tables 7 and 9. All examples used for visualizations in the main paper were randomly sampled so no cherry-picking was involved. We also show several other visualization examples in the Appendix.

**Comment: Why choose Invertible X-Net when it is not always the best? What is the advantage of using Invertible X-Net compared to Latent U-Net? (Lx2Y, vWh3, 74KD)**

> To analyze the overall performance of Invertible X-Net across all 10 datasets of OpenFWI, we have now included the average rank of all methods across all datasets for metrics: MAE, MSE, and SSIM in Appendix Tables 7 and 9. Lower average rank is better. We summarize this analysis of average ranks in the tables below.

***Average Ranks for the Inverse problem:***

| **Model** | **MAE Rank** | **MSE Rank** | **SSIM Rank** |
|-----------------------|----------|----------|-----------|
| Inversion Net     | 2.8        | 2.8        | 3       |
| Auto-Linear        | 3.5        | 3.5        | 3.8    |
| Latent U-Net      | 1.3        | 1.3        | 1.3    |
| Invertible X-Net  | 2.7        | 2.4        | 1.9    |

***Average Ranks for the Forward problem:***

| **Model** | **MAE Rank** | **MSE Rank** | **SSIM Rank** |
|-----------------------|----------|----------|-----------|
| FNO                   | 2.4        | 3.1        | 3.1    |
| Auto-Linear        | 3.9        | 3.5        | 3.5    |
| Latent U-Net      | 2.3        | 2.6        | 2.3    |
| Invertible X-Net  | 1.4        | 1.8        | 1.1    |


Note that it is important to consider all metrics because while SSIM is more suited to capture visually perceptible changes related to geological similarity, MAE and MSE measure average errors across entire domains. From these tables, we can make the following observations:
- **For inverse problems:** Latent U-Net (Large) achieves the *best* performance with an average SSIM rank of 1.3. Invertible X-Net is a *close second* with an average SSIM rank of 1.9.
- **For forward problems:** Invertible X-Net performs the *best* across all metrics with an average SSIM rank of 1.1, followed by Latent U-Net that has an average SSIM rank of 2.3.

Based on these observations, we make the following recommendations:
- If we **only** want to solve **inverse** problems, use Latent U-Net (Large).
- If we **only** want to solve **forward** problems, use Invertible X-Net.
- If we want to solve **both forward and inverse** problems, use Invertible X-Net. Even though Invertible X-Net has second-best performance on inverse problems, it is highly parameter-efficient as it only requires 26M parameters to jointly solve forward and inverse while Latent U-Net (Large) requires a total of 70M parameters (35M x 2).

We have included these recommendations explicitly in Section 5.1.4 to make this analysis reader-friendly.

---

> ### Author Response · Authors · 2024-11-23
> **Continued Global Response**
>
> **Comment: Summarize Key Findings from Appendix Tables and Figures in the Main Paper (vWh3, 74KD, nmvk)**
>  > Thank you for this suggestion. To summarize the key findings across all 10 OpenFWI datasets used in our paper, we have now added a new column of average rank for every model in the Appendix Tables 7 and 9. We will use these average ranks in our discussions of results throughout the main paper. To improve the readability of the Appendix section from the main paper, we have now also added cross-references in the main paper to specific Appendix sections wherever appropriate.
>
>  We hope that our responses to the individual review comments provided below address the main concerns of the reviewers. If we missed any detail, we will be happy to provide more clarifications during the discussion period. If our responses have adequately addressed your concerns, we kindly request that you consider revising your scores. Thank you very much for your time and effort.

---

> > ### Author Response · Authors · 2024-11-26
> > **Closing Remarks**
> >
> > We thank the reviewers for their valuable feedback and support throughout the review process. Their insights have been instrumental in refining our work.
> >
> > We have submitted the final revised PDF,  incorporating changes to address the reviewers' comments and suggestions.  We look forward to any further feedback or questions during the discussion period.
> >
> > Thank you for your time and consideration.

---

### Meta-Review · Area_Chair_Rv3f · 2024-12-21

**Metareview:**

This paper proposes a joint forward--inverse solver for wave imaging problems using two autoencoders whose latent spaces (of possibly different dimensions) are paired via learned "latent space translations". The reviewers praised clarity of presentation, and extensive experiments. Initially there was some concern about the contribution and vagueness, but this has been in good part eliminated in the rebuttal phase. My opinion is that this is a nice paper very slightly above the acceptance bar. It is clear, the prior work section is very nice, and the experiments are complete and informative. I will recommend acceptance but I would not mind if it ends up being rejected. The reason for hesitation is that there is a lot of very related work which has not been acknowledged or compared to, for example

https://arxiv.org/pdf/2405.13220
https://arxiv.org/pdf/2305.13314

but also that I think that as a community we should finally go beyond OpenFWI benchmarks with 2D 70x70 pixel, highly stylized, layered velocity models which have a handful of degrees of freedom.

**Additional Comments On Reviewer Discussion:**

This was a calm, respectful discussion. Lx2Y had a good opinion from the beginning but criticized vagueness and had some concerns about experiments, which were successfully addressed. VWh3 thought the focus is narrow and wondered whether anything "physical" is captured by learned "translations" (I think this is an important question, and I think that the answer is more "no" than "yes", since the data is much too simple here...). They also criticized lack of detail and weak baseline. The authors made many changes and the reviewer raised the score, but remained slightly in the rejection territory. 74KD was generally positive, clarity and originality (and rightly so). Authors' responses prompted them to bump the score to weak accept. nmvk thought the paper is like a survey and had novelty concerns. The authors successfully addressed some of those and the score went up to 6. My recommendation and meta-review are based on the careful study of the discussions, the relative consensus that the work has merit, and on personal familiarity with this space.

---

### Decision · Program_Chairs · 2025-01-22

Accept (Poster)